# Reinforcement Learning with Segment Feedback

## Abstract

Classic reinforcement learning (RL) assumes that an agent can observe a reward for each state-action pair. However, in practical applications, it is often difficult and costly to collect a reward for each state-action pair. While there have been several works considering RL with trajectory feedback, it is unclear if trajectory feedback is inefficient for learning when trajectories are long. In this work, we consider a model named RL with segment feedback, which offers a general paradigm filling the gap between per-state-action feedback and trajectory feedback seemlessly. In this model, we consider an episodic Markov decision process (MDP), where each episode is equally divided into $m$ segments, and the agent observes reward feedback only at the end of each segment. Under this model, we study two popular feedback settings: binary feedback and sum feedback, where the agent observes a binary outcome and a reward sum according to the underlying reward function, respectively. To investigate the impact of the number of segments $m$ on learning performance, we design efficient algorithms and establish regret upper and lower bounds for both feedback settings. Our theoretical and empirical results show that: under binary feedback, increasing the number of segments $m$ decreases the regret at an exponential rate; in contrast, surprisingly under sum feedback, increasing $m$ does not reduce the regret significantly.

## 1 Introduction

Reinforcement learning (RL) is a class of sequential decision-making algorithms, where an agent interacts with an unknown environment through time with the goal of maximizing the obtained reward. RL is applied to extensive fields such as robotics, autonomous driving and game playing.

In classic RL, when the agent takes an action in a state, the environment will provide a reward for this state-action pair. However, in real-world applications, it is often difficult and costly to collect a reward for each state-action pair. For example, in robotics, when we instruct a robot to scramble eggs, it is hard to specify a reward for each individual action. In autonomous driving, it is difficult and onerous to evaluate each action, considering multiple criteria including safety, comfort and speed.

Motivated by this fact, there have been several works that consider RL with trajectory feedback (Efroni et al., 2021; Chatterji et al., 2021). In these works, the agent observes a reward signal only at the end of each episode, instead of at each step, with the signal indicating the quality of the trajectory generated during the episode. While these works mitigate the issue of impractical per-step reward feedback in classic RL, the relationship between the frequency of feedback and the performance of RL algorithms is unknown. In particular, if for example we get feedback twice in each trajectory, does that significantly improve performance over once per trajectory feedback?

To answer this question, we study a general model called RL with segment feedback, which bridges the gap between per-state-action feedback in classic RL (Sutton & Barto, 2018) and trajectory feedback in recent works (Efroni et al., 2021; Chatterji et al., 2021). In this model, we consider an episodic Markov decision process (MDP), where an episode is equally divided into $m$ segments. In each episode, at each step, the agent first observes the current state, and takes an action, and then transitions to a next state according to the transition distribution. The agent *observes a reward signal at the end of each segment*. Under this model, we consider two reward feedback settings: binary feedback and sum feedback. In the binary feedback setting, the agent observes a binary outcome (e.g., thumbs up/down) generated by a sigmoid function of the reward on this segment. In the sum

feedback setting, the agent observes the sum of the rewards over this segment. In our model, the agent needs to learn the underlying reward function (i.e., the expected reward as a function of states and actions) from binary or sum segment feedback, and maximize the expected reward achieved. (Reviewer NUmy) While Tang et al. (2024) also studied this segment model before (they called it bagged reward), their work is mostly an empirical work, and did not provide theoretical guarantees for algorithms and rigorously reveal the influence of segments on learning.

This model is applicable to many scenarios involving human queries. For instance, in autonomous driving, a driving trajectory is often divided into several segments, and human annotators are asked to provide feedback for each segment, e.g., thumbs up/down. Compared to state-action pairs or whole trajectories, segments are easier and more efficient to evaluate, since human annotators can focus on and rate behaviors in each segment, e.g., passing interactions, reversing the car and park.

In this segment model, there is an interesting balance between the number of segments (queries to humans) and the collected observations, i.e., we desire more observations, but we also want to reduce the number of queries. Therefore, in this problem, it is critical to investigate the trade-off between the benefits brought by segments and the increase of queries, which essentially comes down to a question: *How does the number of segments $m$ impact learning performance?*

To answer this question, we design efficient algorithms for binary and sum feedback settings in both known and unknown transition cases. Regret upper and lower bounds are provided to rigorously show the influence of the number of segments on learning performance. We also present experimental results to validate our theoretical findings.

(Reviewer NUmy) (Reviewer R7pS) Note that studying RL with equal segments is an important start point and serves as a foundation for further investigation on more general models and analysis for RL with unequal segments. Even for the equal segment case, this problem is already very challenging: (i) This problem cannot be solved by applying prior trajectory feedback works, e.g., (Efroni et al., 2021), since they use the martingale property of subsequent trajectories in analysis, while subsequent segments are not a martingale due to the dependence of segments within a trajectory. (ii) In prior trajectory feedback works (Efroni et al., 2021; Chatterji et al., 2021), there exists a gap between upper and lower bounds for sum feedback, and there is no lower bound for binary feedback. This fact poses a great challenge for figuring out the essential influence of the number of segments $m$ on learning performance, since one cannot get too many hints from prior works.

Our work resolves the above challenges and makes the following contributions:

1. We study a general model called RL with segment feedback, which bridges the gap between per-state-action feedback in classic RL and trajectory feedback seemlessly. Under this model, we consider two feedback settings: binary feedback and sum feedback.

2. For binary feedback, we design computationally and sample efficient algorithms `SegBiTS` and `SegBiTS-Tran` for known and unknown transitions, respectively. We provide regret upper and lower bounds which depend on $\exp(\frac{H r_{\max}}{2m})$, where $H$ is the length of each episode, and $r_{\max}$ is the universal upper bound of rewards. Our results exhibit that under binary feedback, increasing the number of segments $m$ significantly helps accelerate learning.

3. For sum feedback, we devise algorithms `E-LinUCB` and `LinUCB-Tran` for known and unknown transitions, respectively, which achieve near-optimal regrets in terms of $H$ and $m$. We also establish lower bounds to validate the optimality, and show that optimal regrets do not depend on $m$. Our results reveal that surprisingly, under sum feedback, increasing the number of segments $m$ does not help expedite learning much.

4. We develop novel techniques which can be of independent interest, including the KL divergence analysis to derive an exponential lower bound under binary feedback, and the use of E-optimal experimental design in algorithm `E-LinUCB` to refine the eigenvalue of the covariance matrix and reduce the regret.

## 2 RELATED WORK

In this section, we briefly review prior works related to ours.

The algorithm design and theoretical analysis for classic RL were well studied in the literature (Sutton & Barto, 2018; Jaksch et al., 2010; Azar et al., 2017; Jin et al., 2018; Zanette & Brunskill, 2019). (Reviewer NUmy) Tang et al. (2024) proposed the RL with segment feedback problem (they called it RL with bagged reward) and designed a transformer-based algorithm. However, their work is mostly an empirical work and only considered the sum feedback type. They did not provide theoretical results for their algorithm and rigorously quantified the influence of segments on learning. There are two recent theoretical works (Efroni et al., 2021; Chatterji et al., 2021) studying RL with trajectory feedback, which are most related to our work. Efroni et al. (2021) investigated RL with sum trajectory feedback, and designed upper confidence bound (UCB)-type and Thompson sampling (TS)-type algorithms with regret guarantees. Chatterji et al. (2021) considered RL with binary trajectory feedback, and developed algorithms based on UCB value iteration. For binary trajectory feedback, Chatterji et al. (2021) provided regret upper bounds that exponentially depend on the scale of rewards.

Different from (Efroni et al., 2021; Chatterji et al., 2021), we study RL with segment feedback, which allows feedback from multiple segments within a trajectory, with per-state-action feedback and trajectory feedback as the two extremes. (Reviewer NUmy) Under sum feedback, we improve the regret bound in (Efroni et al., 2021) by $\sqrt{H}$ using experimental design, and demonstrate the optimality of our result by establishing a lower bound. Under binary feedback, we propose TS-style algorithms which are computationally efficient, and also build a lower bound to validate the inevitability of the exponential dependence in the regret bound, which is new to the literature.

Our work is also related to linear bandits (Abbasi-Yadkori et al., 2011) and logistic bandits (Filippi et al., 2010; Faury et al., 2020; Russac et al., 2021), and uses analytical techniques from that literature.

## 3 FORMULATION

In this section, we present the formulation of RL with binary and sum segment feedback.

We consider an episodic MDP denoted by $\mathcal{M}(\mathcal{S}, \mathcal{A}, H, r, p, \rho)$. Here $\mathcal{S}$ is the state space, and $\mathcal{A}$ is the action space. $H$ is the length of each episode. (Reviewer R7pS) $r : \mathcal{S} \times \mathcal{A} \to [-r_{\max}, r_{\max}]$ is an unknown reward function, where $r_{\max} > 0$ is a universal constant and used to prevent the input of binary feedback (the sigmoid function) from being too large. Define the reward parameter $\theta^* := [r(s,a)]_{(s,a) \in \mathcal{S} \times \mathcal{A}} \in \mathbb{R}^{|\mathcal{S}||\mathcal{A}|}$. $p : \mathcal{S} \times \mathcal{A} \to \triangle_{\mathcal{S}}$ is the transition distribution. For any $(s, a, s') \in \mathcal{S} \times \mathcal{A} \times \mathcal{S}$, $p(s'|s, a)$ is the probability of transitioning to $s'$ if action $a$ is taken in state $s$. $\rho \in \triangle_{\mathcal{S}}$ is the initial state distribution.

A policy $\pi : \mathcal{S} \times [H] \to \mathcal{A}$ is defined as a mapping from the state space and step indices to the action space, so that $\pi_h(s)$ specifies what action to take in state $s$ at step $h$. For any policy $\pi$, $h \in [H]$ and $(s, a) \in \mathcal{S} \times \mathcal{A}$, let $V_h^{\pi}(s)$ be the state value function, and $Q_h^{\pi}(s, a)$ be the state-action value function, which denote the cumulative expected reward obtained under policy $\pi$ till the end of an episode, starting from $s$ and $(s, a)$ at step $h$, respectively. Formally,

$$V_h^{\pi}(s) := \mathbb{E}\left[\sum_{t=h}^{H} r(s_t, a_t)|s_h = s, \pi\right], \quad Q_h^{\pi}(s, a) := \mathbb{E}\left[\sum_{t=h}^{H} r(s_t, a_t)|s_h = s, a_h = a, \pi\right].$$

The optimal policy is defined as $\pi^* = \arg\max_{\pi} V_h^{\pi}(s)$ for all $s \in \mathcal{S}$ and $h \in [H]$. For any $s \in \mathcal{S}$ and $h \in [H]$, denote $V_h^*(s) := V_h^{\pi^*}(s)$.

The process of RL with segment feedback is as follows. In each episode $k$, the agent chooses a policy $\pi^k$ at the beginning of this episode, and starts from $s_1^k \sim \rho$. At each step $h \in [H]$, the agent first observes the current state $s_h^k$, and takes an action $a_h^k = \pi_h^k(s_h^k)$ according to her policy, and then transitions to a next state $s_{h+1}^k \sim p(\cdot|s_h^k, a_h^k)$.

Each episode is equally divided into $m$ segments, and each segment is of length $\frac{H}{m}$. For convenience, assume that $H$ is divisible by $m$. For any $k > 0$ and $i \in [m]$, let $\tau^k = (s_1^k, a_1^k, \ldots, s_h^k, a_h^k)$ denote the trajectory in episode $k$, and $\tau_i^k = (s_{\frac{H}{m} \cdot (i-1)+1}^k, a_{\frac{H}{m} \cdot (i-1)+1}^k, \ldots, s_{\frac{H}{m} \cdot i}^k, a_{\frac{H}{m} \cdot i}^k)$ denote the $i$-th segment of the trajectory in episode $k$.

For any trajectory or trajectory segment $\tau$, $\phi^{\tau} \in \mathbb{R}^{|\mathcal{S}||\mathcal{A}|}$ denotes the vector where each entry $\phi^{\tau}(s, a)$ is the number of times $(s, a)$ is visited in $\tau$. For any policy $\pi$, $\phi^{\pi} \in \mathbb{R}^{|\mathcal{S}||\mathcal{A}|}$ denotes the vector where

---

**Algorithm 1:** `SegBiTS`

---

**Input:** $\delta, \delta' := \frac{\delta}{3}, \alpha := \exp(\frac{Hr_{\max}}{m}) + \exp(-\frac{Hr_{\max}}{m}) + 2, \lambda$.

1 **for** $k = 1, \ldots, K$ **do**

2 $\quad \hat{\theta}_{k-1} \leftarrow \operatorname{argmin}_\theta -(\sum_{k'=1}^{k-1} \sum_{i=1}^m (y_i^{k'} \cdot \log(\mu((\phi^{\tau_i^{k'}})^\top \theta)) + (1 - y_i^{k'}) \cdot \log(1 - \mu((\phi^{\tau_i^{k'}})^\top \theta))) - \frac{1}{2}\lambda\|\theta\|_2^2)$;

3 $\quad \Sigma_{k-1} \leftarrow \sum_{k'=1}^{k-1} \sum_{i=1}^m \phi^{\tau_i^{k'}}(\phi^{\tau_i^{k'}})^\top + \alpha\lambda I$;

4 $\quad$ Sample $\xi_k \sim \mathcal{N}(0, \alpha \cdot \nu(k-1)^2 \cdot \Sigma_{k-1}^{-1})$, where $\nu(k-1)$ is defined in Eq. (1);

5 $\quad \tilde{\theta}_k \leftarrow \hat{\theta}_{k-1} + \xi_k$;

6 $\quad \pi^k \leftarrow \operatorname{argmax}_\pi (\phi^\pi)^\top \tilde{\theta}_k$, where $\phi^\pi$ is defined in Eq. (3);

7 $\quad$ Play episode $k$ with policy $\pi^k$. Observe trajectory $\tau^k$ and binary segment feedback $\{y_i^k\}_{i=1}^m$;

---

each entry $\phi^\pi(s,a)$ is the expected number of times $(s,a)$ is visited in an episode under policy $\pi$, i.e., $\phi^\pi(s,a) := \mathbb{E}[\sum_{h=1}^H \mathbb{1}\{s_h = s, a_h = a\}|\pi]$.

In our model, the agent *observes reward feedback only at the end of each segment*, instead of each step as in classic RL. We consider two reward feedback settings as follows.

**Binary Segment Feedback.** Denote the sigmoid function by $\mu(x) := \frac{1}{1+\exp(-x)}$ for any $x \in \mathbb{R}$. In the binary segment feedback setting, in each episode $k$, at the end of each segment $i \in [m]$, the agent observes a binary outcome

$$y_i^k = \begin{cases} 1, & \text{w.p. } \mu\left(-\sum_{t=\frac{H}{m}\cdot(i-1)+1}^{\frac{H}{m}\cdot i} r(s_t^k, a_t^k)\right) = \mu((\phi^{\tau_i^k})^\top \theta^*), \\ 0, & \text{w.p. } 1 - \mu\left(-\sum_{t=\frac{H}{m}\cdot(i-1)+1}^{\frac{H}{m}\cdot i} r(s_t^k, a_t^k)\right) = 1 - \mu((\phi^{\tau_i^k})^\top \theta^*). \end{cases}$$

**Sum Segment Feedback.** In the sum segment feedback setting, in each episode $k$, at each step $h$, the environment generates an underlying random reward $R_h^k = r(s_h^k, s_h^k) + \varepsilon_h^k$, where $\varepsilon_h^k$ is a zero-mean and 1-Sub-Gaussian noise, and independent of the transition distribution. At the end of each segment $i \in [m]$, the agent observes the sum of random rewards

$$R_i^k = \sum_{t=\frac{H}{m}\cdot(i-1)+1}^{\frac{H}{m}\cdot i} R(s_t^k, a_t^k) = (\phi^{\tau_i^k})^\top \theta^* + \sum_{t=\frac{H}{m}\cdot(i-1)+1}^{\frac{H}{m}\cdot i} \varepsilon_t^k.$$

In the sum feedback setting, when $m = H$, our model degenerates to classic RL (Azar et al., 2017; Sutton & Barto, 2018). When $m = 1$, the above two settings reduce to the problems of RL with binary (Chatterji et al., 2021) and sum trajectory feedback (Efroni et al., 2021), respectively.

In our model, the agent needs to infer the reward function from sparse and implicit reward feedback. Let $K$ denote the number of episodes played. The goal of the agent is to minimize the cumulative regret, which is defined as $\mathcal{R}(K) := \sum_{k=1}^K (V_1^*(s_1) - V_1^{\pi^k}(s_1))$.

## 4 REINFORCEMENT LEARNING WITH BINARY SEGMENT FEEDBACK

In this section, we investigate RL with binary segment feedback. To isolate the effect of segment feedback from transition model learning, we first design a computationally-efficient and sample-efficient algorithm `SegBiTS` for the known transition case with a regret guarantee, and establish a nearly matching regret lower bound. Then, we further develop an algorithm `SegBiTS-Tran` with carefully-designed transition bonuses for the unknown transition case, and provide a regret analysis.

### 4.1 ALGORITHM `SegBiTS` AND REGRET UPPER BOUND FOR KNOWN TRANSITION

Building upon the Thompson sampling algorithm (Thompson, 1933), `SegBiTS` adopts the maximum likelihood estimator (MLE) to learn rewards from binary feedback, and performs posterior sampling to compute the optimal policy. (Reviewer NUmy) Different from prior trajectory feedback

algorithms (Chatterji et al., 2021) which are either computationally inefficient or have an $O(K^{\frac{2}{3}})$ regret bound, `SegBiTS` is both computationally efficient and has an $O(\sqrt{K})$ regret bound.

Algorithm 1 presents the procedure of `SegBiTS`. Specifically, in each episode $k$, `SegBiTS` first employs MLE with past binary reward observations to obtain the estimated reward parameter $\hat{\theta}_{k-1}$ (Line 2). Then, `SegBiTS` calculates the feature covariance matrix of past segments $\Sigma_{k-1}$ (Line 3). After that, `SegBiTS` samples a noise $\xi_k$ from Gaussian distribution $\mathcal{N}(0, \alpha \cdot \nu(k-1)^2 \cdot \Sigma_{k-1}^{-1})$ (Line 4). Here $\alpha$ represents the universal upper bound of the inverse of the sigmoid function's derivative. For any $k > 0$, we define

$$\nu(k) := \frac{m\sqrt{\lambda}}{H}\left(1 + \frac{Hr_{\max}\sqrt{|\mathcal{S}||\mathcal{A}|}}{m} + \frac{H}{m\sqrt{\lambda}}\sqrt{1 + \frac{Hr_{\max}\sqrt{|\mathcal{S}||\mathcal{A}|}}{m}}\omega(k) + \frac{H^2}{m^2\lambda}\cdot\omega(k)^2\right)^{\frac{3}{2}}, \quad (1)$$

and

$$\omega(k) := \sqrt{\lambda}\left(r_{\max}\sqrt{|\mathcal{S}||\mathcal{A}|} + \frac{1}{2}\right) + \frac{|\mathcal{S}||\mathcal{A}|}{\sqrt{\lambda}}\log\left(\frac{4}{\delta'}\left(1 + \frac{H^2 k}{4|\mathcal{S}||\mathcal{A}|\lambda m}\right)\right). \quad (2)$$

(Reviewer 1qnB) $\nu(k)$ stands for the confidence radius of the MLE estimate $\hat{\theta}_k$. With high probability, we have $|\phi^\top \theta^* - \phi^\top \hat{\theta}_k| \leq \sqrt{\alpha} \cdot \nu(k)\|\phi\|_{\Sigma_k^{-1}}$, where $\phi$ is the visitation indicator of any trajectory.

Adding noise $\xi_k$ to $\hat{\theta}_{k-1}$, `SegBiTS` obtains a posterior reward estimate $\tilde{\theta}_k$ (Line 5). Then, it computes the optimal policy $\pi^k$ under reward $\tilde{\theta}_k$ (Line 6). Note that the step in Line 6 is computationally efficient, which can be easily solved by any MDP planning algorithm, e.g., value iteration, with reward $\tilde{\theta}_k$. After obtaining $\pi^k$, `SegBiTS` plays episode $k$, and observes trajectory $\tau^k$ and binary feedback $\{y_i^k\}_{i=1}^m$ on each segment (Line 7).

Now we provide a regret upper bound for algorithm `SegBiTS`.

**Theorem 1.** *(Reviewer R7pS) With probability at least $1 - \delta$, for any $K > 0$, the regret of algorithm `SegBiTS` is bounded by*

$$\mathcal{R}(K) = \tilde{O}\Bigg(\exp\left(\frac{Hr_{\max}}{2m}\right)\cdot\left(\sqrt{Km|\mathcal{S}||\mathcal{A}|\max\left\{\frac{H^2}{m\alpha\lambda}, 1\right\}} + H\sqrt{\frac{K}{\alpha\lambda}}\right)\cdot\frac{m\sqrt{\lambda|\mathcal{S}||\mathcal{A}|}}{H}\cdot$$

$$\left(1 + \frac{Hr_{\max}\sqrt{|\mathcal{S}||\mathcal{A}|}}{m} + \frac{H}{m\sqrt{\lambda}}\sqrt{1 + \frac{Hr_{\max}\sqrt{|\mathcal{S}||\mathcal{A}|}}{m}}\left(\sqrt{\lambda}\left(r_{\max}\sqrt{|\mathcal{S}||\mathcal{A}|} + \frac{1}{2}\right) + \frac{|\mathcal{S}||\mathcal{A}|}{\sqrt{\lambda}}\right)\right.$$

$$\left.+ \frac{H^2}{m^2\lambda}\left(\sqrt{\lambda}\left(r_{\max}\sqrt{|\mathcal{S}||\mathcal{A}|} + \frac{1}{2}\right) + \frac{|\mathcal{S}||\mathcal{A}|}{\sqrt{\lambda}}\right)^2\right)^{\frac{3}{2}}\Bigg).$$

(Reviewer 1qnB) Theorem 1 exhibits that the regret of algorithm `SegBiTS` depends on $\exp(\frac{Hr_{\max}}{2m})$, which is usually the dominating factor. This implies that as the number of segments $m$ increases, the regret decays rapidly. Thus, under binary feedback, increasing the number of segments significantly helps accelerate learning. (Reviewer 1qnB) The intuition behind this is that when the reward scale $x = \frac{Hr_{\max}}{m}$ is large, the binary feedback is generated from the range where the sigmoid function $\mu(x) = \frac{1}{1+\exp(-x)}$ is flat, i.e., the derivative of the sigmoid function $\mu'(x)$ is small. Then, the generated binary feedback is likely always 0 or always 1, and it is hard to distinguish between a good action and a bad action, leading to a higher regret; On the contrary, when the reward scale $x = \frac{Hr_{\max}}{m}$ is small, the binary feedback is generated from the range where the sigmoid function $\mu(x)$ is steep, i.e., $\mu'(x)$ is large. Then, the generated binary feedback is more dispersed to be 0 or 1, and it is easier to distinguish between a good action and a bad action, leading to a lower regret. In other words, the regret bound depends on the inverse of the sigmoid function's derivative $\mu'(x) = \frac{1}{\exp(x)+\exp(-x)+2}$.

(Reviewer R7pS) In Theorem 1, the dependence on $|\mathcal{S}|, |\mathcal{A}|, H$ are $|\mathcal{S}|^3, |\mathcal{A}|^3, \exp(\frac{Hr_{\max}}{2m})H^2$, respectively. Since the exponential dependence on $\exp(\frac{Hr_{\max}}{m})$ usually dominates the bound, here we mainly aim to reveal such exponential influence on the regret, and are not pursuing the absolute tightness of every polynomial factor. Below we establish a lower bound to demonstrate the inevitability of this exponential dependence.

## 4.2 Regret Lower Bound for Known Transition

The lower bound for RL with binary segment feedback and known transition is as follows.

**Theorem 2.** *Consider the problem of RL with binary segment feedback and known transition. There exists a distribution of instances where for any $c_0 \in (0, \frac{1}{2})$, when $K \geq \exp(\frac{Hr_{\max}}{m}) \frac{4|\mathcal{S}||\mathcal{A}|m}{H^2 r_{\max}^2 c_0^2}$, the regret of any algorithm must be*

$$\Omega\left(\exp\left(\left(\frac{1}{2} - c_0\right)\frac{Hr_{\max}}{m}\right)\sqrt{|\mathcal{S}||\mathcal{A}|mK}\right).$$

Theorem 2 shows that the exponential dependence on $\frac{Hr_{\max}}{m}$ in the regret is indispensable, and the $\exp(\frac{Hr_{\max}}{2m})$ factor in Theorem 1 nearly matches the exponential dependence in the lower bound up to an arbitrarily small factor $c_0$ in $\exp(\cdot)$. Together with Theorem 1, Theorem 2 reveals that when the number of segments $m$ increases, the regret decreases at an exponential rate. (Reviewer NUmy) In addition, this regret lower bound also holds for the unknown transition case, by constructing the same problem instance as in this lower bound proof.

This lower bound and its analysis is novel to the literature. In analysis, we calculate the KL divergence of Bernoulli distributions with the sigmoid function as parameters. Then, we employ Pinsker's inequality and the fact that $\mu'(x) = \mu(x)(1 - \mu(x))$ to build a connection between the calculated KL divergence and $\mu'(\frac{Hr_{\max}}{m})$. Since $\mu'(\frac{Hr_{\max}}{m})$ contains an exponential factor, we can finally derive an exponential dependence in the lower bound.

## 4.3 Algorithm SegBiTS-Tran and Regret Upper Bound for Unknown Transition

In the following, we extend our results to the unknown transition case. We develop an efficient algorithm SegBiTS-Tran for binary segment feedback and unknown transition. SegBiTS-Tran includes a transition bonus $p_{k-1}^{pv}$ in posterior reward estimate $\tilde{\theta}_k^b$, and replaces visitation indicator $\phi^\pi$ by its estimate $\hat{\phi}_{k-1}^\pi$. For any $(s, a) \in \mathcal{S} \times \mathcal{A}$, $\hat{\phi}_{k-1}^\pi(s, a)$ is the expected number of times $(s, a)$ is visited in an episode under policy $\pi$ on empirical MDP $\hat{p}_{k-1}$, where $\hat{p}_{k-1}$ is the empirical estimate of transition distribution $p$. Then, SegBiTS-Tran finds the optimal policy via $\arg\max_\pi (\hat{\phi}_{k-1}^\pi)^\top \tilde{\theta}_k^b$, which can be efficiently solved by any MDP planning algorithm with transition $\hat{p}_{k-1}$ and reward $\tilde{\theta}_k^b$. We defer the pseudo-code and details of SegBiTS-Tran to Appendix C.3.

The regret performance of algorithm SegBiTS-Tran is stated below.

**Theorem 3.** *With probability at least $1 - \delta$, for any $K > 0$, the regret of algorithm SegBiTS-Tran is bounded by*

$$\tilde{O}\left(\exp\left(\frac{Hr_{\max}}{2m}\right)\nu(K)\sqrt{|\mathcal{S}||\mathcal{A}|}\left(\sqrt{Km|\mathcal{S}||\mathcal{A}|\max\left\{\frac{H^2}{m\alpha\lambda}, 1\right\}} + H\sqrt{\frac{K}{\alpha\lambda}}\right)\right.$$
$$\left. + \left(\nu(K)\sqrt{\frac{|\mathcal{S}||\mathcal{A}|}{\lambda}} + Hr_{\max}\right)|\mathcal{S}|^2|\mathcal{A}|^{\frac{3}{2}}H^{\frac{3}{2}}\sqrt{K}\right).$$

Similar to algorithm SegBiTS (Theorem 1), the regret of algorithm SegBiTS-Tran also has a dependence on $\exp(\frac{Hr_{\max}}{2m})$. When the number of segments $m$ increases, the regret of SegBiTS-Tran significantly decreases. Compared to SegBiTS, the regret of SegBiTS-Tran has an additional term polynomial in $|\mathcal{S}|$, $|\mathcal{A}|$, $H$ and $\sqrt{K}$, which is incurred by learning the unknown transition distribution.

## 5 Reinforcement Learning with Sum Segment Feedback

In this section, we turn to RL with sum segment feedback. (Reviewer NUmy) Different from prior trajectory feedback algorithm (Efroni et al., 2021) which directly uses the least squares estimator and has a suboptimal regret bound, we develop an algorithm E-LinUCB for the known transition case, which employs experimental design to perform an initial exploration and achieves a near-optimal regret with respect to $H$ and $m$. To validate the optimality, we further establish a regret lower bound.

---

**Algorithm 2:** E-LinUCB

---

**Input:** $\delta, \delta' := \frac{\delta}{3}, \lambda := \frac{H}{r_{\max}^2 m}$, rounding procedure ROUND, rounding approximation parameter

$$\gamma := \frac{1}{10}. \ \beta(k) := \sqrt{\frac{H|\mathcal{S}||\mathcal{A}|}{m} \log(1 + \frac{kH^2}{\lambda|\mathcal{S}||\mathcal{A}|m})} + 2\log(\frac{1}{\delta'})} + r_{\max}\sqrt{\lambda|\mathcal{S}||\mathcal{A}|}, \forall k > 0.$$

1 Let $w^* \in \triangle_\Pi$ and $z^*$ be the optimal solution and optimal value of the optimization:

$$\min_{w \in \triangle_\Pi} \left\| \left( \sum_{\pi \in \Pi} w(\pi) \left( \sum_{i=1}^m \mathbb{E}_{\tau_i \sim \pi} \left[ \phi^{\tau_i}(\phi^{\tau_i})^\top \right] \right) \right)^{-1} \right\| \tag{3}$$

2 $K_0 \leftarrow \lceil \max\{26(1+\gamma)^2(z^*)^2 H^4 \log(\frac{2|\mathcal{S}||\mathcal{A}|}{\delta'}), \frac{|\mathcal{S}||\mathcal{A}|}{\gamma^2}\} \rceil$ ;

3 $(\pi^1, \ldots, \pi^{K_0}) \leftarrow$ ROUND$(\{\sum_{i=1}^m \mathbb{E}_{\tau_i \sim \pi} \left[ \phi^{\tau_i}(\phi^{\tau_i})^\top \right]\}_{\pi \in \Pi}, w^*, \gamma, K_0)$ ;

4 Play $K_0$ episodes with policies $\pi^1, \ldots, \pi^{K_0}$. Observe trajectories $\tau^1, \ldots, \tau^{K_0}$ and rewards
$\{R_i^1\}_{i=1}^m, \ldots, \{R_i^{K_0}\}_{i=1}^m$ ;

5 **for** $k = K_0 + 1, \ldots, K$ **do**

6     $\hat{\theta}_{k-1} \leftarrow (\lambda I + \sum_{k'=1}^{k-1} \sum_{i=1}^m \phi^{\tau_i^{k'}}(\phi^{\tau_i^{k'}})^\top)^{-1} \sum_{k'=1}^{k-1} \sum_{i=1}^m \phi^{\tau_i^{k'}} R_i^{k'}$ ;

7     $\Sigma_{k-1} \leftarrow \lambda I + \sum_{k'=1}^{k-1} \sum_{i=1}^m \phi^{\tau_i^{k'}}(\phi^{\tau_i^{k'}})^\top$ ;

8     $\pi^k \leftarrow \operatorname{argmax}_{\pi \in \Pi}((\phi^\pi)^\top \hat{\theta}_{k-1} + \beta(k-1) \cdot \|\phi^\pi\|_{(\Sigma_{k-1})^{-1}})$ ;

9     Play episode $k$ with policy $\pi^k$. Observe trajectory $\tau^k$ and sum segment feedback $\{R_i^k\}_{i=1}^m$ ;

---

Moreover, we design an algorithm LinUCB-Tran equipped with a variance-aware transition bonus to handle the unknown transition scenario.

## 5.1 ALGORITHM E-LinUCB AND REGRET UPPER BOUND FOR KNOWN TRANSITION

If we regard visitation indicators $\phi^{\tau_i^k}$ as feature vectors and $\theta^*$ as the reward parameter, the problem of RL with sum segment feedback and known transition is similar to linear bandits.

Building upon the classic linear bandit algorithm LinUCB (Abbasi-Yadkori et al., 2011), algorithm E-LinUCB adopts the E-optimal design (Pukelsheim, 2006) to conduct an initial exploration to ensure sufficient coverage of the covariance matrix and further sharpen the norm under the inverse of the covariance matrix, resulting in a near-optimal regret.

As shown in Algorithm 2, the procedure of E-LinUCB is as follows. E-LinUCB first performs the E-optimal design to compute an optimal policy distribution $w^*$, which maximizes the minimum eigenvalue of the feature covariance matrix (Line 1). In Line 1, $\sum_{\pi \in \Pi} w(\pi)(\sum_{i=1}^m \mathbb{E}_{\tau_i \sim \pi}[\phi^{\tau_i}(\phi^{\tau_i})^\top])$ is the feature covariance matrix of segments under policy distribution $w$, and we assume that this matrix is invertible. Then, E-LinUCB calculates the number of samples $K_0$ for initial exploration according to the optimal value $z^*$ of the E-optimal design (Line 2).

Then, in Line 3, E-LinUCB calls a rounding procedure ROUND (Allen-Zhu et al., 2021) to transform sampling distribution $w^*$ into discrete sampling sequence $(\pi^1, \ldots, \pi^{K_0})$ which satisfies (see Appendix B for more details of ROUND)

$$\left\| \left( \sum_{k=1}^{K_0} \left( \sum_{i=1}^m \mathbb{E}_{\tau_i \sim \pi_k} \left[ \phi^{\tau_i}(\phi^{\tau_i})^\top \right] \right) \right)^{-1} \right\| \leq (1+\gamma) \left\| \left( K_0 \sum_{\pi \in \Pi} w^*(\pi) \left( \sum_{i=1}^m \mathbb{E}_{\tau_i \sim \pi} \left[ \phi^{\tau_i}(\phi^{\tau_i})^\top \right] \right) \right)^{-1} \right\|.$$

After that, E-LinUCB plays $K_0$ episodes with $(\pi^1, \ldots, \pi^{K_0})$ to perform initial exploration (Line 4). Owing to the E-optimal design, the covariance matrix of initial exploration $\Sigma_{K_0}$ has an optimized minimum eigenvalue, and then $\|\phi^\pi\|_{(\Sigma_{k-1})^{-1}}$ has a sharpened upper bound for any $k > K_0$. This is the key to the optimality of E-LinUCB with respect to $H$ and $m$.

After initial exploration, in each episode $k > K_0$, E-LinUCB first calculates the least squares reward estimate $\hat{\theta}_{k-1}$ with past reward observations and covariance matrix $\Sigma_{k-1}$ (Lines 6-7). Then, it computes the optimal policy with reward estimate $\hat{\theta}_{k-1}$ and bonus $\|\phi^\pi\|_{(\Sigma_{k-1})^{-1}}$ (Line 8). With the computed optimal policy $\pi^k$, E-LinUCB plays episode $k$, and collects trajectory $\tau^k$ and reward observation on each segment $\{R_i^k\}_{i=1}^m$ (Line 9).

Below we present a regret upper bound for algorithm `E-LinUCB`.

**Theorem 4.** *With probability at least $1 - \delta$, for any $K > 0$, the regret of algorithm* `E-LinUCB` *is bounded by*

$$O\left(|\mathcal{S}||\mathcal{A}|\sqrt{HK}\log\left(\left(1 + \frac{KHr_{\max}}{|\mathcal{S}||\mathcal{A}|m}\right)\frac{1}{\delta}\right) + (z^*)^2 H^5 \log\left(\frac{|\mathcal{S}||\mathcal{A}|}{\delta}\right) + |\mathcal{S}||\mathcal{A}|H\right).$$

(Reviewer 1qnB) Surprisingly, under sum feedback, when the number of segments $m$ increases, the regret bound does not decrease significantly as people might expect, e.g., at a rate of $\frac{1}{\sqrt{m}}$ or $\frac{1}{m}$. While this looks surprising at the first glance, we discover an *intuition* through analysis: The objective of RL measures the expected reward sum of an episode, namely, we only need to accurately estimate the expected reward sum of an episode. When the number of segments $m$ increases, although we obtain more observations, the segment features $\phi^{\tau_i^{k'}}$ contributed to covariance matrix $\Sigma_k$ shrink, which makes the reward estimation uncertainty $\|\phi^\pi\|_{(\Sigma_k)^{-1}}$ inflate. When we focus on the estimation performance of the expected reward sum of an episode, these two effects cancel out with each other, and the regret result is not influenced by $m$ distinctly.

(Reviewer NUmy) When $m = 1$, our problem reduces to RL with sum trajectory feedback (Efroni et al., 2021), and our result improves theirs by a factor of $\sqrt{H}$. This improvement comes from that we conduct the E-optimal design and perform an initial exploration to guarantee that $\|\phi^\pi\|_{(\Sigma_{k-1})^{-1}} \le 1$, instead of $\|\phi^\pi\|_{(\Sigma_{k-1})^{-1}} \le \frac{H}{\sqrt{\lambda}}$ as used in (Efroni et al., 2021).

Next, we investigate the lower bound to see if the number of segments $m$ does not influence the regret bound distinctly in essence.

## 5.2 REGRET LOWER BOUND FOR KNOWN TRANSITION

We establish a lower bound for RL with sum segment feedback and known transition as follows.

**Theorem 5.** *Consider the problem of RL with sum segment feedback and known transition. There exists a distribution of instances where the regret of any algorithm must be*

$$\Omega\left(\sqrt{|\mathcal{S}||\mathcal{A}|HK}\right).$$

Theorem 5 demonstrates that our regret upper bound for algorithm `E-LinUCB` (Theorem 4) is optimal with respect to $H$ and $m$ when ignoring logarithmic factors. In addition, this lower bound corroborates that the number of segments $m$ does not impact the regret result essentially.

## 5.3 ALGORITHM `LinUCB-Tran` AND REGRET UPPER BOUND FOR UNKNOWN TRANSITION

Now we study RL with sum segment feedback in the unknown transition scenario. For unknown transition, we design an algorithm `LinUCB-Tran`, which constructs a variance-aware uncertainty bound for the estimated visitation indicator $\hat{\phi}_k^\pi$, and takes into account this uncertainty bound in exploration bonuses. In analysis, we handle the estimation error of visitation indicators $\|\hat{\phi}_k^\pi - \phi^\pi\|_1$ by this variance-aware uncertainty bound, which enables us to achieve a near-optimal regret in terms of $H$. The pseudo-code and details of `LinUCB-Tran` are deferred to Appendix D.3.

In the following, we state the regret performance of algorithm `LinUCB-Tran`.

**Theorem 6.** *With probability at least $1 - \delta$, for any $K > 0$, the regret of algorithm* `LinUCB-Tran` *is bounded by*

$$\tilde{O}\left((1 + r_{\max})|\mathcal{S}|^{\frac{5}{2}}|\mathcal{A}|^2 H\sqrt{K}\right).$$

Theorem 6 shows that similar to algorithm `E-LinUCB`, the regret of `LinUCB-Tran` does not depend on the number of segments $m$ when ignoring logarithmic factors. The heavier dependence on $|\mathcal{S}|$, $|\mathcal{A}|$ and $H$ is due to the estimation of the unknown transition distribution. We also provide a lower bound for the unknown transition case, which demonstrates that the optimal regret indeed does not depend on $m$ and our upper bound is near-optimal in terms of $H$ (see Appendix D.5).

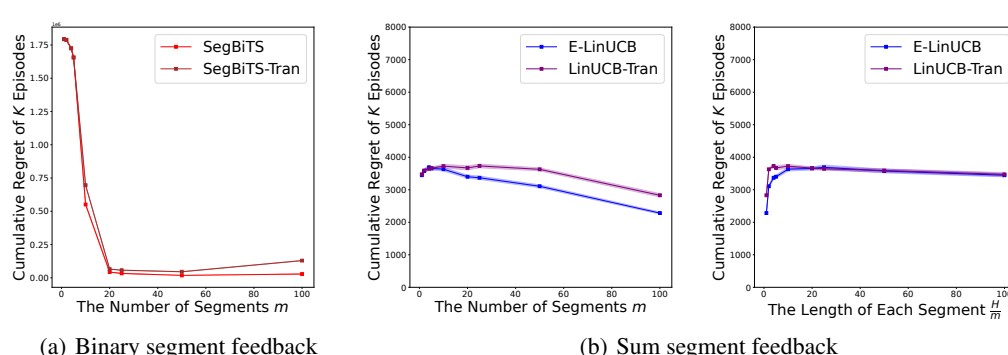

(a) Binary segment feedback

(b) Sum segment feedback

Figure 1: Experimental results for RL with binary or sum segment feedback.

## 6 EXPERIMENTS

Below we present experiments for RL with segment feedback to validate our theoretical results.

For the binary feedback setting, we evaluate our algorithms `SegBiTS` and `SegBiTS-Tran` in known and unknown transition environments, respectively, and we set $|\mathcal{S}| = 9$, $|\mathcal{A}| = 5$ and $K = 30000$. For the sum feedback setting, similarly, we run our algorithms `E-LinUCB` and `LinUCB-Tran` in known and unknown transition environments, respectively. Since `E-LinUCB` and `LinUCB-Tran` are computationally inefficient (mainly designed to exhibit the optimal dependence on $m$), we use a small MDP with $|\mathcal{S}| = 3$ and $|\mathcal{A}| = 5$, and set $K = 1000$. The details of the instances used in our experiments are described in Appendix A. In both settings, we set $r_{\max} = 0.5$, $\delta = 0.005$, $H = 100$ and $m \in \{1, 2, 4, 5, 10, 20, 25, 50, 100\}$. We perform 20 independent runs for each algorithm, and plot the average cumulative regret up to episode $K$ across runs with a 90% confidence interval.

Figure 1(a) draws the regrets of algorithms `SegBiTS` and `SegBiTS-Tran` under binary feedback. One sees that as the number of segments $m$ increases, the regret decreases rapidly. Specifically, when $m$ decreases from 20 to 1, i.e., $\frac{H}{2m}$ increases from $\exp(2.5)$ to $\exp(50)$, the regret grows explosively. This matches our theoretical results (Theorems 1 and 3) which show a dependence on $\exp(\frac{Hr_{\max}}{2m})$.

Figure 1(b) plots the regrets of algorithms `E-LinUCB` and `LinUCB-Tran` under sum feedback. To see the impact of segments on regrets clearly, here we show the regrets with respect to the number of segments $m$ and the length of each segment $\frac{H}{m}$ in the left and right subfigures, respectively. In the left subfigure, when $m$ increases, the regrets almost keep the same for small $m$ and slightly decrease for large $m$. To see the dependence on $m$ more clearly, we turn to the right subfigure: When the length of each segment $\frac{H}{m}$ increases, the regrets slightly increase in a logarithmic trend. This also matches our theoretical bounds, which do not depend on $m$ except for the $\log(\frac{H}{m})$ factor (Theorems 4 and 6).

## 7 CONCLUSION

In this work, we formulate a model named RL with segment feedback, which offers a general paradigm for feedback, bridging the gap between per-state-action feedback in classic RL and trajectory feedback. In the binary feedback setting, we deign efficient algorithms `SegBiTS` and `SegBiTS-Tran`, and provide regret upper and lower bounds which show a dependence on $\exp(\frac{Hr_{\max}}{2m})$. These results reveal that under binary feedback, increasing the number of segment $m$ greatly helps expedite learning. In the sum feedback setting, we develop near-optimal algorithms `E-LinUCB` and `LinUCB-Tran` in terms of $H$ and $m$, where the regret results do not depend on $m$ when ignoring logarithmic factors. These results exhibit that under sum feedback, increasing $m$ does not help accelerate learning much.

There are several interesting directions worth further investigation. One direction is to consider segments of unequal lengths and study how to divide segments to optimize learning. Another direction is to generalize the results to the function approximation setting.

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

# APPENDIX

## CONTENTS

# A  DETAILS OF THE EXPERIMENTAL SETUP

In this section, we detail the instances used in our experiments.

For the binary segment feedback setting, we consider an MDP as in Figure 1(a): There are $9$ states and $5$ actions. For any $a \in \mathcal{A}$, we have $r(s_0, a) = 0$, $r(s_i, a) = r_{\max}$ for any $i \in \{1, 3, 5, 7\}$ (called good states), and $r(s_i, a) = -r_{\max}$ for any $i \in \{2, 4, 6, 8\}$ (called bad states). There is an optimal action $a^*$ and four suboptimal actions $a^{\text{sub}}$ for all states. The agent starts from an initial state $s_0$. For any $0 \le i \le 6$, in state $s_i$, under the optimal action $a^*$, the agent transitions to the good state and bad state at the next horizon with probabilities $0.9$ and $0.1$, respectively; Under the suboptimal action $a^{\text{sub}}$, the agent transitions to the good state and bad state at the next horizon with probabilities $0.1$ and $0.9$, respectively. In $s_7$ or $s_8$, under the optimal action $a^*$, the agent transitions to $s_1$ and $s_2$ with probabilities $0.9$ and $0.1$, respectively; Under the suboptimal action $a^{\text{sub}}$, the agent transitions to $s_1$ and $s_2$ with probabilities $0.1$ and $0.9$, respectively.

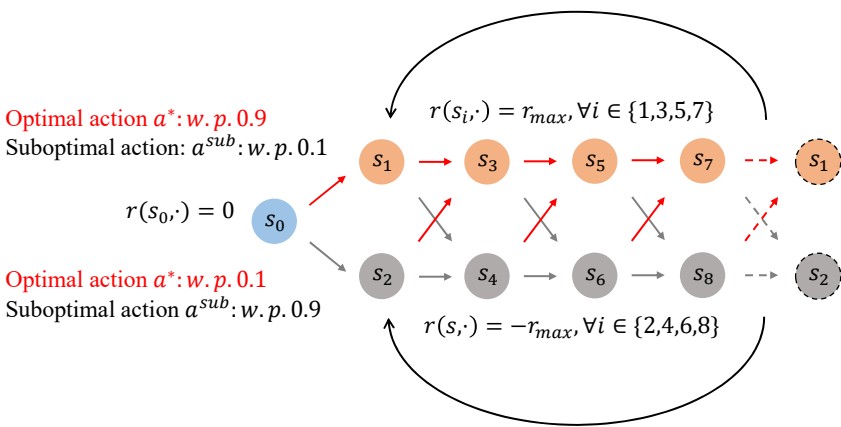

Figure 2: Instance used in the experiment for RL with binary segment feedback.

For the sum segment feedback setting, since algorithms `E-LinUCB` and `LinUCB-Tran` are computationally inefficient (which are mainly designed for revealing the optimal dependency on $H$ and $m$), we consider a smaller MDP as in Figure 1(b): There are $3$ states and $5$ actions. For any $a \in \mathcal{A}$, we have $r(s_0, a) = 0$, $r(s_1, a) = r_{\max}$ (called a good state), and $r(s_2, a) = -r_{\max}$ (called a bad state). There is an optimal action $a^*$ and four suboptimal actions $a^{\text{sub}}$ for all states. The agent starts from an initial state $s_0$. In any state $s \in \mathcal{S}$, under the optimal action $a^*$, the agent transitions to $s_1$ and $s_2$ with probabilities $0.9$ and $0.1$, respectively; Under the suboptimal action $a^{\text{sub}}$, the agent transitions to $s_1$ and $s_2$ with probabilities $0.1$ and $0.9$, respectively.

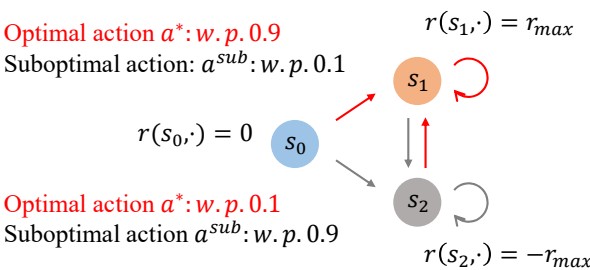

Figure 3: Instance used in the experiment for RL with sum segment feedback.

## B  ROUNDING PROCEDURE ROUND

Algorithm E-LinUCB calls a rounding procedure ROUND (Allen-Zhu et al., 2021) in the experimental design literature. Taking $X_1, \ldots, X_n \in \mathbb{S}_+^d$, distribution $w \in \triangle_{\{X_1,\ldots,X_n\}}$, rounding approximation error $\gamma > 0$ and the number of samples $T \geq \frac{d}{\gamma^2}$ as inputs, ROUND rounds sampling distribution $w$ into a discrete sampling sequence $(Y_1, \ldots, Y_T) \in \{X_1, \ldots, X_n\}^T$ that satisfies

$$\left\| \left( \sum_{t=1}^{T} Y_t \right)^{-1} \right\| \leq (1+\gamma) \left\| \left( T \sum_{i \in [n]} w(X_i) X_i \right)^{-1} \right\|.$$

In implementation, we can regard $xx^\top$ in (Allen-Zhu et al., 2021) as $\sum_{i=1}^{m} \mathbb{E}_{\tau_i \sim \pi}[\phi^{\tau_i}(\phi^{\tau_i})^\top]$, and regard sampling weight on $x$ as the sampling weight on $\pi$ in our work.

## C  PROOFS FOR RL WITH BINARY SEGMENT FEEDBACK

In this section, we present the proofs for RL with binary segment feedback.

### C.1  PROOF FOR THE REGRET UPPER BOUND WITH KNOWN TRANSITION

First, we prove the regret upper bound (Theorem 1) of algorithm SegBiTS for known transition.

For any $k > 0$ and $\theta \in \Theta$, define

$$Z_k := \sum_{k'=1}^{k} \sum_{i=1}^{m} \varepsilon_{k',i} \cdot \phi^{\tau_i^{k'}},$$

$$g_k(\theta) := \sum_{k'=1}^{k} \sum_{i=1}^{m} \mu((\phi^{\tau_i^{k'}})^\top \theta) \cdot \phi^{\tau_i^{k'}} + \lambda\theta, \tag{4}$$

$$\Lambda_k(\theta) := \sum_{k'=1}^{k} \sum_{i=1}^{m} \mu'((\phi^{\tau_i^{k'}})^\top \theta) \cdot \phi^{\tau_i^{k'}}(\phi^{\tau_i^{k'}})^\top + \lambda I. \tag{5}$$

**Lemma 1.** *For any $k > 0$ and $\theta \in \Theta$, we have*

$$\det(\Lambda_k(\theta)) \leq \left( \frac{H^2 \mu'_{\max} k}{|\mathcal{S}||\mathcal{A}|m} + \lambda \right)^{|\mathcal{S}||\mathcal{A}|}.$$

*Proof.* For any $k > 0$, we have

$$\det(\Lambda_k(\theta)) \leq \left( \frac{\operatorname{tr}(\Lambda_k(\theta))}{|\mathcal{S}||\mathcal{A}|} \right)^{|\mathcal{S}||\mathcal{A}|}$$

$$\leq \left( \frac{1}{|\mathcal{S}||\mathcal{A}|} \cdot \left( km \cdot \mu'_{\max} \cdot \left( \frac{H}{m} \right)^2 + \lambda|\mathcal{S}||\mathcal{A}| \right) \right)^{|\mathcal{S}||\mathcal{A}|}$$

$$= \left( \frac{H^2 \mu'_{\max} k}{|\mathcal{S}||\mathcal{A}|m} + \lambda \right)^{|\mathcal{S}||\mathcal{A}|}.$$

$\square$

For any $k > 0$, let $F_k$ denote the filtration that includes all events up to the end of episode $k$, and $\tilde{F}_k$ denote the filtration that includes all events before playing $\pi^k$ in episode $k$. Then, $\pi^k$ is $\tilde{F}_k$-measurable.

For any $k > 0$ and $i \in [m]$, let $\varepsilon_{k,i} := y_i^k - (\phi^{\tau_i^k})^\top \theta^*$ denote the noise of binary feedback, and $v_{k,i}^2 := \mathbb{E}[\varepsilon_{k,i}^2 | \tilde{F}_k] = (\phi^{\tau_i^k})^\top \theta^* \cdot (1 - (\phi^{\tau_i^k})^\top \theta^*) = \mu'((\phi^{\tau_i^k})^\top \theta^*)$ denote the variance of $\varepsilon_{k,i}$ conditioning on $\tilde{F}_k$.

Then, we have

$$\Lambda_k(\theta^*) := \sum_{k'=1}^{k} \sum_{i=1}^{m} \mu'((\phi_i^{\tau_i^{k'}})^\top \theta^*) \cdot \phi_i^{\tau_i^{k'}} (\phi_i^{\tau_i^{k'}})^\top + \lambda I$$

$$= \sum_{k'=1}^{k} \sum_{i=1}^{m} v_{k',i}^2 \cdot \phi_i^{\tau_i^{k'}} (\phi_i^{\tau_i^{k'}})^\top + \lambda I.$$

**Lemma 2** (Concentration of Noises under Binary Feedback). *With probability at least* $1 - \delta'$, *for any* $k > 0$,

$$\left\| \sum_{k'=1}^{k} \sum_{i=1}^{m} \varepsilon_{k',i} \cdot \phi_i^{\tau_i^{k'}} \right\|_{\Lambda_k^{-1}(\theta^*)} \leq \frac{\sqrt{\lambda}}{2} + \frac{|\mathcal{S}||\mathcal{A}|}{\sqrt{\lambda}} \log \left( \frac{4}{\delta'} \cdot \left( 1 + \frac{H^2 \mu'_{\max} k}{|\mathcal{S}||\mathcal{A}|m\lambda} \right) \right).$$

*Proof.* According to Theorem 1 in (Faury et al., 2020), we have that with probability at least $1 - \delta'$, for any $k > 0$,

$$\left\| \sum_{k'=1}^{k} \sum_{i=1}^{m} \varepsilon_{k',i} \cdot \phi_i^{\tau_i^{k'}} \right\|_{\Lambda_k^{-1}(\theta^*)} \leq \frac{\sqrt{\lambda}}{2} + \frac{2}{\sqrt{\lambda}} \log \left( \frac{\det(\Lambda_k(\theta^*))^{\frac{1}{2}} \cdot \lambda^{-\frac{|\mathcal{S}||\mathcal{A}|}{2}}}{\delta'} \right) + \frac{2}{\sqrt{\lambda}} |\mathcal{S}||\mathcal{A}| \log(2)$$

$$\overset{(a)}{\leq} \frac{\sqrt{\lambda}}{2} + \frac{2}{\sqrt{\lambda}} \log \left( \frac{1}{\delta'} \left( 1 + \frac{H^2 \mu'_{\max} k}{|\mathcal{S}||\mathcal{A}|m\lambda} \right)^{\frac{|\mathcal{S}||\mathcal{A}|}{2}} \right) + \frac{2}{\sqrt{\lambda}} |\mathcal{S}||\mathcal{A}| \log(2)$$

$$\leq \frac{\sqrt{\lambda}}{2} + \frac{|\mathcal{S}||\mathcal{A}|}{\sqrt{\lambda}} \log \left( \frac{1}{\delta'} \left( 1 + \frac{H^2 \mu'_{\max} k}{|\mathcal{S}||\mathcal{A}|m\lambda} \right) \right) + \frac{2}{\sqrt{\lambda}} |\mathcal{S}||\mathcal{A}| \log(2)$$

$$\leq \frac{\sqrt{\lambda}}{2} + \frac{|\mathcal{S}||\mathcal{A}|}{\sqrt{\lambda}} \log \left( \frac{4}{\delta'} \left( 1 + \frac{H^2 \mu'_{\max} k}{|\mathcal{S}||\mathcal{A}|m\lambda} \right) \right),$$

where (a) uses Lemma 1. $\qquad\square$

Define event

$$\mathcal{E} := \left\{ \left\| g_k(\hat{\theta}_k) - g_k(\theta^*) \right\|_{\Lambda_k^{-1}(\theta^*)} \leq \omega(k), \ \forall k > 0 \right\}.$$

**Lemma 3.** *It holds that*

$$\Pr[\mathcal{E}] \geq 1 - \delta'.$$

*Proof.* This proof is similar to that for Lemma 8 in (Faury et al., 2020).

Define

$$\mathcal{L}_k(\theta) := - \left( \sum_{k'=1}^{k} \sum_{i=1}^{m} \left( y_i^{k'} \cdot \log \left( \mu((\phi_i^{\tau_i^{k'}})^\top \theta) \right) + (1 - y_i^{k'}) \cdot \log \left( 1 - \mu((\phi_i^{\tau_i^{k'}})^\top \theta) \right) \right) - \frac{1}{2} \lambda \|\theta\|_2^2 \right).$$

Recall that $\hat{\theta}_k = \arg\min_\theta \mathcal{L}_k(\theta)$. Using the facts that $\nabla \mathcal{L}_k(\hat{\theta}_k) = 0$ and $\mu'(x) = \mu(x)(1 - \mu(x))$, we have

$$\underbrace{\sum_{k'=1}^{k} \sum_{i=1}^{m} \mu((\phi_i^{\tau_i^{k'}})^\top \hat{\theta}_k) \cdot \phi_i^{\tau_i^{k'}} + \lambda \hat{\theta}_k}_{g_k(\hat{\theta}_k)} = \sum_{k'=1}^{k} \sum_{i=1}^{m} y_i^{k'} \cdot \phi_i^{\tau_i^{k'}}.$$

Hence, we have

$$g_k(\hat{\theta}_k) - g_k(\theta^*) = \sum_{k'=1}^{k} \sum_{i=1}^{m} y_i^{k'} \cdot \phi_i^{\tau_i^{k'}} - \left( \sum_{k'=1}^{k} \sum_{i=1}^{m} \mu((\phi_i^{\tau_i^{k'}})^\top \theta^*) \cdot \phi_i^{\tau_i^{k'}} + \lambda \theta^* \right)$$

$$= \sum_{k'=1}^{k} \sum_{i=1}^{m} \varepsilon_{k',i} \cdot \phi^{\tau_i^{k'}} - \lambda \theta^*. \tag{6}$$

Then, using Lemma 2, we have that with probability at least $1 - \delta'$, for any $k > 0$,

$$
\left\| g_k(\hat{\theta}_k) - g_k(\theta^*) \right\|_{\Lambda_k^{-1}(\theta^*)} \leq \left\| \sum_{k'=1}^{k} \sum_{i=1}^{m} \varepsilon_{k',i} \cdot \phi^{\tau_i^{k'}} \right\|_{\Lambda_k^{-1}(\theta^*)} + r_{\max} \sqrt{\lambda |\mathcal{S}||\mathcal{A}|}
$$

$$
\leq \frac{\sqrt{\lambda}}{2} + \frac{|\mathcal{S}||\mathcal{A}|}{\sqrt{\lambda}} \log \left( \frac{4}{\delta'} \cdot \left( 1 + \frac{H^2 \mu'_{\max} k}{|\mathcal{S}||\mathcal{A}| m \lambda} \right) \right) + r_{\max} \sqrt{\lambda |\mathcal{S}||\mathcal{A}|}
$$

$$
= \omega(k).
$$

$\square$

For any $\phi \in \mathbb{R}^{|\mathcal{S}||\mathcal{A}|}$ and $\theta_1, \theta_2 \in \Theta$, define

$$
b(\phi, \theta_1, \theta_2) := \int_{z=0}^{1} \mu'((1-z) \cdot \phi^\top \theta_1 + z \cdot \phi^\top \theta_2) dz.
$$

For any $k > 0$ and $\theta_1, \theta_2 \in \Theta$, define

$$
\Gamma_k(\theta_1, \theta_2) := \sum_{k'=1}^{k} \sum_{i=1}^{m} b(\phi, \theta_1, \theta_2) \cdot \phi^{\tau_i^{k'}} (\phi^{\tau_i^{k'}})^\top + \lambda I.
$$

In the definitions of $b(\phi, \theta_1, \theta_2)$ and $\Gamma_k(\theta_1, \theta_2)$, $\theta_1$ and $\theta_2$ have the same roles and can be interchanged.

Recall that

$$
\alpha := \exp(\frac{H r_{\max}}{m}) + \exp(-\frac{H r_{\max}}{m}) + 2.
$$

Then, we have

$$
\sup_{\tau^{\text{seg}}, \theta} \frac{1}{\mu'((\phi^{\tau^{\text{seg}}})^\top \theta)} \leq \alpha,
$$

where $\tau^{\text{seg}}$ denotes the visitation indicator of any possible trajectory segment.

**Lemma 4.** *For any $k \geq 1$ and $\theta \in \Theta$, we have*

$$
\Sigma_k \preceq \alpha \Lambda_k(\theta).
$$

*Proof.* We have

$$
\frac{1}{\alpha} = \inf_{\tau^{\text{seg}}, \theta} \mu'((\phi^{\tau^{\text{seg}}})^\top \theta).
$$

Then, it holds that

$$
\Sigma_k = \sum_{k'=1}^{k} \sum_{i=1}^{m} \phi^{\tau_i^{k'}} (\phi^{\tau_i^{k'}})^\top + \alpha \lambda I
$$

$$
= \alpha \left( \sum_{k'=1}^{k} \sum_{i=1}^{m} \frac{1}{\alpha} \cdot \phi^{\tau_i^{k'}} (\phi^{\tau_i^{k'}})^\top + \lambda I \right)
$$

$$
\preceq \alpha \left( \sum_{k'=1}^{k} \sum_{i=1}^{m} \mu'((\phi^{\tau_i^{k'}})^\top \theta) \cdot \phi^{\tau_i^{k'}} (\phi^{\tau_i^{k'}})^\top + \lambda I \right)
$$

$$
= \alpha \Lambda_k(\theta).
$$

$\square$

**Lemma 5.** *For any $\phi \in \mathbb{R}^{|\mathcal{S}||\mathcal{A}|}$ and $\theta_1, \theta_2 \in \Theta$, we have*

$$\mu(\phi^\top \theta_1) - \mu(\phi^\top \theta_2) = b(\phi, \theta_2, \theta_1) \cdot \phi^\top (\theta_1 - \theta_2).$$

*In addition, for any $k > 0$ and $\theta_1, \theta_2 \in \Theta$, we have*

$$\|\theta_1 - \theta_2\|_{\Gamma_k(\theta_2, \theta_1)} = \|g_k(\theta_1) - g_k(\theta_2)\|_{\Gamma_k^{-1}(\theta_2, \theta_1)}.$$

*Proof.* The first statement follows from the mean-value theorem.

Then, using the first statement, we have that for any $k > 0$,

$$g_k(\theta_1) - g_k(\theta_2) = \sum_{k'=1}^{k} \sum_{i=1}^{m} \left( \mu((\phi^{\tau_i^{k'}})^\top \theta_1) - \mu((\phi^{\tau_i^{k'}})^\top \theta_2) \right) \cdot \phi^{\tau_i^{k'}} + \lambda (\theta_1 - \theta_2)$$

$$= \sum_{k'=1}^{k} \sum_{i=1}^{m} b(\phi^{\tau_i^{k'}}, \theta_2, \theta_1) \cdot \phi^{\tau_i^{k'}} (\phi^{\tau_i^{k'}})^\top (\theta_1 - \theta_2) + \lambda (\theta_1 - \theta_2)$$

$$= \Gamma_k(\theta_2, \theta_1) \cdot (\theta_1 - \theta_2),$$

and thus

$$\|\theta_1 - \theta_2\|_{\Gamma_k(\theta_2, \theta_1)} = \sqrt{(\theta_1 - \theta_2)^\top \cdot \Gamma_k(\theta_2, \theta_1) \cdot (\theta_1 - \theta_2)}$$

$$= \sqrt{(\theta_1 - \theta_2)^\top \cdot \Gamma_k(\theta_2, \theta_1) \cdot \Gamma_k^{-1}(\theta_2, \theta_1) \cdot \Gamma_k(\theta_2, \theta_1) \cdot (\theta_1 - \theta_2)}$$

$$= \|g_k(\theta_1) - g_k(\theta_2)\|_{\Gamma_k^{-1}(\theta_2, \theta_1)},$$

which gives the second statement. $\qquad\square$

Recall that for any $k > 0$, $Z_k := \sum_{k'=1}^{k} \sum_{i=1}^{m} \varepsilon_{k', i} \cdot \phi^{\tau_i^{k'}}$.

**Lemma 6.** *For any $k > 0$, we have*

$$\Gamma_k(\theta^*, \hat{\theta}_k) \succeq \left( 1 + \frac{H r_{\max} \sqrt{|\mathcal{S}||\mathcal{A}|}}{m} + \frac{H}{m \sqrt{\lambda}} \|Z_k\|_{\Gamma_k^{-1}(\theta^*, \hat{\theta}_k)} \right)^{-1} \Lambda_k(\theta^*),$$

$$\|Z_k\|_{\Gamma_k^{-1}(\theta^*, \hat{\theta}_k)} \leq \sqrt{1 + \frac{H r_{\max} \sqrt{|\mathcal{S}||\mathcal{A}|}}{m}} \|Z_k\|_{\Lambda_k^{-1}(\theta^*)} + \frac{H}{m \sqrt{\lambda}} \|Z_k\|_{\Lambda_k^{-1}(\theta^*)}^2.$$

*Furthermore, assuming that event $\mathcal{E}$ holds, we have*

$$\|Z_k\|_{\Gamma_k^{-1}(\theta^*, \hat{\theta}_k)} \leq \sqrt{1 + \frac{H r_{\max} \sqrt{|\mathcal{S}||\mathcal{A}|}}{m}} \cdot \omega(k) + \frac{H}{m \sqrt{\lambda}} \cdot \omega(k)^2.$$

*Proof.* This proof follows the analysis of Proposition 6 and Corollary 5 in (Russac et al., 2021).

From Eq. (6), we have that for any $k > 0$,

$$g_k(\hat{\theta}_k) - g_k(\theta^*) = Z_k - \lambda \theta^*.$$

Using Lemma 34, we have that for any $\phi \in \mathbb{R}^{|\mathcal{S}||\mathcal{A}|}$ such that $\|\phi\|_2 \leq L_\phi$,

$$b(\phi, \theta^*, \hat{\theta}_k) \geq \left( 1 + \left| \phi^\top (\theta^* - \hat{\theta}_k) \right| \right)^{-1} \mu'(\phi^\top \theta^*)$$

$$= \left( 1 + \left| \phi^\top \Gamma_k^{-1}(\theta^*, \hat{\theta}_k) \cdot (g_k(\theta^*) - g_k(\hat{\theta}_k)) \right| \right)^{-1} \mu'(\phi^\top \theta^*)$$

$$\geq \left( 1 + \|\phi\|_{\Gamma_k^{-1}(\theta^*, \hat{\theta}_k)} \left\| g_k(\theta^*) - g_k(\hat{\theta}_k) \right\|_{\Gamma_k^{-1}(\theta^*, \hat{\theta}_k)} \right)^{-1} \mu'(\phi^\top \theta^*)$$

$$\geq \left( 1 + \frac{L_\phi}{\sqrt{\lambda}} \left\| g_k(\theta^*) - g_k(\hat{\theta}_k) \right\|_{\Gamma_k^{-1}(\theta^*, \hat{\theta}_k)} \right)^{-1} \mu'(\phi^\top \theta^*)$$

$$= \left(1 + \frac{L_\phi}{\sqrt{\lambda}} \|Z_k - \lambda\theta^*\|_{\Gamma_k^{-1}(\theta^*, \hat{\theta}_k)}\right)^{-1} \mu'(\phi^\top \theta^*)$$

$$\geq \left(1 + L_\phi r_{\max}\sqrt{|\mathcal{S}||\mathcal{A}|} + \frac{L_\phi}{\sqrt{\lambda}} \|Z_k\|_{\Gamma_k^{-1}(\theta^*, \hat{\theta}_k)}\right)^{-1} \mu'(\phi^\top \theta^*).$$

Using the above equation with $\phi = \phi^{\tau_i^{k'}}$ and $L_\phi = \frac{H}{m}$, we have

$$\Gamma_k(\theta^*, \hat{\theta}_k) := \sum_{k'=1}^{k} \sum_{i=1}^{m} b(\phi^{\tau_i^{k'}}, \theta^*, \hat{\theta}_k) \cdot \phi^{\tau_i^{k'}}(\phi^{\tau_i^{k'}})^\top + \lambda I$$

$$\succeq \sum_{k'=1}^{k} \sum_{i=1}^{m} \left(1 + \frac{Hr_{\max}\sqrt{|\mathcal{S}||\mathcal{A}|}}{m} + \frac{H}{m\sqrt{\lambda}} \|Z_k\|_{\Gamma_k^{-1}(\theta^*, \hat{\theta}_k)}\right)^{-1} \mu'(\phi^\top \theta^*) \cdot \phi^{\tau_i^{k'}}(\phi^{\tau_i^{k'}})^\top + \lambda I$$

$$= \left(1 + \frac{Hr_{\max}\sqrt{|\mathcal{S}||\mathcal{A}|}}{m} + \frac{H}{m\sqrt{\lambda}} \|Z_k\|_{\Gamma_k^{-1}(\theta^*, \hat{\theta}_k)}\right)^{-1} \Lambda_k(\theta^*).$$

This implies

$$\|Z_k\|_{\Gamma_k^{-1}(\theta^*, \hat{\theta}_k)}^2 \leq \left(1 + \frac{Hr_{\max}\sqrt{|\mathcal{S}||\mathcal{A}|}}{m} + \frac{H}{m\sqrt{\lambda}} \|Z_k\|_{\Gamma_k^{-1}(\theta^*, \hat{\theta}_k)}\right) \|Z_k\|_{\Lambda_k^{-1}(\theta^*)}^2,$$

which is equivalent to

$$\|Z_k\|_{\Gamma_k^{-1}(\theta^*, \hat{\theta}_k)}^2 - \frac{H}{m\sqrt{\lambda}} \|Z_k\|_{\Lambda_k^{-1}(\theta^*)}^2 \|Z_k\|_{\Gamma_k^{-1}(\theta^*, \hat{\theta}_k)} - \left(1 + \frac{Hr_{\max}\sqrt{|\mathcal{S}||\mathcal{A}|}}{m}\right) \|Z_k\|_{\Lambda_k^{-1}(\theta^*)}^2 \leq 0.$$

By analysis of quadratic functions, we have

$$\|Z_k\|_{\Gamma_k^{-1}(\theta^*, \hat{\theta}_k)} \leq \sqrt{1 + \frac{Hr_{\max}\sqrt{|\mathcal{S}||\mathcal{A}|}}{m}} \|Z_k\|_{\Lambda_k^{-1}(\theta^*)} + \frac{H}{m\sqrt{\lambda}} \|Z_k\|_{\Lambda_k^{-1}(\theta^*)}^2$$

$$\leq \sqrt{1 + \frac{Hr_{\max}\sqrt{|\mathcal{S}||\mathcal{A}|}}{m}} \cdot \omega(k) + \frac{H}{m\sqrt{\lambda}} \cdot \omega(k)^2.$$

$$\square$$

**Lemma 7** (Concentration of $\phi^\top \hat{\theta}_k$ under Binary Feedback). *Assume that event $\mathcal{E}$ holds. Then, for any $k > 0$ and $\phi \in \mathbb{R}^{|\mathcal{S}||\mathcal{A}|}$,*

$$|\phi^\top \theta^* - \phi^\top \hat{\theta}_k| \leq \sqrt{\alpha} \cdot \nu(k) \|\phi\|_{\Sigma_k^{-1}}.$$

*Proof.* We have

$$|\phi^\top \theta^* - \phi^\top \hat{\theta}_k|$$

$$= \|\phi\|_{\Gamma_k^{-1}(\theta^*, \hat{\theta}_k)} \left\|\theta^* - \hat{\theta}_k\right\|_{\Gamma_k(\theta^*, \hat{\theta}_k)}$$

$$\overset{(a)}{\leq} \sqrt{1 + \frac{Hr_{\max}\sqrt{|\mathcal{S}||\mathcal{A}|}}{m} + \frac{H}{m\sqrt{\lambda}} \|Z_k\|_{\Gamma_k^{-1}(\theta^*, \hat{\theta}_k)}} \|\phi\|_{\Lambda_k^{-1}(\theta^*)} \left\|g_k(\theta^*) - g_k(\hat{\theta}_k)\right\|_{\Gamma_k^{-1}(\theta^*, \hat{\theta}_k)}$$

$$= \sqrt{1 + \frac{Hr_{\max}\sqrt{|\mathcal{S}||\mathcal{A}|}}{m} + \frac{H}{m\sqrt{\lambda}} \|Z_k\|_{\Gamma_k^{-1}(\theta^*, \hat{\theta}_k)}} \|\phi\|_{\Lambda_k^{-1}(\theta^*)} \|Z_k - \lambda\theta^*\|_{\Gamma_k^{-1}(\theta^*, \hat{\theta}_k)}$$

$$\leq \sqrt{1 + \frac{Hr_{\max}\sqrt{|\mathcal{S}||\mathcal{A}|}}{m} + \frac{H}{m\sqrt{\lambda}} \|Z_k\|_{\Gamma_k^{-1}(\theta^*, \hat{\theta}_k)}} \|\phi\|_{\Lambda_k^{-1}(\theta^*)} \left(\|Z_k\|_{\Gamma_k^{-1}(\theta^*, \hat{\theta}_k)} + r_{\max}\sqrt{\lambda|\mathcal{S}||\mathcal{A}|}\right)$$

$$= \frac{m\sqrt{\lambda}}{H} \sqrt{1 + \frac{Hr_{\max}\sqrt{|\mathcal{S}||\mathcal{A}|}}{m} + \frac{H}{m\sqrt{\lambda}} \|Z_k\|_{\Gamma_k^{-1}(\theta^*, \hat{\theta}_k)} } \, \|\phi\|_{\Lambda_k^{-1}(\theta^*)} \cdot$$

$$\left( \frac{H}{m\sqrt{\lambda}} \|Z_k\|_{\Gamma_k^{-1}(\theta^*, \hat{\theta}_k)} + \frac{Hr_{\max}\sqrt{|\mathcal{S}||\mathcal{A}|}}{m} \right)$$

$$\leq \frac{m\sqrt{\lambda}}{H} \left( 1 + \frac{Hr_{\max}\sqrt{|\mathcal{S}||\mathcal{A}|}}{m} + \frac{H}{m\sqrt{\lambda}} \|Z_k\|_{\Gamma_k^{-1}(\theta^*, \hat{\theta}_k)} \right)^{\frac{3}{2}} \|\phi\|_{\Lambda_k^{-1}(\theta^*)}$$

$$\overset{(b)}{\leq} \frac{m\sqrt{\alpha\lambda}}{H} \left( 1 + \frac{Hr_{\max}\sqrt{|\mathcal{S}||\mathcal{A}|}}{m} + \frac{H}{m\sqrt{\lambda}} \left( \sqrt{1 + \frac{Hr_{\max}\sqrt{|\mathcal{S}||\mathcal{A}|}}{m}\omega(k) + \frac{H}{m\sqrt{\lambda}}\omega(k)^2} \right) \right)^{\frac{3}{2}} \|\phi\|_{\Sigma_k^{-1}}$$

$$= \sqrt{\alpha} \cdot \nu(k) \|\phi\|_{\Sigma_k^{-1}} \, .$$

where inequality (a) is due to Lemmas 5 and 6, and inequality (b) follows from Lemmas 4 and 6. $\qquad\square$

**Lemma 8** (Gaussian Anti-Concentration). *Assume that event $\mathcal{E}$ holds. Then, for any $k > 0$ and $F_{k-1}$-measurable random variable $X \in \mathbb{R}^{|\mathcal{S}||\mathcal{A}|}$, we have*

$$\Pr\left[ X^\top \tilde{\theta}_k > X^\top \theta^* \mid F_{k-1} \right] \geq \frac{1}{2\sqrt{2\pi e}}.$$

*Proof.* This proof is originated from the analysis of Lemma 11 in (Efroni et al., 2021).

Using Lemma 7, we have that for any $k > 0$,

$$|X^\top \theta^* - X^\top \hat{\theta}_{k-1}| \leq \sqrt{\alpha} \cdot \nu(k-1) \|X\|_{\Sigma_{k-1}^{-1}} \, .$$

It holds that

$$\Pr\left[ X^\top \tilde{\theta}_k > X^\top \theta^* \mid F_{k-1} \right]$$

$$= \Pr\left[ \frac{X^\top \tilde{\theta}_k - X^\top \hat{\theta}_{k-1}}{\sqrt{\alpha} \cdot \nu(k-1)\|X\|_{\Sigma_{k-1}^{-1}}} > \frac{X^\top \theta^* - X^\top \hat{\theta}_{k-1}}{\sqrt{\alpha} \cdot \nu(k-1)\|X\|_{\Sigma_{k-1}^{-1}}} \mid F_{k-1} \right].$$

Here given $F_{k-1}$, $X^\top \tilde{\theta}_k - X^\top \hat{\theta}_{k-1} = X^\top \xi_k$ is a Gaussian random variable with mean $0$ and standard deviation $\sqrt{\alpha} \cdot \nu(k-1)\|X\|_{\Sigma_{k-1}^{-1}}$.

Since when event $\mathcal{E}$ holds,

$$\frac{X^\top \theta^* - X^\top \hat{\theta}_{k-1}}{\sqrt{\alpha} \cdot \nu(k-1)\|X\|_{\Sigma_{k-1}^{-1}}} \leq \frac{\sqrt{\alpha} \cdot \nu(k-1)\|X\|_{\Sigma_{k-1}^{-1}}}{\sqrt{\alpha} \cdot \nu(k-1)\|X\|_{\Sigma_{k-1}^{-1}}} = 1,$$

we have

$$\Pr\left[ X^\top \tilde{\theta}_k > X^\top \theta^* \mid F_{k-1} \right] \geq \Pr\left[ \frac{X^\top \tilde{\theta}_k - X^\top \hat{\theta}_{k-1}}{\sqrt{\alpha} \cdot \nu(k-1)\|X\|_{\Sigma_{k-1}^{-1}}} > 1 \mid F_{k-1} \right]$$

$$= \Pr\left[ \frac{X^\top \xi_k}{\sqrt{\alpha} \cdot \nu(k-1)\|X\|_{\Sigma_{k-1}^{-1}}} > 1 \mid F_{k-1} \right]$$

$$\overset{(a)}{\geq} \frac{1}{2\sqrt{2\pi e}},$$

where inequality (a) comes from that if $Z \sim \mathcal{F}_{\mathrm{UTran}}^{\mathrm{B}}(0,1)$, $\Pr[Z > z] \geq \frac{1}{\sqrt{2\pi}} \cdot \frac{z}{1+z^2} e^{-\frac{z^2}{2}}$ (Borjesson & Sundberg, 1979). $\qquad\square$

**Lemma 9.** *Let $\xi_k, \xi_k' \in \mathbb{R}^{|\mathcal{S}||\mathcal{A}|}$ be i.i.d. random variables given $F_{k-1}$. Let $\tilde{p}$ be a $F_{k-1}$-measurable transition model, and $x_{k-1} \in \mathbb{R}^{|\mathcal{S}||\mathcal{A}|}$ be a $F_{k-1}$-measurable random variable. For any policy $\pi$,*

*denote the visitation indicator under policy $\pi$ on MDP $\tilde{p}$ by $\tilde{\phi}^\pi$. Let $\tilde{\pi}^k := \mathrm{argmax}_\pi (\tilde{\phi}^\pi)^\top (x_{k-1} + \xi_k)$. Then, we have*

$$\mathbb{E}\left[\left((\tilde{\phi}^{\tilde{\pi}^k})^\top (x_{k-1} + \xi_k) - \mathbb{E}\left[(\tilde{\phi}^{\tilde{\pi}^k})^\top (x_{k-1} + \xi_k) \mid F_{k-1}\right]\right)^+ \,\Big|\, F_{k-1}\right]$$

$$\leq \mathbb{E}\left[|(\tilde{\phi}^{\tilde{\pi}^k})^\top \xi_k| + |(\tilde{\phi}^{\tilde{\pi}^k})^\top \xi'_k| \,\Big|\, F_{k-1}\right].$$

*Proof.* This proof is originated from Lemma 12 in (Efroni et al., 2021).

First, using the definition of $\tilde{\pi}^k$ and the fact that $\xi_k$ and $\xi'_k$ follow the same distribution, we have

$$\mathbb{E}\left[(\tilde{\phi}^{\tilde{\pi}^k})^\top (x_{k-1} + \xi_k) \mid F_{k-1}\right] = \mathbb{E}\left[\max_\pi (\tilde{\phi}^\pi)^\top (x_{k-1} + \xi'_k) \mid F_{k-1}\right]. \tag{7}$$

Then, since given $F_{k-1}$, $\xi_k$ and $\xi'_k$ are independent, we have

$$\mathbb{E}\left[\max_\pi (\tilde{\phi}^\pi)^\top (x_{k-1} + \xi'_k) \mid F_{k-1}\right] = \mathbb{E}\left[\max_\pi (\tilde{\phi}^\pi)^\top (x_{k-1} + \xi'_k) \mid F_{k-1}, \xi_k, \tilde{\pi}_k\right]$$

$$\geq \mathbb{E}\left[(\phi^{\tilde{\pi}_k})^\top (x_{k-1} + \xi'_k) \mid F_{k-1}, \xi_k, \tilde{\pi}_k\right]. \tag{8}$$

Hence, combining Eqs. (7) and (8), we have

$$\mathbb{E}\left[\left((\phi^{\tilde{\pi}_k})^\top (x_{k-1} + \xi_k) - \mathbb{E}\left[(\phi^{\tilde{\pi}_k})^\top (x_{k-1} + \xi_k) \mid F_{k-1}\right]\right)^+ \,\Big|\, F_{k-1}\right]$$

$$\leq \mathbb{E}\left[\left((\phi^{\tilde{\pi}_k})^\top (x_{k-1} + \xi_k) - \mathbb{E}\left[(\phi^{\tilde{\pi}_k})^\top (x_{k-1} + \xi'_k) \mid F_{k-1}, \xi_k, \tilde{\pi}_k\right]\right)^+ \,\Big|\, F_{k-1}\right]$$

$$= \mathbb{E}\left[\left(\mathbb{E}\left[(\phi^{\tilde{\pi}_k})^\top (x_{k-1} + \xi_k) - (\phi^{\tilde{\pi}_k})^\top (x_{k-1} + \xi'_k) \mid F_{k-1}, \xi_k, \tilde{\pi}_k\right]\right)^+ \,\Big|\, F_{k-1}\right]$$

$$= \mathbb{E}\left[\left(\mathbb{E}\left[(\phi^{\tilde{\pi}_k})^\top \xi_k - (\phi^{\tilde{\pi}_k})^\top \xi'_k \mid F_{k-1}, \xi_k, \tilde{\pi}_k\right]\right)^+ \,\Big|\, F_{k-1}\right]$$

$$\leq \mathbb{E}\left[\left|\mathbb{E}\left[(\phi^{\tilde{\pi}_k})^\top \xi_k - (\phi^{\tilde{\pi}_k})^\top \xi'_k \mid F_{k-1}, \xi_k, \tilde{\pi}_k\right]\right| \,\Big|\, F_{k-1}\right]$$

$$\leq \mathbb{E}\left[\mathbb{E}\left[|(\phi^{\tilde{\pi}_k})^\top \xi_k - (\phi^{\tilde{\pi}_k})^\top \xi'_k| \mid F_{k-1}, \xi_k, \tilde{\pi}_k\right] \,\Big|\, F_{k-1}\right]$$

$$= \mathbb{E}\left[|(\phi^{\tilde{\pi}_k})^\top \xi_k - (\phi^{\tilde{\pi}_k})^\top \xi'_k| \,\Big|\, F_{k-1}\right]$$

$$\leq \mathbb{E}\left[|(\phi^{\tilde{\pi}_k})^\top \xi_k| \,\Big|\, F_{k-1}\right] + \mathbb{E}\left[|(\phi^{\tilde{\pi}_k})^\top \xi'_k| \,\Big|\, F_{k-1}\right].$$

$\square$

For any $k > 0$ and $\delta_k \in (0, 1)$, define event

$$\mathcal{M}_k(\delta_k) := \left\{\forall \phi \in \mathbb{R}^{|\mathcal{S}||\mathcal{A}|} : |\phi^\top \xi_k| \leq \sqrt{\alpha} \cdot \nu(k-1)\left(\sqrt{|\mathcal{S}||\mathcal{A}|} + 2\sqrt{\log\left(\frac{1}{\delta_k}\right)}\right) \|\phi\|_{\Sigma_{k-1}^{-1}}\right\}.$$

**Lemma 10.** *For any $k > 0$ and $\delta_k \in (0, 1)$, we have*

$$\Pr\left[\mathcal{M}_k(\delta_k) \mid F_{k-1}\right] \geq 1 - \delta_k.$$

*In addition, for a random variable $X \in \mathbb{R}^{|\mathcal{S}||\mathcal{A}|}$ such that $\|X\|_{\Sigma_{k-1}^{-1}} \leq L_X$, we have*

$$\mathbb{E}\left[|X^\top \xi_k| \mid F_{k-1}\right] \leq \sqrt{\alpha} \cdot \nu(k-1)\left(\sqrt{|\mathcal{S}||\mathcal{A}|} + 2\sqrt{\log\left(\frac{1}{\delta_k}\right)}\right) \mathbb{E}\left[\|X\|_{\Sigma_{k-1}^{-1}} \mid F_{k-1}\right]$$

$$+ \sqrt{\alpha} \cdot \nu(k-1) \cdot L_X \sqrt{|\mathcal{S}||\mathcal{A}|\delta_k}.$$

*Proof.* This proof is similar to the analysis of Lemma 13 in (Efroni et al., 2021).

First, we prove the first statement.

For any $\phi \in \mathbb{R}^{|\mathcal{S}||\mathcal{A}|}$, we have

$$
\begin{aligned}
|\phi^\top \xi_k| &= |\phi^\top \Sigma_{k-1}^{-\frac{1}{2}} \Sigma_{k-1}^{\frac{1}{2}} \xi_k| \\
&\leq \left\| \Sigma_{k-1}^{-\frac{1}{2}} \phi \right\|_2 \left\| \Sigma_{k-1}^{\frac{1}{2}} \xi_k \right\|_2 \\
&= \sqrt{\alpha} \cdot \nu(k-1) \|\phi\|_{\Sigma_{k-1}^{-1}} \left\| \frac{1}{\sqrt{\alpha} \cdot \nu(k-1)} \Sigma_{k-1}^{\frac{1}{2}} \xi_k \right\|_2 .
\end{aligned}
\tag{9}
$$

Since given $F_{k-1}$, $\frac{1}{\sqrt{\alpha}\cdot\nu(k-1)} \Sigma_{k-1}^{\frac{1}{2}} \xi_k \in \mathbb{R}^{|\mathcal{S}||\mathcal{A}|}$ is a vector with each entry being a standard Gaussian random variable, we have that $\|\frac{1}{\sqrt{\alpha}\cdot\nu(k-1)} \Sigma_{k-1}^{\frac{1}{2}} \xi_k\|_2$ is chi-distributed with parameter $|\mathcal{S}||\mathcal{A}|$.

Then, using Lemma 1 in (Laurent & Massart, 2000), we have that with probability at least $1 - \delta_k$,

$$
\begin{aligned}
\left\| \frac{1}{\sqrt{\alpha} \cdot \nu(k-1)} \Sigma_{k-1}^{\frac{1}{2}} \xi_k \right\|_2 &\leq \sqrt{ |\mathcal{S}||\mathcal{A}| + 2\sqrt{|\mathcal{S}||\mathcal{A}| \log\left(\frac{1}{\delta_k}\right)} + 2\log\left(\frac{1}{\delta_k}\right) } \\
&= \sqrt{ \left( \sqrt{|\mathcal{S}||\mathcal{A}|} + \sqrt{\log\left(\frac{1}{\delta_k}\right)} \right)^2 + \log\left(\frac{1}{\delta_k}\right) } \\
&\leq \sqrt{|\mathcal{S}||\mathcal{A}|} + 2\sqrt{\log\left(\frac{1}{\delta_k}\right)} .
\end{aligned}
$$

Next, we prove the second statement.

For a random variable $X \in \mathbb{R}^{|\mathcal{S}||\mathcal{A}|}$, we have

$$
\begin{aligned}
\mathbb{E}\left[ |X^\top \xi_k| \,|F_{k-1} \right] &= \Pr\left[\mathcal{M}_k(\delta_k)\right] \cdot \mathbb{E}\left[ |X^\top \xi_k| \,|F_{k-1}, \mathcal{M}_k(\delta_k) \right] \\
&\quad + \Pr\left[\bar{\mathcal{M}}_k(\delta_k)\right] \cdot \mathbb{E}\left[ |X^\top \xi_k| \,|F_{k-1}, \bar{\mathcal{M}}_k(\delta_k) \right] \\
&\leq \sqrt{\alpha} \cdot \nu(k-1) \left( \sqrt{|\mathcal{S}||\mathcal{A}|} + 2\sqrt{\log\left(\frac{1}{\delta_k}\right)} \right) \mathbb{E}\left[ \|X\|_{\Sigma_{k-1}^{-1}} \,|F_{k-1} \right] \\
&\quad + \sqrt{ \Pr\left[\bar{\mathcal{M}}_k(\delta_k)\right] \cdot \mathbb{E}\left[ |X^\top \xi_k|^2 \,|F_{k-1}, \bar{\mathcal{M}}_k(\delta_k) \right] } \\
&\overset{(a)}{\leq} \sqrt{\alpha} \cdot \nu(k-1) \left( \sqrt{|\mathcal{S}||\mathcal{A}|} + 2\sqrt{\log\left(\frac{1}{\delta_k}\right)} \right) \mathbb{E}\left[ \|X\|_{\Sigma_{k-1}^{-1}} \,|F_{k-1} \right] \\
&\quad + \sqrt{\alpha} \cdot \nu(k-1) \sqrt{ \delta_k \mathbb{E}\left[ \|X\|_{\Sigma_{k-1}^{-1}}^2 \cdot \left\| \frac{1}{\sqrt{\alpha} \cdot \nu(k-1)} \Sigma_{k-1}^{\frac{1}{2}} \xi_k \right\|_2^2 \,|F_{k-1} \right] } \\
&\leq \sqrt{\alpha} \cdot \nu(k-1) \left( \sqrt{|\mathcal{S}||\mathcal{A}|} + 2\sqrt{\log\left(\frac{1}{\delta_k}\right)} \right) \mathbb{E}\left[ \|X\|_{\Sigma_{k-1}^{-1}} \,|F_{k-1} \right] \\
&\quad + \sqrt{\alpha} \cdot \nu(k-1) \sqrt{ \delta_k L_X^2 \mathbb{E}\left[ \left\| \frac{1}{\sqrt{\alpha} \cdot \nu(k-1)} \Sigma_{k-1}^{\frac{1}{2}} \xi_k \right\|_2^2 \,|F_{k-1} \right] } \\
&\overset{(b)}{\leq} \sqrt{\alpha} \cdot \nu(k-1) \left( \sqrt{|\mathcal{S}||\mathcal{A}|} + 2\sqrt{\log\left(\frac{1}{\delta_k}\right)} \right) \mathbb{E}\left[ \|X\|_{\Sigma_{k-1}^{-1}} \,|F_{k-1} \right] \\
&\quad + \sqrt{\alpha} \cdot \nu(k-1) \cdot L_X \sqrt{|\mathcal{S}||\mathcal{A}|\delta_k} .
\end{aligned}
$$

Here inequality (a) follows from the Cauchy-Schwarz inequality. Inequality (b) is due to the fact that given $F_{k-1}$, $\|\frac{1}{\sqrt{\alpha}\cdot\nu(k-1)} \Sigma_{k-1}^{\frac{1}{2}} \xi_k\|_2$ is chi-distributed with parameter $|\mathcal{S}||\mathcal{A}|$, and then $\mathbb{E}[\|\frac{1}{\sqrt{\alpha}\cdot\nu(k-1)} \Sigma_{k-1}^{\frac{1}{2}} \xi_k\|_2^2 \,|F_{k-1}] = |\mathcal{S}||\mathcal{A}|$. $\qquad\square$

Define event

$$
\begin{aligned}
\mathcal{F}^{\mathrm{B}}_{\mathrm{KTran}} := \Bigg\{ &\left| \sum_{k'=1}^{k} \left( \mathbb{E}_{\tau \sim \pi^{k'}} \left[ \|\phi^\tau\|_{(\Sigma_{k'-1})^{-1}} | F_{k'-1} \right] - \|\phi^\tau\|_{(\Sigma_{k'-1})^{-1}} \right) \right| \leq 4H \sqrt{\frac{k}{\alpha\lambda} \log\left(\frac{4k}{\delta'}\right)}, \\
&\left| \sum_{k'=1}^{k} \left( \mathbb{E}\left[ (\phi^{\pi^{k'}})^\top \theta^* | F_{k'-1} \right] - (\phi^{\pi^{k'}})^\top \theta^* \right) \right| \leq 4H r_{\max} \sqrt{k \log\left(\frac{4k}{\delta'}\right)}, \ \forall k > 0 \Bigg\}.
\end{aligned}
\tag{10}
$$

**Lemma 11.** *It holds that*

$$
\Pr\left[ \mathcal{F}^{\mathrm{B}}_{\mathrm{KTran}} \right] \geq 1 - 2\delta'.
$$

*Proof.* We prove the first inequality as follows.

For any $k' \geq 1$, we have that $\|\phi^\tau\|_{(\Sigma_{k'-1})^{-1}} \leq \frac{H}{\sqrt{\alpha\lambda}}$, and then $\left| \mathbb{E}_{\tau \sim \pi^{k'}}[\|\phi^\tau\|_{(\Sigma_{k'-1})^{-1}} | F_{k'-1}] - \|\phi^\tau\|_{(\Sigma_{k'-1})^{-1}} \right| \leq \frac{2H}{\sqrt{\alpha\lambda}}$.

Using the Azuma-Hoeffding inequality, we have that for any fixed $k > 0$, with probability at least $1 - \frac{\delta'}{2k^2}$,

$$
\left| \sum_{k'=1}^{k} \left( \mathbb{E}_{\tau \sim \pi^{k'}} \left[ \|\phi^\tau\|_{(\Sigma_{k'-1})^{-1}} | F_{k'-1} \right] - \|\phi^\tau\|_{(\Sigma_{k'-1})^{-1}} \right) \right| \leq \sqrt{2 \cdot \frac{4H^2}{\alpha\lambda} \cdot k \log\left(\frac{4k^2}{\delta'}\right)}.
$$

Since $\sum_{k=1}^{\infty} \frac{\delta'}{2k^2} \leq \delta'$, by a union bound over $k$, we have that with probability at least $\delta'$, for any $k \geq 1$,

$$
\begin{aligned}
\left| \sum_{k'=1}^{k} \left( \mathbb{E}_{\tau \sim \pi^{k'}} \left[ \|\phi^\tau\|_{(\Sigma_{k'-1})^{-1}} | F_{k'-1} \right] - \|\phi^\tau\|_{(\Sigma_{k'-1})^{-1}} \right) \right| &\leq \sqrt{2 \cdot \frac{4H^2}{\alpha\lambda} \cdot k \log\left(\frac{4k^2}{\delta'}\right)} \\
&\leq 4H \sqrt{\frac{k}{\alpha\lambda} \log\left(\frac{4k}{\delta'}\right)}.
\end{aligned}
$$

The second inequality can be obtained by a similar argument and the fact that $|(\phi^{\pi^k})^\top \theta^*| \leq H r_{\max}$ for any $k > 0$. $\square$

**Lemma 12.** *For any $K \geq 1$, we have*

$$
\sum_{k=1}^{K} \sum_{i=1}^{m} \left\| \phi^{\tau_i^k} \right\|_{(\Sigma_{k-1})^{-1}} \leq \sqrt{2Km|\mathcal{S}||\mathcal{A}| \cdot \max\left\{ \frac{H^2}{m\alpha\lambda}, 1 \right\} \cdot \log\left( 1 + \frac{KH^2}{\alpha\lambda|\mathcal{S}||\mathcal{A}|m} \right)}.
$$

*Proof.* We have

$$
\begin{aligned}
\sum_{k=1}^{K} \sum_{i=1}^{m} \left\| \phi^{\tau_i^k} \right\|_{(\Sigma_{k-1})^{-1}} &\leq \sqrt{Km \sum_{k=1}^{K} \sum_{i=1}^{m} \left\| \phi^{\tau_i^k} \right\|_{(\Sigma_{k-1})^{-1}}^2} \\
&\overset{(a)}{\leq} \sqrt{2Km \cdot \max\left\{ \frac{H^2}{m\alpha\lambda}, 1 \right\} \cdot \sum_{k=1}^{K} \log\left( 1 + \sum_{i=1}^{m} \left\| \phi^{\tau_i^k} \right\|_{(\Sigma_{k-1})^{-1}}^2 \right)} \\
&= \sqrt{2Km \cdot \max\left\{ \frac{H^2}{m\alpha\lambda}, 1 \right\} \cdot \log\left( \frac{\det(\Sigma_K)}{\det(\alpha\lambda I)} \right)} \\
&\leq \sqrt{2Km|\mathcal{S}||\mathcal{A}| \cdot \max\left\{ \frac{H^2}{m\alpha\lambda}, 1 \right\} \cdot \log\left( 1 + \frac{KH^2}{\alpha\lambda|\mathcal{S}||\mathcal{A}|m} \right)}, \tag{11}
\end{aligned}
$$

where inequality (a) is due to that for any $x \in [0, c]$ with constant $c \geq 0$, it holds that $x \leq 2\max\{c, 1\} \cdot \log(1 + x)$. $\square$

*Proof of Theorem 1.* Letting $\delta' = \frac{\delta}{3}$, we have $\Pr[\mathcal{E} \cap \mathcal{F}_{\mathrm{KTran}}^{\mathrm{B}}] \leq 1 - \delta$. Then, to prove this theorem, it suffices to prove the regret bound when event $\mathcal{E} \cap \mathcal{F}_{\mathrm{KTran}}^{\mathrm{B}}$ holds.

Assume that event $\mathcal{E} \cap \mathcal{F}_{\mathrm{KTran}}^{\mathrm{B}}$ holds. Then, we have

$$
\mathcal{R}(K) = \sum_{k=1}^{K} \left( (\phi^{\pi^*})^\top \theta^* - (\phi^{\pi^k})^\top \theta^* \right)
$$

$$
= \sum_{k=1}^{K} \left( \mathbb{E}\left[ (\phi^{\pi^*})^\top \theta^* - (\phi^{\pi^k})^\top \theta^* | F_{k-1} \right] + \mathbb{E}\left[ (\phi^{\pi^k})^\top \theta^* | F_{k-1} \right] - (\phi^{\pi^k})^\top \theta^* \right)
$$

$$
\leq \sum_{k=1}^{K} \left( \mathbb{E}\left[ (\phi^{\pi^*})^\top \theta^* - (\phi^{\pi^k})^\top \theta^* | F_{k-1} \right] \right) + 4 H r_{\max} \sqrt{K \log\left( \frac{4K}{\delta'} \right)}. \tag{12}
$$

For the first term, we have

$$
\sum_{k=1}^{K} \mathbb{E}\left[ (\phi^{\pi^*})^\top \theta^* - (\phi^{\pi^k})^\top \theta^* | F_{k-1} \right]
$$

$$
= \sum_{k=1}^{K} \left( \mathbb{E}\left[ (\phi^{\pi^*})^\top \theta^* - (\phi^{\pi^k})^\top \tilde{\theta}_k | F_{k-1} \right] + \mathbb{E}\left[ (\phi^{\pi^k})^\top \tilde{\theta}_k - (\phi^{\pi^k})^\top \theta^* | F_{k-1} \right] \right). \tag{13}
$$

In the following, we prove

$$
\mathbb{E}\left[ (\phi^{\pi^*})^\top \theta^* - (\phi^{\pi^k})^\top \tilde{\theta}_k | F_{k-1} \right] \leq 2\sqrt{2\pi e} \cdot \mathbb{E}\left[ \left( (\phi^{\pi^k})^\top \tilde{\theta}_k - \mathbb{E}\left[ (\phi^{\pi^k})^\top \tilde{\theta}_k | F_{k-1} \right] \right)^+ | F_{k-1} \right]. \tag{14}
$$

If $\mathbb{E}[(\phi^{\pi^*})^\top \theta^* - (\phi^{\pi^k})^\top \tilde{\theta}_k | F_{k-1}] < 0$, then Eq. (14) trivially holds.

Otherwise, letting $z := \mathbb{E}[(\phi^{\pi^*})^\top \theta^* - (\phi^{\pi^k})^\top \tilde{\theta}_k | F_{k-1}]$, we have

$$
\mathbb{E}\left[ \left( (\phi^{\pi^k})^\top \tilde{\theta}_k - \mathbb{E}\left[ (\phi^{\pi^k})^\top \tilde{\theta}_k | F_{k-1} \right] \right)^+ | F_{k-1} \right]
$$

$$
\geq z \Pr\left[ (\phi^{\pi^k})^\top \tilde{\theta}_k - \mathbb{E}\left[ (\phi^{\pi^k})^\top \tilde{\theta}_k | F_{k-1} \right] \geq z | F_{k-1} \right]
$$

$$
\geq \left( \mathbb{E}\left[ (\phi^{\pi^*})^\top \theta^* - (\phi^{\pi^k})^\top \tilde{\theta}_k | F_{k-1} \right] \right) \cdot \Pr\left[ (\phi^{\pi^k})^\top \tilde{\theta}_k \geq (\phi^{\pi^*})^\top \theta^* | F_{k-1} \right]
$$

$$
\overset{(a)}{\geq} \left( \mathbb{E}\left[ (\phi^{\pi^*})^\top \theta^* - (\phi^{\pi^k})^\top \tilde{\theta}_k | F_{k-1} \right] \right) \cdot \Pr\left[ (\phi^{\pi^*})^\top \tilde{\theta}_k \geq (\phi^{\pi^*})^\top \theta^* | F_{k-1} \right]
$$

$$
\overset{(b)}{\geq} \left( \mathbb{E}\left[ (\phi^{\pi^*})^\top \theta^* - (\phi^{\pi^k})^\top \tilde{\theta}_k | F_{k-1} \right] \right) \cdot \frac{1}{2\sqrt{2\pi e}},
$$

where inequality (a) uses the definition of $\pi^k$, and inequality (b) follows from Lemma 8. Thus, we complete the proof of Eq. (14).

Let $\xi_k' \in \mathbb{R}^{|\mathcal{S}||\mathcal{A}|}$ be a random variable that is i.i.d. with $\xi$ given $F_{k-1}$. Then, using Lemma 9 with $p' = p$, $x_{k-1} = \hat{\theta}_{k-1}$ and $\tilde{\pi}^k = \pi^k$, we have

$$
\mathbb{E}\left[ (\phi^{\pi^*})^\top \theta^* - (\phi^{\pi^k})^\top \tilde{\theta}_k | F_{k-1} \right] \leq 2\sqrt{2\pi e} \cdot \mathbb{E}\left[ \left( (\phi^{\pi^k})^\top \tilde{\theta}_k - \mathbb{E}\left[ (\phi^{\pi^k})^\top \tilde{\theta}_k | F_{k-1} \right] \right)^+ | F_{k-1} \right]
$$

$$
\leq 2\sqrt{2\pi e} \cdot \mathbb{E}\left[ |\phi(\pi^k)^\top \xi_k| + |\phi(\pi^k)^\top \xi_k'| \, | F_{k-1} \right].
$$

Plugging the above inequality into Eq. (13) and using Lemma 10 with $\delta_k = \frac{1}{k^4}$ and $L_X = \frac{H}{\sqrt{\alpha \lambda}}$, we have

$$
\sum_{k=1}^{K} \mathbb{E}\left[ (\phi^{\pi^*})^\top \theta^* - (\phi^{\pi^k})^\top \theta^* | F_{k-1} \right]
$$

$$
= \sum_{k=1}^{K} \left( 2\sqrt{2\pi e}\, \mathbb{E}\left[ |(\phi^{\pi^k})^\top \xi_k| + |(\phi^{\pi^k})^\top \xi_k'| \,|F_{k-1}\right] + \mathbb{E}\left[ (\phi^{\pi^k})^\top \left(\hat{\theta}_{k-1} + \xi_k\right) - (\phi^{\pi^k})^\top \theta^* |F_{k-1}\right] \right)
$$

$$
= \sum_{k=1}^{K} \left( \left( 2\sqrt{2\pi e} + 1 \right) \cdot \mathbb{E}\left[ |(\phi^{\pi^k})^\top \xi_k| \,|F_{k-1}\right] + 2\sqrt{2\pi e} \cdot \mathbb{E}\left[ |(\phi^{\pi^k})^\top \xi_k'| \,|F_{k-1}\right] \right.
$$

$$
\left. + \mathbb{E}\left[ (\phi^{\pi^k})^\top \hat{\theta}_{k-1} - (\phi^{\pi^k})^\top \theta^* |F_{k-1}\right] \right)
$$

$$
\overset{(a)}{\le} \sum_{k=1}^{K} \left( \left( 4\sqrt{2\pi e} + 2 \right) \sqrt{\alpha} \cdot \nu(k-1)\left( \sqrt{|\mathcal{S}||\mathcal{A}|} + 4\sqrt{\log(k)} \right) \cdot \mathbb{E}\left[ \left\| \phi^{\pi^k} \right\|_{\Sigma_{k-1}^{-1}} |F_{k-1} \right] \right.
$$

$$
\left. + \left( 4\sqrt{2\pi e} + 1 \right) \sqrt{\alpha} \cdot \nu(k-1)\frac{\sqrt{|\mathcal{S}||\mathcal{A}|}}{k^2} \cdot \frac{H}{\sqrt{\alpha\lambda}} \right), \tag{15}
$$

where inequality (a) uses Lemmas 7 and 10.

Here according to the definition of event $\mathcal{F}_{\text{KTran}}^{\text{B}}$ and Lemma 12, we have

$$
\sum_{k=1}^{K} \mathbb{E}\left[ \left\| \phi^{\pi^k} \right\|_{\Sigma_{k-1}^{-1}} |F_{k-1} \right] = \sum_{k=1}^{K} \left( \mathbb{E}\left[ \left\| \phi^{\pi^k} \right\|_{\Sigma_{k-1}^{-1}} |F_{k-1} \right] - \left\| \phi^{\pi^k} \right\|_{\Sigma_{k-1}^{-1}} \right) + \sum_{k=1}^{K} \left\| \phi^{\pi^k} \right\|_{\Sigma_{k-1}^{-1}}
$$

$$
\le 4H\sqrt{\frac{K}{\alpha\lambda} \log\left(\frac{4K}{\delta'}\right)}
$$

$$
+ \sqrt{2Km|\mathcal{S}||\mathcal{A}| \cdot \max\left\{ \frac{H^2}{m\alpha\lambda}, 1 \right\} \cdot \log\left( 1 + \frac{KH^2}{\alpha\lambda|\mathcal{S}||\mathcal{A}|m} \right)}. \tag{16}
$$

Therefore, plugging the above two equations into Eq. (12), we have

$$
\mathcal{R}(K) \le \left( 4\sqrt{2\pi e} + 2 \right) \sqrt{\alpha} \cdot \nu(K)\left( \sqrt{|\mathcal{S}||\mathcal{A}|} + 4\sqrt{\log(K)} \right) \cdot
$$

$$
\left( 4H\sqrt{\frac{K}{\alpha\lambda} \log\left(\frac{4K}{\delta'}\right)} + \sqrt{2Km|\mathcal{S}||\mathcal{A}| \max\left\{ \frac{H^2}{m\alpha\lambda}, 1 \right\} \log\left( 1 + \frac{KH^2}{\alpha\lambda|\mathcal{S}||\mathcal{A}|m} \right)} \right)
$$

$$
+ 2\left( 4\sqrt{2\pi e} + 1 \right) H \cdot \nu(K)\sqrt{\frac{|\mathcal{S}||\mathcal{A}|}{\lambda}} + 4Hr_{\max}\sqrt{K\log\left(\frac{4K}{\delta'}\right)}
$$

$$
\overset{(a)}{=} \tilde{O}\left( \exp(\frac{Hr_{\max}}{2m}) \cdot \nu(K)\sqrt{|\mathcal{S}||\mathcal{A}|}\left( \sqrt{Km|\mathcal{S}||\mathcal{A}| \cdot \max\left\{ \frac{H^2}{m\alpha\lambda}, 1 \right\}} + H\sqrt{\frac{K}{\alpha\lambda}} \right) \right),
$$

where in equality (a), the last two terms are absorbed into $\tilde{O}(\cdot)$. $\qquad\square$

## C.2 Proof for the Regret Lower Bound with Known Transition

In the following, we prove the regret lower bound (Theorem 2) for RL with binary segment feedback and known transition.

*Proof of Theorem 2.* We construct a random instance $\mathcal{I}$ as follows. As shown in Figure 4, there are $n$ bandit states $s_1, \ldots, s_n$ (i.e., there is an optimal action and multiple suboptimal actions), a good absorbing state $s_{n+1}$ and a bad absorbing state $s_{n+2}$. The agent starts from $s_1, \ldots, s_n$ with equal probability $\frac{1}{n}$. For any $i \in [n]$, in state $s_i$, one action $a_J$ is uniformly chosen from $\mathcal{A}$ as the optimal action. In state $s_i$, under the optimal action $a_J$, the agent transitions to $s_{n+1}$ deterministically, and $r(s_i, a_J) = r_{\max}$; Under any suboptimal action $a \in \mathcal{A} \setminus \{s_J\}$, the agent transitions to $s_{n+2}$ deterministically, and $r(s_i, a) = (1 - \varepsilon)r_{\max}$, where $\varepsilon \in (0, \frac{1}{2})$ is a parameter specified later. For all actions $a \in \mathcal{A}$, $r(s_{n+1}, a) = r_{\max}$ and $r(s_{n+2}, a) = (1 - \varepsilon)r_{\max}$.

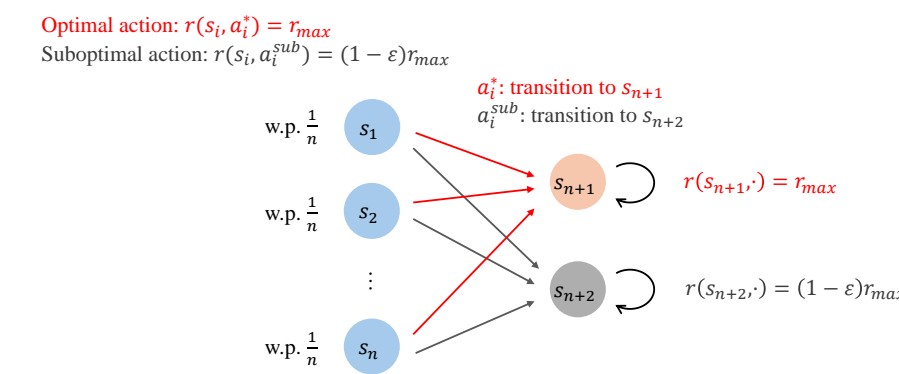

Figure 4: Instance for the lower bound under binary segment feedback and known transition.

In this proof, we will also use an alternative uniform instance $\mathcal{I}_{\text{unif}}$. The only difference between $\mathcal{I}_{\text{unif}}$ and $\mathcal{I}$ is that for any $i \in [n]$, in state $s_i$, under all actions $a \in \mathcal{A}$, the agent transitions to $s_{n+2}$ deterministically, and $r(s_i, a) = (1 - \varepsilon)r_{\max}$.

Fix an algorithm $\mathbb{A}$. Let $\mathbb{E}_{\text{unif}}[\cdot]$ denote the expectation with respect to $\mathcal{I}_{\text{unif}}$. Let $\mathbb{E}_*[\cdot]$ denote the expectation with respect to $\mathcal{I}$. For any $i \in [n]$ and $j \in [|\mathcal{A}|]$, let $\mathbb{E}_{i,j}[\cdot]$ denote the expectation with respect to the case where $a_j$ is the optimal action in state $s_i$, and $N_{i,j}$ denote the number of episodes where algorithm $\mathbb{A}$ chooses $a_j$ in state $s_i$, i.e., $N_{i,j} = \sum_{k=1}^{K} \mathbb{1}\{\pi_1^k(s_i) = a_j\}$.

The KL divergence of binary observations if taking $a_J$ in $s_i$ in each episode between $\mathcal{I}_{\text{unif}}$ and $\mathcal{I}$ is

$$\sum_{i=1}^{m} \text{KL}\left(\text{Ber}\left(\mu\left((1-\varepsilon)r_{\max} \cdot \frac{H}{m}\right)\right) \middle\| \text{Ber}\left(\mu\left(r_{\max} \cdot \frac{H}{m}\right)\right)\right)$$

$$\overset{(a)}{\le} m \cdot \frac{\left(\mu\left((1-\varepsilon)r_{\max} \cdot \frac{H}{m}\right) - \mu\left(r_{\max} \cdot \frac{H}{m}\right)\right)^2}{\mu'\left(r_{\max} \cdot \frac{H}{m}\right)}$$

$$\overset{(b)}{\le} m \cdot \frac{\mu'\left((1-\varepsilon)\frac{Hr_{\max}}{m}\right)^2 \left(\varepsilon \cdot \frac{Hr_{\max}}{m}\right)^2}{\mu'\left(\frac{Hr_{\max}}{m}\right)},$$

where inequality (a) uses the fact that $\text{KL}(\text{Ber}(p)\|\text{Ber}(q)) \le \frac{(p-q)^2}{q(1-q)}$, and inequality (b) is due to that $\mu'(x)$ is monotonically decreasing when $x > 0$.

In addition, the agent has probability only $\frac{1}{n}$ to arrive at (observe) state $s_i$.

Thus, using Lemma A.1 in (Auer et al., 2002), we have that for any $i \in [n]$, in state $s_i$,

$$\mathbb{E}_{i,j}[N_{i,j}] \le \mathbb{E}_{\text{unif}}[N_{i,j}] + \frac{K}{2}\sqrt{\frac{1}{n} \cdot \mathbb{E}_{\text{unif}}[N_{i,j}] \cdot m \cdot \frac{\mu'\left((1-\varepsilon)\frac{Hr_{\max}}{m}\right)^2 \left(\varepsilon \cdot \frac{Hr_{\max}}{m}\right)^2}{\mu'\left(\frac{Hr_{\max}}{m}\right)}}$$

$$= \mathbb{E}_{\text{unif}}[N_{i,j}] + \frac{K}{2} \cdot \varepsilon \cdot \frac{Hr_{\max}}{m}\sqrt{\frac{m}{n} \cdot \mathbb{E}_{\text{unif}}[N_{i,j}] \cdot \frac{\mu'\left((1-\varepsilon)\frac{Hr_{\max}}{m}\right)^2}{\mu'\left(\frac{Hr_{\max}}{m}\right)}}.$$

Summing over $j \in [|\mathcal{A}|]$, using the Cauchy-Schwarz inequality and the fact that $\sum_{j=1}^{|\mathcal{A}|} \mathbb{E}_{\text{unif}}[N_{i,j}] = K$, we have

$$\sum_{j=1}^{|\mathcal{A}|} \mathbb{E}_{i,j}[N_{i,j}] \le K + \frac{KHr_{\max}\varepsilon}{2}\sqrt{\frac{|\mathcal{A}|K}{mn} \cdot \frac{\mu'\left((1-\varepsilon)\frac{Hr_{\max}}{m}\right)^2}{\mu'\left(\frac{Hr_{\max}}{m}\right)}}$$

$$\le K + \frac{KHr_{\max}\varepsilon}{2}\sqrt{\frac{|\mathcal{A}|K}{mn} \cdot \frac{\mu'\left((1-c_0)\frac{Hr_{\max}}{m}\right)^2}{\mu'\left(\frac{Hr_{\max}}{m}\right)}},$$

where $c_0 \in (0, \frac{1}{2})$ is a constant which satisfies $c_0 \geq \varepsilon$. We will specify how to make $c_0 \geq \varepsilon$ to satisfy this condition later.

Then, we have

$$\mathcal{R}(K) = \sum_{k=1}^{K} \mathbb{E}_* \left[ V^* - V^{\pi^k} \right]$$

$$= r_{\max} H K - \frac{1}{n} \sum_{i=1}^{n} \left( (1-\varepsilon) r_{\max} H K + \varepsilon r_{\max} H \cdot \frac{1}{|\mathcal{A}|} \sum_{j=1}^{|\mathcal{A}|} \mathbb{E}_{i,j}[N_{i,j}] \right)$$

$$\geq \varepsilon r_{\max} H \left( K - \frac{K}{|\mathcal{A}|} - \frac{K H r_{\max} \varepsilon}{2} \sqrt{\frac{K}{|\mathcal{A}| m n} \cdot \frac{\mu'\left((1-c_0)\frac{H r_{\max}}{m}\right)^2}{\mu'\left(\frac{H r_{\max}}{m}\right)}} \right).$$

Let

$$\varepsilon = \frac{1}{2 H r_{\max}} \sqrt{\frac{|\mathcal{A}| m n}{K} \cdot \frac{\mu'\left(\frac{H r_{\max}}{m}\right)}{\mu'\left((1-c_0)\frac{H r_{\max}}{m}\right)^2}}.$$

Then, the constant $c_0$ should satisfy

$$\varepsilon = \frac{1}{2 H r_{\max}} \sqrt{\frac{|\mathcal{A}| m n}{K} \cdot \frac{\mu'\left(\frac{H r_{\max}}{m}\right)}{\mu'\left((1-c_0)\frac{H r_{\max}}{m}\right)^2}} \leq c_0.$$

Since

$$\frac{\mu'\left(\frac{H r_{\max}}{m}\right)}{\mu'\left((1-c_0)\frac{H r_{\max}}{m}\right)^2} = \frac{\left(\exp\left((1-c_0)\frac{H r_{\max}}{m}\right) + \exp\left(-(1-c_0)\frac{H r_{\max}}{m}\right) + 2\right)^2}{\exp\left(\frac{H r_{\max}}{m}\right) + \exp\left(-\frac{H r_{\max}}{m}\right) + 2}$$

$$\leq \frac{\left(4 \exp\left((1-c_0)\frac{H r_{\max}}{m}\right)\right)^2}{\exp\left(\frac{H r_{\max}}{m}\right)}$$

$$= 16 \exp\left(\left(1 - 2c_0\right)\frac{H r_{\max}}{m}\right),$$

it suffices to let $c_0$ satisfy

$$\frac{1}{2 H r_{\max}} \sqrt{\frac{|\mathcal{A}| m n}{K} \cdot 16 \exp\left(\left(1 - 2c_0\right)\frac{H r_{\max}}{m}\right)} \leq c_0,$$

which is equivalent to $K \geq \frac{4|\mathcal{A}| m n}{H^2 r_{\max}^2 c_0^2} \exp((1 - 2c_0)\frac{H r_{\max}}{m})$.

It suffices to let

$$K \geq \frac{4|\mathcal{A}| m n}{H^2 r_{\max}^2 c_0^2} \exp\left(\frac{H r_{\max}}{m}\right),$$

and then $c_0$ can be any constant in $(0, \frac{1}{2})$.

Let $|\mathcal{S}| \geq 3$, $|\mathcal{A}| \geq 2$, $c_0 \in (0, \frac{1}{2})$ and $K \geq \frac{4|\mathcal{A}| m n}{H^2 r_{\max}^2 c_0^2} \exp(\frac{H r_{\max}}{m})$. Since

$$\frac{\mu'\left(\frac{H r_{\max}}{m}\right)}{\mu'\left((1-c_0)\frac{H r_{\max}}{m}\right)^2} = \frac{\left(\exp\left((1-c_0)\frac{H r_{\max}}{m}\right) + \exp\left(-(1-c_0)\frac{H r_{\max}}{m}\right) + 2\right)^2}{\exp\left(\frac{H r_{\max}}{m}\right) + \exp\left(-\frac{H r_{\max}}{m}\right) + 2}$$

$$\geq \frac{\left(\exp\left((1-c_0)\frac{H r_{\max}}{m}\right)\right)^2}{4 \exp\left(\frac{H r_{\max}}{m}\right)}$$

---

**Algorithm 3:** `SegBiTS-Tran`

**Input:** $\delta, \delta' := \frac{\delta}{8}, \lambda$.

**1 for** $k = 1, \ldots, K$ **do**

**2** $\quad$ $\hat{\theta}_{k-1} \leftarrow \text{argmin}_\theta -(\sum_{k'=1}^{k-1} \sum_{i=1}^m (y_i^{k'} \cdot \log(\mu((\phi^{\tau_i^{k'}})^\top \theta)) + (1 - y_i^{k'}) \cdot \log(1 -$
$\quad\quad \mu((\phi^{\tau_i^{k'}})^\top \theta))) - \frac{1}{2}\lambda\|\theta\|_2^2)$;

**3** $\quad$ $\Sigma_{k-1} \leftarrow \sum_{k'=1}^{k-1} \sum_{i=1}^m \phi^{\tau_i^{k'}} (\phi^{\tau_i^{k'}})^\top + \alpha\lambda I$;

**4** $\quad$ Draw a noise $\xi_k \sim \mathcal{N}(0, \alpha \cdot \nu(k-1)^2 \cdot \Sigma_{k-1}^{-1})$, where $\nu(k-1)$ is defined in Eq. (1);

**5** $\quad$ $b_{k-1}^{pv}(s,a) \leftarrow \min\{2Hr_{\max}\sqrt{\frac{\log\left(\frac{KH|\mathcal{S}||\mathcal{A}|}{\delta'}\right)}{n_{k-1}(s,a)}}, \ Hr_{\max}\}$ for any $(s,a) \in \mathcal{S} \times \mathcal{A}$;

**6** $\quad$ $\tilde{\theta}_k^b \leftarrow \hat{\theta}_{k-1} + \xi_k + b_{k-1}^{pv}$;

**7** $\quad$ $\pi^k \leftarrow \text{argmax}_\pi (\hat{\phi}_{k-1}^\pi)^\top \tilde{\theta}_k^b$, where $\hat{\phi}_{k-1}^\pi$ is defined in Eq. (17);

**8** $\quad$ Play episode $k$ with policy $\pi^k$. Observe $\tau^k$ and binary segment feedback $\{y_i^k\}_{i=1}^m$;

---

$$= \frac{1}{4} \exp\left(\left(1 - 2c_0\right)\frac{Hr_{\max}}{m}\right),$$

we have

$$\mathcal{R}(K) \geq \frac{1}{2Hr_{\max}} \sqrt{\frac{|\mathcal{A}|mn}{K} \cdot \frac{\mu'\left(\frac{Hr_{\max}}{m}\right)}{\mu'\left((1-c_0)\frac{Hr_{\max}}{m}\right)^2}} \cdot r_{\max}H\left(K - \frac{K}{|\mathcal{A}|} - \frac{K}{4}\right)$$

$$= \Omega\left(\sqrt{\exp\left((1-2c_0)\frac{Hr_{\max}}{m}\right)|\mathcal{S}||\mathcal{A}|mK}\right)$$

$$= \Omega\left(\exp\left(\left(\frac{1}{2} - c_0\right)\frac{Hr_{\max}}{m}\right)\sqrt{|\mathcal{S}||\mathcal{A}|mK}\right).$$

$\square$

### C.3 PSEUDO-CODE AND DETAILED DESCRIPTION OF ALGORITHM `SegBiTS-Tran`

Algorithm 3 illustrates the procedure of `SegBiTS-Tran`. In episode $k$, similar to `SegBiTS`, `SegBiTS-Tran` first uses MLE with past binary segment observations to obtain a reward estimate $\hat{\theta}_{k-1}$, and calculates the covariance matrix of past observations $\Sigma_{k-1}$ (Lines 2-3). After that, `SegBiTS-Tran` samples a Gaussian noise $\xi_k$ using $\Sigma_{k-1}$ (Line 3).

For any $k > 0$ and $(s,a) \in \mathcal{S} \times \mathcal{A}$, let $\hat{p}_k(\cdot|s,a)$ denote the empirical estimate of $p(\cdot|s,a)$, and $n_k(s,a)$ denote the number of times $(s,a)$ was visited at the end of episode $k$. Then, `SegBiTS-Tran` constructs a transition bonus $b_{k-1}^{pv}(s,a)$, which represents the uncertainty on transition estimation. Incorporating the MLE estimate $\hat{\theta}_{k-1}$, noise $\xi_k$ and transition bonus $b_{k-1}^{pv}(s,a)$, `SegBiTS-Tran` constitutes a posterior estimate of the reward parameter $\tilde{\theta}_k$ (Line 6).

For any policy $\pi$, $k > 0$ and $(s,a) \in \mathcal{S} \times \mathcal{A}$, we define

$$\hat{\phi}_k^\pi(s,a) := \mathbb{E}_{\hat{p}_k}\left[\sum_{h=1}^H \mathbb{1}\{s_h = s, a_h = a\}|\pi\right], \tag{17}$$

which denotes the expected number of times $(s,a)$ is visited in an episode under policy $\pi$ on the empirical MDP $\hat{p}_k$. In addition, let $\hat{\phi}_k^\pi := [\hat{\phi}_k^\pi(s,a)]_{(s,a)\in\mathcal{S}\times\mathcal{A}} \in \mathbb{R}^{|\mathcal{S}||\mathcal{A}|}$.

Then, `SegBiTS-Tran` finds the optimal policy via $\text{argmax}_\pi (\hat{\phi}_{k-1}^\pi)^\top \tilde{\theta}_k^b$, which can be efficiently solved by any MDP planning algorithm with transition $\hat{p}_{k-1}$ and reward $\tilde{\theta}_k^b$ (Line 7). With the computed optimal policy $\pi^k$, `SegBiTS-Tran` plays episode $k$, and observes a trajectory and binary feedback on each segment (Line 8).

### C.4 PROOF FOR THE REGRET UPPER BOUND WITH UNKNOWN TRANSITION

In the following, we prove the regret upper bound (Theorem 3) of algorithm `SegBiTS-Tran` for unknown transition.

Define event

$$
\mathcal{G}_{\text{Hoeff}} := \left\{ \left| \hat{p}_{k-1}(\cdot|s,a)^\top V_{h+1}^* - p(\cdot|s,a)^\top V_{h+1}^* \right| \leq \left( 2Hr_{\max} \sqrt{ \frac{ \log\left( \frac{KH|\mathcal{S}||\mathcal{A}|}{\delta'} \right) }{ n_{k-1}(s,a) } } \wedge Hr_{\max} \right), \right.
$$

$$
\left. \forall (s,a) \in \mathcal{S} \times \mathcal{A},\ \forall k > 0 \right\}.
$$

**Lemma 13.** *It holds that*

$$
\Pr\left[ \mathcal{G}_{\text{Hoeff}} \right] \geq 1 - 2\delta'.
$$

*Proof.* This lemma follows from the Hoeffding inequality and a union bound over $n_{k-1}(s,a) \in [KH]$ and $(s,a) \in \mathcal{S} \times \mathcal{A}$. $\qquad\square$

**Lemma 14** (Optimism of Thompson Sampling with Unknown Transition). *Assume that event $\mathcal{E}$ and $\mathcal{G}_{\text{Hoeff}}$ holds. Then, for any $k > 0$, we have*

$$
\Pr\left[ \hat{\phi}_{k-1}(\pi^k)^\top \tilde{\theta}_k^b > (\phi^{\pi^*})^\top \theta^* \mid F_{k-1} \right] \geq \frac{1}{2\sqrt{2\pi e}}.
$$

*Proof.* This proof follows the analysis of Lemma 17 in (Efroni et al., 2021).

Using the value difference lemma (see Lemma 35), we have

$$
\hat{\phi}_{k-1}(\pi^*)^\top \tilde{\theta}_k^b - (\phi^{\pi^*})^\top \theta^*
$$

$$
= \mathbb{E}_{\hat{p}_{k-1}, \pi^*} \left[ \sum_{h=1}^{H} \left( \tilde{\theta}_k^b(s_h, a_h) - \theta^*(s_h, a_h) + (\hat{p}_{k-1}(\cdot|s_h, a_h) - p(\cdot|s_h, a_h))^\top V_{h+1}^* \right) \right]
$$

$$
= \mathbb{E}_{\hat{p}_{k-1}, \pi^*} \left[ \sum_{h=1}^{H} \left( \tilde{\theta}_k(s_h, a_h) - \theta^*(s_h, a_h) + b_{k-1}^{pv}(s_h, a_h) + (\hat{p}_{k-1}(\cdot|s_h, a_h) - p(\cdot|s_h, a_h))^\top V_{h+1}^* \right) \right]
$$

$$
\overset{(a)}{\geq} \mathbb{E}_{\hat{p}_{k-1}, \pi^*} \left[ \sum_{h=1}^{H} \left( \tilde{\theta}_k(s_h, a_h) - \theta^*(s_h, a_h) + b_{k-1}^{pv}(s_h, a_h) - b_{k-1}^{pv}(s_h, a_h) \right) \right]
$$

$$
= \mathbb{E}_{\hat{p}_{k-1}, \pi^*} \left[ \sum_{h=1}^{H} \left( \tilde{\theta}_k(s_h, a_h) - \theta^*(s_h, a_h) \right) \right]
$$

$$
= \hat{\phi}_{k-1}(\pi^*)^\top \tilde{\theta}_k - \hat{\phi}_{k-1}(\pi^*)^\top \theta^*,
$$

where inequality (a) uses the definition of event $\mathcal{G}_{\text{Hoeff}}$.

Thus, by the definition of $\pi^k$, we have

$$
\Pr\left[ \hat{\phi}_{k-1}(\pi^k)^\top \tilde{\theta}_k^b > (\phi^{\pi^*})^\top \theta^* \mid F_{k-1} \right] \overset{(a)}{\geq} \Pr\left[ \hat{\phi}_{k-1}(\pi^*)^\top \tilde{\theta}_k^b > (\phi^{\pi^*})^\top \theta^* \mid F_{k-1} \right]
$$

$$
= \Pr\left[ \hat{\phi}_{k-1}(\pi^*)^\top \tilde{\theta}_k^b - (\phi^{\pi^*})^\top \theta^* > 0 \mid F_{k-1} \right]
$$

$$
\geq \Pr\left[ \hat{\phi}_{k-1}(\pi^*)^\top \tilde{\theta}_k - \hat{\phi}_{k-1}(\pi^*)^\top \theta^* > 0 \mid F_{k-1} \right]
$$

$$
\overset{(b)}{\geq} \frac{1}{2\sqrt{2\pi e}},
$$

where inequality (a) is due to the definition of $\pi^k$, and inequality (b) follows from Lemma 8. $\qquad\square$

Define event

$$\mathcal{G}_{\mathrm{KL}} := \left\{ \mathrm{KL}(\hat{p}_{k-1}(\cdot|s,a), p(\cdot|s,a)) \le \frac{L}{n_{k-1}(s,a)}, \ \forall k > 0, \forall (s,a) \in \mathcal{S} \times \mathcal{A} \right\}. \quad (18)$$

**Lemma 15** (Concentration of Transition). *It holds that*

$$\Pr[\mathcal{G}_{\mathrm{KL}}] \ge 1 - \delta'.$$

*Proof.* This lemma can be obtained by Theorem 3 and Lemma 3 in (Ménard et al., 2021). $\qquad\square$

Recall that for any $k > 0$ and $(s,a) \in \mathcal{S} \times \mathcal{A}$, $n_k(s,a)$ denotes the cumulative number of times that $(s,a)$ is visited at the end of episode $k$. For any $k > 0$, $h \in [H]$ and $(s,a) \in \mathcal{S} \times \mathcal{A}$, let $w_{k,h}(s,a)$ denote the probability that $(s,a)$ is visited at step $h$ in episode $k$, and let $w_k(s,a) := \sum_{h=1}^{H} w_{k,h}(s,a)$.

Define event

$$\mathcal{H} := \left\{ n_k(s,a) \ge \frac{1}{2} \sum_{k'=1}^{k} w_{k'}(s,a) - H \log\left( \frac{|\mathcal{S}||\mathcal{A}|H}{\delta'} \right), \ \forall k > 0, \forall (s,a) \in \mathcal{S} \times \mathcal{A} \right\}. \quad (19)$$

**Lemma 16** (Concentration of the Number of Visitations). *It holds that*

$$\Pr[\mathcal{H}] \ge 1 - \delta'.$$

*Proof.* This lemma can be obtained from Lemma F.4 in (Dann et al., 2017) and summing over $h \in [H]$. $\qquad\square$

Define event

$$\mathcal{F}_{\mathrm{UTran}}^{\mathrm{B}} := \Bigg\{ \left| \sum_{k'=1}^{k} \left( \mathbb{E}\left[ (\phi^{\pi^{k'}})^\top b_{k'-1}^{pv} | F_{k'-1} \right] - (\phi^{\pi^{k'}})^\top b_{k'-1}^{pv} \right) \right| \le 4H^2 r_{\max} \sqrt{k \log\left( \frac{4k}{\delta'} \right)},$$

$$\left| \sum_{k'=1}^{k} \left( \mathbb{E}\left[ \left\| \hat{\phi}_{k'-1}(\pi^{k'}) - \phi(\pi^{k'}) \right\|_1 | F_{k'-1} \right] - \left\| \hat{\phi}_{k'-1}(\pi^{k'}) - \phi(\pi^{k'}) \right\|_1 \right) \right|$$

$$\le 8H \sqrt{k \log\left( \frac{4k}{\delta'} \right)}, \ \forall k > 0 \Bigg\}.$$

**Lemma 17.** *It holds that*

$$\Pr\left[ \mathcal{F}_{\mathrm{UTran}}^{\mathrm{B}} \right] \ge 1 - 2\delta'.$$

*Proof.* This lemma can be obtained by a similar analysis as Lemma 11, and the facts that $|(\phi^{\pi^k})^\top b_{k-1}^{pv}| \le H^2 r_{\max}$ and $\|\hat{\phi}_{k-1}(\pi^k) - \phi^{\pi^k}\|_1 \le 2H$ for any $k \ge 1$. $\qquad\square$

**Lemma 18.** *Assume that event $\mathcal{F}_{\mathrm{UTran}}^{\mathrm{B}} \cap \mathcal{G}_{\mathrm{KL}} \cap \mathcal{H}$ holds. Then, we have*

$$\sum_{k=1}^{K} \mathbb{E}\left[ \left\| \hat{\phi}_{k-1}(\pi^k) - \phi^{\pi^k} \right\|_1 | F_{k-1} \right] \le 24 e^{12} |\mathcal{S}|^{\frac{3}{2}} |\mathcal{A}|^{\frac{3}{2}} H^{\frac{3}{2}} \sqrt{KL \log(2KH)}$$

$$+ 192 e^{12} |\mathcal{S}|^2 |\mathcal{A}|^2 H^2 L \log\left( \frac{2KH|\mathcal{S}||\mathcal{A}|}{\delta'} \right).$$

*Proof.* First, from Lemmas 29 and 30, we have

$$\sum_{k=1}^{K} \left\| \hat{\phi}_{k-1}(\pi) - \phi(\pi) \right\|_1$$

$$\le e^{12} |\mathcal{S}||\mathcal{A}| \sum_{k=1}^{K} \sum_{h=1}^{H} \sum_{(s,a) \in D_k} w_h^{\pi^k}(s,a) \left( 8H \sqrt{\frac{L}{n_{k-1}(s,a)}} + \frac{46H^2 L}{n_{k-1}(s,a)} \right)$$

$$+ e^{12}|\mathcal{S}||\mathcal{A}|H \sum_{k=1}^{K} \sum_{h=1}^{H} \sum_{(s,a) \notin D_k} w_h^{\pi^k}(s,a)$$

$$\leq 8e^{12}|\mathcal{S}||\mathcal{A}|H\sqrt{L} \sqrt{\sum_{k=1}^{K} \sum_{h=1}^{H} \sum_{(s,a) \in D_k} w_h^{\pi^k}(s,a)} \sqrt{\sum_{k=1}^{K} \sum_{h=1}^{H} \sum_{(s,a) \in D_k} \frac{w_h^{\pi^k}(s,a)}{n_{k-1}(s,a)}}$$

$$+ 46e^{12}|\mathcal{S}||\mathcal{A}|H^2 L \sum_{k=1}^{K} \sum_{h=1}^{H} \sum_{(s,a) \in D_k} \frac{w_h^{\pi^k}(s,a)}{n_{k-1}(s,a)} + 8e^{12}|\mathcal{S}|^2|\mathcal{A}|^2 H^2 \log\left(\frac{|\mathcal{S}||\mathcal{A}|H}{\delta'}\right)$$

$$\leq 16e^{12}|\mathcal{S}|^{\frac{3}{2}}|\mathcal{A}|^{\frac{3}{2}} H^{\frac{3}{2}} \sqrt{KL\log(2KH)} + 184e^{12}|\mathcal{S}|^2|\mathcal{A}|^2 H^2 L \log(2KH)$$

$$+ 8e^{12}|\mathcal{S}|^2|\mathcal{A}|^2 H^2 \log\left(\frac{|\mathcal{S}||\mathcal{A}|H}{\delta'}\right)$$

$$\leq 16e^{12}|\mathcal{S}|^{\frac{3}{2}}|\mathcal{A}|^{\frac{3}{2}} H^{\frac{3}{2}} \sqrt{KL\log(2KH)} + 192e^{12}|\mathcal{S}|^2|\mathcal{A}|^2 H^2 L \log\left(\frac{2KH|\mathcal{S}||\mathcal{A}|}{\delta'}\right).$$

Next, we have

$$\sum_{k=1}^{K} \mathbb{E}\left[\left\|\hat{\phi}_{k-1}(\pi^k) - \phi^{\pi^k}\right\|_1 |F_{k-1}\right]$$

$$\leq \sum_{k=1}^{K} \left\|\hat{\phi}_{k-1}(\pi^k) - \phi^{\pi^k}\right\|_1 + \sum_{k=1}^{K} \left(\mathbb{E}\left[\left\|\hat{\phi}_{k-1}(\pi^k) - \phi^{\pi^k}\right\|_1 |F_{k-1}\right] - \left\|\hat{\phi}_{k-1}(\pi^k) - \phi^{\pi^k}\right\|_1\right)$$

$$\leq 16e^{12}|\mathcal{S}|^{\frac{3}{2}}|\mathcal{A}|^{\frac{3}{2}} H^{\frac{3}{2}} \sqrt{KL\log(2KH)} + 192e^{12}|\mathcal{S}|^2|\mathcal{A}|^2 H^2 L \log\left(\frac{2KH|\mathcal{S}||\mathcal{A}|}{\delta'}\right)$$

$$+ 8H\sqrt{K\log\left(\frac{4K}{\delta'}\right)}$$

$$\leq 24e^{12}|\mathcal{S}|^{\frac{3}{2}}|\mathcal{A}|^{\frac{3}{2}} H^{\frac{3}{2}} \sqrt{KL\log(2KH)} + 192e^{12}|\mathcal{S}|^2|\mathcal{A}|^2 H^2 L \log\left(\frac{2KH|\mathcal{S}||\mathcal{A}|}{\delta'}\right).$$

$\square$

**Lemma 19.** *Assume that event $\mathcal{F}_{\text{UTran}}^{\text{B}}$ holds. Then, we have*

$$\sum_{k=1}^{K} \mathbb{E}\left[(\phi^{\pi^k})^\top b_{k-1}^{pv}|F_{k-1}\right] \leq 20|\mathcal{S}||\mathcal{A}|H^2 r_{\max}\sqrt{K}\log\left(\frac{4KH|\mathcal{S}||\mathcal{A}|}{\delta'}\right).$$

*Proof.* It holds that

$$\sum_{k=1}^{K} \mathbb{E}\left[(\phi^{\pi^k})^\top b_{k-1}^{pv}|F_{k-1}\right]$$

$$= \sum_{k=1}^{K} (\phi^{\pi^k})^\top b_{k-1}^{pv} + \sum_{k=1}^{K} \left(\mathbb{E}\left[(\phi^{\pi^k})^\top b_{k-1}^{pv}|F_{k-1}\right] - (\phi^{\pi^k})^\top b_{k-1}^{pv}\right)$$

$$\leq \sum_{k=1}^{K} \sum_{h=1}^{H} \sum_{s,a} w_h^{\pi^k}(s,a) \left(2Hr_{\max}\sqrt{\frac{\log\left(\frac{KH|\mathcal{S}||\mathcal{A}|}{\delta'}\right)}{n_{k-1}(s,a)}} \wedge Hr_{\max}\right) + 4H^2 r_{\max}\sqrt{K\log\left(\frac{4K}{\delta'}\right)}$$

$$\leq 2Hr_{\max}\sqrt{\log\left(\frac{KH|\mathcal{S}||\mathcal{A}|}{\delta'}\right)} \sum_{k=1}^{K} \sum_{h=1}^{H} \sum_{(s,a) \in D_k} \frac{w_h^{\pi^k}(s,a)}{\sqrt{n_{k-1}(s,a)}}$$

$$+ Hr_{\max} \sum_{k=1}^{K} \sum_{h=1}^{H} \sum_{(s,a)\notin D_k} w_h^{\pi^k}(s,a) + 4H^2 r_{\max} \sqrt{K \log\left(\frac{4K}{\delta'}\right)}$$

$$\leq 2Hr_{\max} \sqrt{\log\left(\frac{KH|\mathcal{S}||\mathcal{A}|}{\delta'}\right)} \cdot \sqrt{KH} \cdot \sqrt{\sum_{k=1}^{K} \sum_{h=1}^{H} \sum_{(s,a)\in D_k} \frac{w_h^{\pi^k}(s,a)}{n_{k-1}(s,a)}}$$

$$+ 8|\mathcal{S}||\mathcal{A}|H^2 r_{\max} \log\left(\frac{|\mathcal{S}||\mathcal{A}|H}{\delta'}\right) + 4H^2 r_{\max} \sqrt{K \log\left(\frac{4K}{\delta'}\right)}$$

$$\leq 2Hr_{\max} \sqrt{\log\left(\frac{KH|\mathcal{S}||\mathcal{A}|}{\delta'}\right)} \cdot \sqrt{KH} \cdot \sqrt{4|\mathcal{S}||\mathcal{A}| \log(2KH)}$$

$$+ 8|\mathcal{S}||\mathcal{A}|H^2 r_{\max} \log\left(\frac{|\mathcal{S}||\mathcal{A}|H}{\delta'}\right) + 4H^2 r_{\max} \sqrt{K \log\left(\frac{4K}{\delta'}\right)}$$

$$\leq 16|\mathcal{S}||\mathcal{A}|H^2 r_{\max} \sqrt{K} \log\left(\frac{4KH|\mathcal{S}||\mathcal{A}|}{\delta'}\right).$$

$\square$

*Proof of Theorem 3.* Letting $\delta' = \frac{\delta}{8}$, we have $\Pr[\mathcal{E} \cap \mathcal{F}_{\text{KTran}}^{\text{B}} \cap \mathcal{G}_{\text{Hoeff}} \cap \mathcal{G}_{\text{KL}} \cap \mathcal{H} \cap \mathcal{F}_{\text{UTran}}^{\text{B}}] \leq 1 - \delta$. Then, to prove this theorem, it suffices to prove the regret bound when event $\mathcal{E} \cap \mathcal{F}_{\text{KTran}}^{\text{B}} \cap \mathcal{G}_{\text{Hoeff}} \cap \mathcal{G}_{\text{KL}} \cap \mathcal{H} \cap \mathcal{F}_{\text{UTran}}^{\text{B}}$ holds.

Assume that event $\mathcal{E} \cap \mathcal{F}_{\text{KTran}}^{\text{B}} \cap \mathcal{G}_{\text{Hoeff}} \cap \mathcal{G}_{\text{KL}} \cap \mathcal{H} \cap \mathcal{F}_{\text{UTran}}^{\text{B}}$ holds. Then, we have

$$\mathcal{R}(K) = \sum_{k=1}^{K} \left((\phi^{\pi^*})^\top \theta^* - (\phi^{\pi^k})^\top \theta^*\right)$$

$$= \sum_{k=1}^{K} \left(\mathbb{E}\left[(\phi^{\pi^*})^\top \theta^* - (\phi^{\pi^k})^\top \theta^* | F_{k-1}\right] + \mathbb{E}\left[(\phi^{\pi^k})^\top \theta^* | F_{k-1}\right] - (\phi^{\pi^k})^\top \theta^*\right)$$

$$= \sum_{k=1}^{K} \left(\mathbb{E}\left[(\phi^{\pi^*})^\top \theta^* - (\phi^{\pi^k})^\top \theta^* | F_{k-1}\right]\right) + 4Hr_{\max} \sqrt{K \log\left(\frac{4K}{\delta'}\right)}. \tag{20}$$

For the first term, we have

$$\sum_{k=1}^{K} \mathbb{E}\left[(\phi^{\pi^*})^\top \theta^* - (\phi^{\pi^k})^\top \theta^* | F_{k-1}\right]$$

$$= \sum_{k=1}^{K} \left(\mathbb{E}\left[(\phi^{\pi^*})^\top \theta^* - \hat{\phi}_{k-1}(\pi^k)^\top \tilde{\theta}_k^b | F_{k-1}\right] + \mathbb{E}\left[\hat{\phi}_{k-1}(\pi^k)^\top \tilde{\theta}_k^b - (\phi^{\pi^k})^\top \theta^* | F_{k-1}\right]\right). \tag{21}$$

In the following, we prove

$$\mathbb{E}\left[(\phi^{\pi^*})^\top \theta^* - \hat{\phi}_{k-1}(\pi^k)^\top \tilde{\theta}_k^b | F_{k-1}\right]$$

$$\leq 2\sqrt{2\pi e} \cdot \mathbb{E}\left[\left(\hat{\phi}_{k-1}(\pi^k)^\top \tilde{\theta}_k^b - \mathbb{E}\left[\hat{\phi}_{k-1}(\pi^k)^\top \tilde{\theta}_k^b | F_{k-1}\right]\right)^+ | F_{k-1}\right]. \tag{22}$$

If $\mathbb{E}[(\phi^{\pi^*})^\top \theta^* - \hat{\phi}_{k-1}(\pi^k)^\top \tilde{\theta}_k^b | F_{k-1}] < 0$, then Eq. (22) trivially holds.

Otherwise, letting $z := \mathbb{E}[(\phi^{\pi^*})^\top \theta^* - \hat{\phi}_{k-1}(\pi^k)^\top \tilde{\theta}_k^b | F_{k-1}]$, we have

$$\mathbb{E}\left[\left(\hat{\phi}_{k-1}(\pi^k)^\top \tilde{\theta}_k^b - \mathbb{E}\left[\hat{\phi}_{k-1}(\pi^k)^\top \tilde{\theta}_k^b | F_{k-1}\right]\right)^+ | F_{k-1}\right]$$

$$\geq z \Pr\left[\hat{\phi}_{k-1}(\pi^k)^\top \tilde{\theta}_k^b - \mathbb{E}\left[\hat{\phi}_{k-1}(\pi^k)^\top \tilde{\theta}_k^b | F_{k-1}\right] \geq z | F_{k-1}\right]$$

$$\geq \left(\mathbb{E}\left[(\phi^{\pi^*})^\top \theta^* - \hat{\phi}_{k-1}(\pi^k)^\top \tilde{\theta}_k^b | F_{k-1}\right]\right) \cdot \Pr\left[\hat{\phi}_{k-1}(\pi^k)^\top \tilde{\theta}_k^b \geq (\phi^{\pi^*})^\top \theta^* | F_{k-1}\right]$$

$$\overset{(a)}{\geq} \left(\mathbb{E}\left[(\phi^{\pi^*})^\top \theta^* - \hat{\phi}_{k-1}(\pi^k)^\top \tilde{\theta}_k^b | F_{k-1}\right]\right) \cdot \frac{1}{2\sqrt{2\pi e}},$$

where inequality (a) uses Lemma 14. Thus, we complete the proof of Eq. (22).

Let $\xi'_k \in \mathbb{R}^{|\mathcal{S}||\mathcal{A}|}$ be an i.i.d. random variable with $\xi$ given $F_{k-1}$. Then, using Lemma 9 with $p' = \hat{p}_{k-1}$, $x_{k-1} = \hat{\theta}_{k-1} + b^{pv}_{k-1}$ and $\tilde{\pi}^k = \pi^k$, we have

$$\mathbb{E}\left[(\phi^{\pi^*})^\top \theta^* - \hat{\phi}_{k-1}(\pi^k)^\top \tilde{\theta}_k^b | F_{k-1}\right]$$

$$\leq 2\sqrt{2\pi e} \cdot \mathbb{E}\left[\left(\hat{\phi}_{k-1}(\pi^k)^\top \tilde{\theta}_k^b - \mathbb{E}\left[\hat{\phi}_{k-1}(\pi^k)^\top \tilde{\theta}_k^b | F_{k-1}\right]\right)^+ | F_{k-1}\right]$$

$$\leq 2\sqrt{2\pi e} \cdot \mathbb{E}\left[|\hat{\phi}_{k-1}(\pi^k)^\top \xi_k| + |\hat{\phi}_{k-1}(\pi^k)^\top \xi'_k| \,| F_{k-1}\right].$$

Plugging the above inequality into Eq. (21) and using Lemma 10 with $\delta_k = \frac{1}{k^4}$ and $L_X = \frac{H}{\sqrt{\alpha\lambda}}$, we have

$$\sum_{k=1}^K \mathbb{E}\left[(\phi^{\pi^*})^\top \theta^* - (\phi^{\pi^k})^\top \theta^* | F_{k-1}\right]$$

$$= \sum_{k=1}^K \left(2\sqrt{2\pi e} \cdot \mathbb{E}\left[|\hat{\phi}_{k-1}(\pi^k)^\top \xi_k| + |\hat{\phi}_{k-1}(\pi^k)^\top \xi'_k| \,| F_{k-1}\right]\right.$$

$$\left. + \mathbb{E}\left[\hat{\phi}_{k-1}(\pi^k)^\top \left(\hat{\theta}_{k-1} + b^{pv}_{k-1} + \xi_k\right) - (\phi^{\pi^k})^\top \theta^* | F_{k-1}\right]\right)$$

$$= \sum_{k=1}^K \left(\left(2\sqrt{2\pi e} + 1\right) \cdot \mathbb{E}\left[|\hat{\phi}_{k-1}(\pi^k)^\top \xi_k| \,| F_{k-1}\right] + 2\sqrt{2\pi e} \cdot \mathbb{E}\left[|\hat{\phi}_{k-1}(\pi^k)^\top \xi'_k| \,| F_{k-1}\right]\right.$$

$$\left. + \mathbb{E}\left[\hat{\phi}_{k-1}(\pi^k)^\top \left(\hat{\theta}_{k-1} + b^{pv}_{k-1}\right) - (\phi^{\pi^k})^\top \theta^* | F_{k-1}\right]\right)$$

$$\leq \sum_{k=1}^K \left(\left(4\sqrt{2\pi e} + 1\right)\sqrt{\alpha} \cdot \nu(k-1)\left(\sqrt{|\mathcal{S}||\mathcal{A}|} + 4\sqrt{\log(k)}\right)\mathbb{E}\left[\left\|\hat{\phi}_{k-1}(\pi^k)\right\|_{\Sigma_{k-1}^{-1}} | F_{k-1}\right]\right.$$

$$+ \left(4\sqrt{2\pi e} + 1\right)\sqrt{\alpha} \cdot \nu(k-1)\frac{\sqrt{|\mathcal{S}||\mathcal{A}|}}{k^2} \cdot \frac{H}{\sqrt{\alpha\lambda}}$$

$$\left. + \mathbb{E}\left[\hat{\phi}_{k-1}(\pi^k)^\top \left(\hat{\theta}_{k-1} + b^{pv}_{k-1}\right) - (\phi^{\pi^k})^\top \theta^* | F_{k-1}\right]\right). \tag{23}$$

We have

$$\mathbb{E}\left[\hat{\phi}_{k-1}(\pi^k)^\top \left(\hat{\theta}_{k-1} + b^{pv}_{k-1}\right) - (\phi^{\pi^k})^\top \theta^* | F_{k-1}\right]$$

$$= \mathbb{E}\left[\hat{\phi}_{k-1}(\pi^k)^\top \left(\hat{\theta}_{k-1} - \theta^*\right) | F_{k-1}\right] + \mathbb{E}\left[\left(\hat{\phi}_{k-1}(\pi^k) - \phi^{\pi^k}\right)^\top \theta^* | F_{k-1}\right]$$

$$+ \mathbb{E}\left[\hat{\phi}_{k-1}(\pi^k)^\top b^{pv}_{k-1} | F_{k-1}\right]$$

$$\leq \sqrt{\alpha} \cdot \nu(k-1)\mathbb{E}\left[\left\|\hat{\phi}_{k-1}(\pi^k)\right\|_{\Sigma_{k-1}^{-1}} | F_{k-1}\right] + r_{\max}\mathbb{E}\left[\left\|\hat{\phi}_{k-1}(\pi^k) - \phi^{\pi^k}\right\|_1 | F_{k-1}\right]$$

$$+ \mathbb{E}\left[\hat{\phi}_{k-1}(\pi^k)^\top b^{pv}_{k-1} | F_{k-1}\right].$$

Hence, plugging the above inequality into Eq. (23), we have

$$\sum_{k=1}^{K} \mathbb{E}\left[(\phi^{\pi^*})^\top \theta^* - (\phi^{\pi^k})^\top \theta^* | F_{k-1}\right]$$

$$\leq \sum_{k=1}^{K} \left( \left(4\sqrt{2\pi e} + 2\right)\sqrt{\alpha} \cdot \nu(k-1)\left(\sqrt{|\mathcal{S}||\mathcal{A}|} + 4\sqrt{\log(k)}\right) \cdot \mathbb{E}\left[\left\|\hat{\phi}_{k-1}(\pi^k)\right\|_{\Sigma_{k-1}^{-1}} | F_{k-1}\right] \right.$$

$$+ \left(4\sqrt{2\pi e} + 1\right) \cdot \nu(k-1)\frac{H}{k^2}\sqrt{\frac{|\mathcal{S}||\mathcal{A}|}{\lambda}} + r_{\max}\mathbb{E}\left[\left\|\hat{\phi}_{k-1}(\pi^k) - \phi^{\pi^k}\right\|_1 | F_{k-1}\right]$$

$$+ \mathbb{E}\left[\hat{\phi}_{k-1}(\pi^k)^\top b_{k-1}^{pv} | F_{k-1}\right] \right).$$

Here we have

$$\sum_{k=1}^{K} \mathbb{E}\left[\left\|\hat{\phi}_{k-1}(\pi^k)\right\|_{\Sigma_{k-1}^{-1}} | F_{k-1}\right]$$

$$\leq \sum_{k=1}^{K} \mathbb{E}\left[\left\|\phi^{\pi^k}\right\|_{\Sigma_{k-1}^{-1}} | F_{k-1}\right] + \sum_{k=1}^{K} \mathbb{E}\left[\left\|\hat{\phi}_{k-1}(\pi^k) - \phi^{\pi^k}\right\|_{\Sigma_{k-1}^{-1}} | F_{k-1}\right]$$

$$\leq \sum_{k=1}^{K} \mathbb{E}\left[\left\|\phi^{\pi^k}\right\|_{\Sigma_{k-1}^{-1}} | F_{k-1}\right] + \frac{1}{\sqrt{\alpha\lambda}}\sum_{k=1}^{K} \mathbb{E}\left[\left\|\hat{\phi}_{k-1}(\pi^k) - \phi^{\pi^k}\right\|_1 | F_{k-1}\right]$$

$$\overset{(a)}{\leq} 4H\sqrt{\frac{K}{\alpha\lambda}\log\left(\frac{4K}{\delta'}\right)} + \sqrt{2Km|\mathcal{S}||\mathcal{A}| \cdot \max\left\{\frac{H^2}{m\alpha\lambda}, 1\right\} \cdot \log\left(1 + \frac{KH^2}{\alpha\lambda|\mathcal{S}||\mathcal{A}|m}\right)}$$

$$+ \frac{1}{\sqrt{\alpha\lambda}}\sum_{k=1}^{K} \mathbb{E}\left[\left\|\hat{\phi}_{k-1}(\pi^k) - \phi^{\pi^k}\right\|_1 | F_{k-1}\right],$$

where inequality (a) uses Eq. (16).

In addition, we have

$$\sum_{k=1}^{K} \mathbb{E}\left[\hat{\phi}_{k-1}(\pi^k)^\top b_{k-1}^{pv} | F_{k-1}\right]$$

$$\leq \sum_{k=1}^{K} \mathbb{E}\left[(\phi^{\pi^k})^\top b_{k-1}^{pv} | F_{k-1}\right] + \sum_{k=1}^{K} \mathbb{E}\left[\left\|\hat{\phi}_{k-1}(\pi^k) - \phi^{\pi^k}\right\|_1 \left\|b_{k-1}^{pv}\right\|_\infty | F_{k-1}\right]$$

$$\leq \sum_{k=1}^{K} \mathbb{E}\left[(\phi^{\pi^k})^\top b_{k-1}^{pv} | F_{k-1}\right] + Hr_{\max}\sum_{k=1}^{K} \mathbb{E}\left[\left\|\hat{\phi}_{k-1}(\pi^k) - \phi^{\pi^k}\right\|_1 | F_{k-1}\right].$$

Therefore, plugging the above three equations into Eq. (20), we have

$$\mathcal{R}(K)$$

$$\leq \left(4\sqrt{2\pi e} + 2\right)\sqrt{\alpha} \cdot \nu(K)\left(\sqrt{|\mathcal{S}||\mathcal{A}|} + 4\sqrt{\log(K)}\right) \cdot$$

$$\left(4H\sqrt{\frac{K}{\alpha\lambda}\log\left(\frac{4K}{\delta'}\right)} + \sqrt{2Km|\mathcal{S}||\mathcal{A}|\max\left\{\frac{H^2}{m\alpha\lambda}, 1\right\}\log\left(1 + \frac{KH^2}{\alpha\lambda|\mathcal{S}||\mathcal{A}|m}\right)}\right)$$

$$+ \left(\left(4\sqrt{2\pi e} + 2\right)\frac{\nu(K)}{\sqrt{\lambda}}\left(\sqrt{|\mathcal{S}||\mathcal{A}|} + 4\sqrt{\log(K)}\right) + 2Hr_{\max}\right)\sum_{k=1}^{K} \mathbb{E}\left[\left\|\hat{\phi}_{k-1}(\pi^k) - \phi^{\pi^k}\right\|_1 | F_{k-1}\right]$$

$$+ \sum_{k=1}^{K} \mathbb{E}\left[(\phi^{\pi^k})^\top b_{k-1}^{pv} | F_{k-1}\right] + 2\left(4\sqrt{2\pi e} + 1\right)H \cdot \nu(K)\sqrt{\frac{|\mathcal{S}||\mathcal{A}|}{\lambda}} + 4Hr_{\max}\sqrt{K\log\left(\frac{4K}{\delta'}\right)}$$

$$\stackrel{(a)}{=} \tilde{O}\Bigg( \exp\left(\frac{Hr_{\max}}{m}\right) \nu(K)\sqrt{|\mathcal{S}||\mathcal{A}|} \left(\sqrt{Km|\mathcal{S}||\mathcal{A}|\max\left\{\frac{H^2}{m\alpha\lambda}, 1\right\}} + H\sqrt{\frac{K}{\alpha\lambda}}\right)$$

$$+ \left(\nu(K)\sqrt{\frac{|\mathcal{S}||\mathcal{A}|}{\lambda}} + Hr_{\max}\right) |\mathcal{S}|^2 |\mathcal{A}|^{\frac{3}{2}} H^{\frac{3}{2}} \sqrt{K}\Bigg),$$

where in equality (a), we use Lemmas 18 and 19, and the last three terms are absorbed into $\tilde{O}(\cdot)$. $\quad\square$

## D  PROOFS FOR RL WITH SUM SEGMENT FEEDBACK

In this section, we provide the proofs for RL with sum segment feedback.

### D.1  PROOF FOR THE REGRET UPPER BOUND WITH KNOWN TRANSITION

We first prove the regret upper bound (Theorem 4) of algorithm E-LinUCB for known transition.

Define event

$$\mathcal{J} := \Bigg\{ \left\| \sum_{k=1}^{K_0} \left( \sum_{i=1}^{m} \phi^{\tau_i^k}(\phi^{\tau_i^k})^\top - \mathbb{E}_{\tau_i \sim \pi^k}\left[\sum_{i=1}^{m} \phi(\tau_i)\phi(\tau_i)^\top\right]\right) \right\|$$

$$\leq \frac{4H^2}{m}\sqrt{K_0 \log\left(\frac{2|\mathcal{S}||\mathcal{A}|}{\delta'}\right)} + \frac{4H^2}{m}\log\left(\frac{2|\mathcal{S}||\mathcal{A}|}{\delta'}\right) \Bigg\}. \tag{24}$$

**Lemma 20** (Concentration of Initial Sampling). *It holds that*

$$\Pr[\mathcal{J}] \geq 1 - \delta'.$$

*Proof.* Note that $\pi^1, \ldots, \pi^{K_0}$ and $K_0$ are fixed before sampling, $\mathbb{E}[\sum_{i=1}^{m} \phi^{\tau_i^k}(\phi^{\tau_i^k})^\top] = \mathbb{E}_{\tau_i \sim \pi^k}\left[\sum_{i=1}^{m} \phi(\tau_i)\phi(\tau_i)^\top\right]$, and $\|\sum_{i=1}^{m} \phi^{\tau_i^k}(\phi^{\tau_i^k})^\top\| \leq \frac{H^2}{m}$. Then, using the matrix Bernstein inequality (Theorem 6.1.1 in (Tropp et al., 2015)), we can obtain this lemma. $\quad\square$

**Lemma 21** (E-optimal Design). *Assume that event $\mathcal{J}$ holds. Then, we have*

$$\left\| \left( \sum_{k=1}^{K_0} \sum_{i=1}^{m} \phi^{\tau_i^k}(\phi^{\tau_i^k})^\top \right)^{-1} \right\| \leq \frac{1}{H^2}.$$

*Proof.* Using the guarantee of the rounding procedure ROUND (Theorem 1.1 in (Allen-Zhu et al., 2021)) and the fact that $K_0 \geq \frac{|\mathcal{S}||\mathcal{A}|}{\gamma^2}$, we have

$$\left\| \left( \sum_{k=1}^{K_0} \mathbb{E}_{\tau_i \sim \pi^k}\left[\sum_{i=1}^{m} \phi(\tau_i)\phi(\tau_i)^\top\right] \right)^{-1} \right\|$$

$$\leq (1+\gamma) \left\| \left( K_0 \sum_{\pi \in \Pi} w^*(\pi) \cdot \mathbb{E}_{\tau_i \sim \pi^k}\left[\sum_{i=1}^{m} \phi(\tau_i)\phi(\tau_i)^\top\right] \right)^{-1} \right\|$$

$$\leq \frac{(1+\gamma)z^*}{K_0}.$$

Let $\sigma_{\min}(\cdot)$ denote the minimum eigenvalue. Then, we have

$$\sigma_{\min}\left( \sum_{k=1}^{K_0} \sum_{i=1}^{m} \phi^{\tau_i^k}(\phi^{\tau_i^k})^\top \right)$$

$$= \sigma_{\min}\left( \sum_{k=1}^{K_0} \mathbb{E}_{\tau_i \sim \pi^k}\left[\sum_{i=1}^{m} \phi(\tau_i)\phi(\tau_i)^\top\right] + \sum_{k=1}^{K_0}\sum_{i=1}^{m} \phi^{\tau_i^k}(\phi^{\tau_i^k})^\top - \sum_{k=1}^{K_0} \mathbb{E}_{\tau_i \sim \pi^k}\left[\sum_{i=1}^{m} \phi(\tau_i)\phi(\tau_i)^\top\right] \right)$$

$$\geq \sigma_{\min}\left(\sum_{k=1}^{K_0}\mathbb{E}_{\tau_i\sim\pi^k}\left[\sum_{i=1}^{m}\phi(\tau_i)\phi(\tau_i)^{\top}\right]\right) - \left\|\sum_{k=1}^{K_0}\sum_{i=1}^{m}\phi^{\tau_i^k}(\phi^{\tau_i^k})^{\top} - \sum_{k=1}^{K_0}\mathbb{E}_{\tau_i\sim\pi^k}\left[\sum_{i=1}^{m}\phi(\tau_i)\phi(\tau_i)^{\top}\right]\right\|$$

$$\geq \frac{K_0}{(1+\gamma)z^*} - \frac{4H^2}{m}\sqrt{\log\left(\frac{2|\mathcal{S}||\mathcal{A}|}{\delta'}\right)}\cdot\sqrt{K_0} - \frac{4H^2}{m}\log\left(\frac{2|\mathcal{S}||\mathcal{A}|}{\delta'}\right). \tag{25}$$

Let $x = \sqrt{K_0}$ and

$$f(x) = \frac{1}{(1+\gamma)z^*}\cdot x^2 - \frac{4H^2}{m}\sqrt{\log\left(\frac{2|\mathcal{S}||\mathcal{A}|}{\delta'}\right)}\cdot x - \frac{4H^2}{m}\log\left(\frac{2|\mathcal{S}||\mathcal{A}|}{\delta'}\right) - H^2.$$

According to the property of quadratic functions, when

$$x \geq \frac{\frac{4H^2}{m}\sqrt{\log\left(\frac{2|\mathcal{S}||\mathcal{A}|}{\delta'}\right)} + \sqrt{\left(\frac{4H^2}{m}\sqrt{\log\left(\frac{2|\mathcal{S}||\mathcal{A}|}{\delta'}\right)}\right)^2 + 4\cdot\frac{1}{(1+\gamma)z^*}\left(\frac{4H^2}{m}\log\left(\frac{2|\mathcal{S}||\mathcal{A}|}{\delta'}\right) + H^2\right)}}{2\cdot\frac{1}{(1+\gamma)z^*}}, \tag{26}$$

we have $f(x) \geq 0$.

To make Eq. (26) hold, it suffices to set

$$K_0 \geq \frac{(1+\gamma)^2(z^*)^2}{4}\cdot\left(2\cdot\left(\frac{4H^2}{m}\sqrt{\log\left(\frac{2|\mathcal{S}||\mathcal{A}|}{\delta'}\right)}\right)^2 + 2\cdot\left(\frac{4H^2}{m}\sqrt{\log\left(\frac{2|\mathcal{S}||\mathcal{A}|}{\delta'}\right)}\right)^2\right.$$

$$+ \frac{8}{(1+\gamma)z^*}\cdot 5H^2\log\left(\frac{2|\mathcal{S}||\mathcal{A}|}{\delta'}\right)\Bigg)$$

$$= \left(\frac{16H^4(1+\gamma)^2(z^*)^2}{m^2} + 10H^2(1+\gamma)z^*\right)\log\left(\frac{2|\mathcal{S}||\mathcal{A}|}{\delta'}\right).$$

Furthermore, since $\|\sum_{\pi\in\Pi}w^*(\pi)\mathbb{E}_{\tau_i\sim\pi^k}\left[\sum_{i=1}^{m}\phi(\tau_i)\phi(\tau_i)^{\top}\right]\| \leq H^2$ and then $z^* \geq \frac{1}{H^2}$, to make the right-hand-side in Eq. (25) no smaller than $H^2$, it suffices to set

$$K_0 \geq 26H^4(1+\gamma)^2(z^*)^2\log\left(\frac{2|\mathcal{S}||\mathcal{A}|}{\delta'}\right).$$

Therefore, combining the definition of $K_0$ and Eq. (25), we have

$$\sigma_{\min}\left(\sum_{k=1}^{K_0}\phi(\tau^k)\phi(\tau^k)^{\top}\right) \geq H^2,$$

which completes the proof. $\qquad\qquad\square$

**Lemma 22.** *For any $k > 0$,*

$$\sum_{k'=1}^{k}\log\left(1 + \sum_{i=1}^{m}\left\|\phi_i^{\tau_i^{k'}}\right\|_{(\Sigma_{k'-1})^{-1}}^2\right) = \log\left(\frac{\det(\Sigma_k)}{\det(\lambda I)}\right) \leq |\mathcal{S}||\mathcal{A}|\log\left(1 + \frac{kH^2}{\lambda|\mathcal{S}||\mathcal{A}|m}\right).$$

*Proof.* For any $k > 0$, it holds that

$$\det(\Sigma_k) = \det\left(\Sigma_{k-1} + \sum_{i=1}^{m}\phi^{\tau_i^k}(\phi^{\tau_i^k})^{\top}\right)$$

$$= \det(\Sigma_{k-1})\det\left(I + \sum_{i=1}^{m}(\Sigma_{k-1})^{-\frac{1}{2}}\phi^{\tau_i^k}(\phi^{\tau_i^k})^{\top}(\Sigma_{k-1})^{-\frac{1}{2}}\right)$$

$$= \det(\Sigma_{k-1}) \left( 1 + \sum_{i=1}^{m} \left\| \phi^{\tau_i^k} \right\|_{(\Sigma_{k-1})^{-1}}^2 \right)$$

$$= \det(\lambda I) \prod_{k'=1}^{k} \left( 1 + \sum_{i=1}^{m} \left\| \phi^{\tau_i^{k'}} \right\|_{(\Sigma_{k'-1})^{-1}}^2 \right).$$

Taking the logarithm on both sides, we have

$$\log \det(\Sigma_k) = \log \det(\lambda I) + \sum_{k'=1}^{k} \log \left( 1 + \sum_{i=1}^{m} \left\| \phi^{\tau_i^{k'}} \right\|_{(\Sigma_{k'-1})^{-1}}^2 \right).$$

Then,

$$\sum_{k'=1}^{k} \log \left( 1 + \sum_{i=1}^{m} \left\| \phi^{\tau_i^{k'}} \right\|_{(\Sigma_{k'-1})^{-1}}^2 \right) = \log \left( \frac{\det(\Sigma_k)}{\det(\lambda I)} \right)$$

$$\overset{(a)}{\le} \log \left( \frac{\left( \frac{\mathrm{tr}(\Sigma_k)}{|\mathcal{S}||\mathcal{A}|} \right)^{|\mathcal{S}||\mathcal{A}|}}{\lambda^{|\mathcal{S}||\mathcal{A}|}} \right)$$

$$= |\mathcal{S}||\mathcal{A}| \log \left( \frac{\mathrm{tr}(\Sigma_k)}{\lambda |\mathcal{S}||\mathcal{A}|} \right)$$

$$\le |\mathcal{S}||\mathcal{A}| \log \left( \frac{\lambda |\mathcal{S}||\mathcal{A}| + km \cdot \frac{H^2}{m^2}}{\lambda |\mathcal{S}||\mathcal{A}|} \right)$$

$$= |\mathcal{S}||\mathcal{A}| \log \left( 1 + \frac{kH^2}{\lambda |\mathcal{S}||\mathcal{A}|m} \right),$$

where (a) uses the arithmetic mean-geometric mean inequality. $\qquad \square$

**Lemma 23** (Elliptical Potential with Optimized Initialization). *Assume that event $\mathcal{J}$ holds. Then, for any $k \ge K_0 + 1$,*

$$\sum_{i=1}^{m} \left\| \phi^{\tau_i^k} \right\|_{(\Sigma_{k-1})^{-1}}^2 \le 1.$$

*Furthermore, for any $K \ge K_0 + 1$,*

$$\sum_{k=K_0+1}^{K} \sum_{i=1}^{m} \left\| \phi^{\tau_i^k} \right\|_{(\Sigma_{k-1})^{-1}} \le \sqrt{2Km|\mathcal{S}||\mathcal{A}| \log \left( 1 + \frac{KH^2}{\lambda |\mathcal{S}||\mathcal{A}|m} \right)}.$$

*Proof.* Using Lemma 21, for any $k \ge K_0 + 1$, we have

$$\sum_{i=1}^{m} \left\| \phi^{\tau_i^k} \right\|_{(\Sigma_{k-1})^{-1}}^2 = \sum_{i=1}^{m} \left\| \phi^{\tau_i^k} \right\|_{\left( \lambda I + \sum_{k'=1}^{K_0} \phi^{\tau^{k'}} (\phi^{\tau^{k'}})^\top + \sum_{k'=K_0+1}^{k-1} \phi^{\tau^{k'}} (\phi^{\tau^{k'}})^\top \right)^{-1}}^2$$

$$\le \sum_{i=1}^{m} \left\| \phi^{\tau_i^i} \right\|_{\left( \sum_{k'=1}^{K_0} \phi^{\tau^{k'}} (\phi^{\tau^{k'}})^\top \right)^{-1}}^2$$

$$\le m \cdot \frac{H^2}{m^2} \cdot \frac{1}{H^2}$$

$$\le 1.$$

Then, we have

$$\sum_{k=K_0+1}^{K} \sum_{i=1}^{m} \left\| \phi^{\tau_i^k} \right\|_{(\Sigma_{k-1})^{-1}} \le \sqrt{Km \sum_{k=K_0+1}^{K} \sum_{i=1}^{m} \left\| \phi^{\tau_i^k} \right\|_{(\Sigma_{k-1})^{-1}}^2}$$

$$\overset{(a)}{\leq} \sqrt{Km \cdot 2 \sum_{k=K_0+1}^{K} \log\left(1 + \sum_{i=1}^{m} \left\|\phi^{\tau_i^k}\right\|_{(\Sigma_{k-1})^{-1}}^2\right)}$$

$$\leq \sqrt{Km \cdot 2 \sum_{k=1}^{K} \log\left(1 + \sum_{i=1}^{m} \left\|\phi^{\tau_i^k}\right\|_{(\Sigma_{k-1})^{-1}}^2\right)}$$

$$\overset{(b)}{\leq} \sqrt{2Km|\mathcal{S}||\mathcal{A}| \log\left(1 + \frac{KH^2}{\lambda|\mathcal{S}||\mathcal{A}|m}\right)},$$

where (a) uses the fact that $x \leq 2\log(1+x)$ for any $x \in [0,1]$, and (b) follows from Lemma 22. $\qquad\square$

Define event

$$\mathcal{K} := \left\{ \left\|\hat{\theta}_k - \theta^*\right\|_{\Sigma_k} \leq \sqrt{\frac{H|\mathcal{S}||\mathcal{A}|}{m} \log\left(1 + \frac{kH^2}{\lambda|\mathcal{S}||\mathcal{A}|m}\right) + 2\log\left(\frac{1}{\delta'}\right)} + r_{\max}\sqrt{\lambda|\mathcal{S}||\mathcal{A}|}, \ \forall k > 0 \right\}. \tag{27}$$

**Lemma 24** (Concentration of $\hat{\theta}_k$ under Sum Feedback). *It holds that*

$$\Pr[\mathcal{K}] \geq 1 - \delta'.$$

*Proof.* Since the sum feedback on each segment is $\frac{H}{m}$-sub-Gaussian given the observation of transition and $\|\theta^*\| \leq r_{\max}\sqrt{|\mathcal{S}||\mathcal{A}|}$, using Lemma 2 in (Abbasi-Yadkori et al., 2011), we can obtain this lemma. $\qquad\square$

Define event

$$\mathcal{F}_{\mathrm{opt}}^{\mathsf{S}} := \left\{ \left| \sum_{k'=K_0+1}^{k} \left( \mathbb{E}_{\tau \sim \pi^{k'}} \left[ \|\phi^\tau\|_{(\Sigma_{k'-1})^{-1}} \mid F_{k'-1} \right] - \|\phi^\tau\|_{(\Sigma_{k'-1})^{-1}} \right) \right| \right.$$

$$\left. \leq 4\sqrt{k \log\left(\frac{4k}{\delta'}\right)}, \ \forall k \geq K_0 + 1 \right\}. \tag{28}$$

**Lemma 25** (Concentration of Visitation Indicators). *It holds that*

$$\Pr\left[\mathcal{F}_{\mathrm{opt}}^{\mathsf{S}}\right] \geq 1 - \delta'.$$

*Proof.* According to Lemma 23, we have that for any $k' \geq K_0 + 1$, $\|\phi^\tau\|_{(\Sigma_{k'-1})^{-1}} \leq 1$, and then $|\mathbb{E}_{\tau \sim \pi^{k'}}[\|\phi^\tau\|_{(\Sigma_{k'-1})^{-1}} \mid F_{k'-1}] - \|\phi^\tau\|_{(\Sigma_{k'-1})^{-1}}| \leq 2$.

Using the Azuma-Hoeffding inequality, we have that for any fixed $k \geq K_0 + 1$, with probability at least $1 - \frac{\delta'}{2k^2}$,

$$\left| \sum_{k'=K_0+1}^{k} \left( \mathbb{E}_{\tau \sim \pi^{k'}} \left[ \|\phi^\tau\|_{(\Sigma_{k'-1})^{-1}} \mid F_{k'-1} \right] - \|\phi^\tau\|_{(\Sigma_{k'-1})^{-1}} \right) \right| \leq \sqrt{2 \cdot 4(k - K_0 - 1) \log\left(\frac{4k^2}{\delta'}\right)}.$$

Since $\sum_{k=K_0+1}^{\infty} \frac{\delta'}{2k^2} \leq \delta'$, by a union bound over $k$, we have that with probability at least $\delta'$, for any $k \geq K_0 + 1$,

$$\left| \sum_{k'=K_0+1}^{k} \left( \mathbb{E}_{\tau \sim \pi^{k'}} \left[ \|\phi^\tau\|_{(\Sigma_{k'-1})^{-1}} \mid F_{k'-1} \right] - \|\phi^\tau\|_{(\Sigma_{k'-1})^{-1}} \right) \right| \leq \sqrt{2 \cdot 4(k - K_0 - 1) \log\left(\frac{4k^2}{\delta'}\right)}$$

$$\leq 4\sqrt{k \log\left(\frac{4k}{\delta'}\right)}.$$

$\square$

*Proof of Theorem 4.* Let $\delta' = \frac{\delta}{3}$. We have $\Pr[\mathcal{J} \cap \mathcal{K} \cap \mathcal{F}_{\mathrm{opt}}^{\mathrm{S}}] \geq 1 - \delta$. To prove this theorem, it suffices to prove the regret bound when event $\mathcal{J} \cap \mathcal{K} \cap \mathcal{F}_{\mathrm{opt}}^{\mathrm{S}}$ holds.

Assume that event $\mathcal{J} \cap \mathcal{K} \cap \mathcal{F}_{\mathrm{opt}}^{\mathrm{S}}$ holds. Then, we have

$$
\mathcal{R}(K) = \sum_{k=1}^{K} \left( (\phi^{\pi^*})^\top \theta - (\phi^{\pi^k})^\top \theta \right)
$$

$$
\overset{(a)}{\leq} \sum_{k=K_0+1}^{K} \left( (\phi^{\pi^*})^\top \hat{\theta}_{k-1} + \beta(k-1) \cdot \|\phi^{\pi^*}\|_{(\Sigma_{k-1})^{-1}} - (\phi^{\pi^k})^\top \theta \right) + K_0 H
$$

$$
\overset{(b)}{\leq} \sum_{k=K_0+1}^{K} \left( (\phi^{\pi^k})^\top \hat{\theta}_{k-1} + \beta(k-1) \cdot \|\phi^{\pi^k}\|_{(\Sigma_{k-1})^{-1}} - (\phi^{\pi^k})^\top \theta \right) + K_0 H
$$

$$
\leq \sum_{k=K_0+1}^{K} 2\beta(k-1) \cdot \|\phi^{\pi^k}\|_{(\Sigma_{k-1})^{-1}} + K_0 H
$$

$$
= 2\beta(K) \sum_{k=K_0+1}^{K} \|\mathbb{E}_{\tau \sim \pi^k}[\phi^\tau | F_{k-1}]\|_{(\Sigma_{k-1})^{-1}} + K_0 H
$$

$$
\overset{(c)}{\leq} 2\beta(K) \sum_{k=K_0+1}^{K} \mathbb{E}_{\tau \sim \pi^k}\left[ \|\phi^\tau\|_{(\Sigma_{k-1})^{-1}} | F_{k-1} \right] + K_0 H
$$

$$
= 2\beta(K) \sum_{k=K_0+1}^{K} \left( \mathbb{E}_{\tau \sim \pi^k}\left[ \|\phi^\tau\|_{(\Sigma_{k-1})^{-1}} | F_{k-1} \right] - \|\phi^\tau\|_{(\Sigma_{k-1})^{-1}} + \|\phi^\tau\|_{(\Sigma_{k-1})^{-1}} \right) + K_0 H
$$

$$
\leq 2\beta(K) \sum_{k=K_0+1}^{K} \left( \mathbb{E}_{\tau \sim \pi^k}\left[ \|\phi^\tau\|_{(\Sigma_{k-1})^{-1}} | F_{k-1} \right] - \|\phi^\tau\|_{(\Sigma_{k-1})^{-1}} + \sum_{i=1}^{m} \left\| \phi^{\tau_i^k} \right\|_{(\Sigma_{k-1})^{-1}} \right)
$$
$$
+ K_0 H, \tag{29}
$$

where (a) follows from Eq. (27), (b) is due to the definition of $\pi^k$, and (c) uses the Jensen inequality. Plugging Eq. (28) and Lemma 23 into Eq. (29) and using the fact that $\lambda := \frac{H}{r_{\max}^2 m}$, we have

$$
\mathcal{R}(K) \leq 2 \left( \sqrt{\frac{H|\mathcal{S}||\mathcal{A}|}{m} \log\left(1 + \frac{KH^2}{\lambda|\mathcal{S}||\mathcal{A}|m}\right)} + 2\log\left(\frac{1}{\delta'}\right) + r_{\max}\sqrt{\lambda|\mathcal{S}||\mathcal{A}|} \right) \cdot
$$
$$
\left( 4\sqrt{K\log\left(\frac{4K}{\delta}\right)} + \sqrt{2Km|\mathcal{S}||\mathcal{A}|\log\left(1 + \frac{KH^2}{\lambda|\mathcal{S}||\mathcal{A}|m}\right)} \right)
$$
$$
+ H\left[ \max\left\{ 26H^4(1+\gamma)^2(z^*)^2 \log\left(\frac{2|\mathcal{S}||\mathcal{A}|}{\delta'}\right), \frac{|\mathcal{S}||\mathcal{A}|}{\gamma^2} \right\} \right]
$$
$$
= O\left( |\mathcal{S}||\mathcal{A}|\sqrt{HK} \log\left( \left(1 + \frac{KHr_{\max}^2}{|\mathcal{S}||\mathcal{A}|m}\right)\frac{1}{\delta} \right) + (z^*)^2 H^5 \log\left(\frac{|\mathcal{S}||\mathcal{A}|}{\delta}\right) + |\mathcal{S}||\mathcal{A}|H \right).
$$

$\square$

## D.2 PROOF FOR THE REGRET LOWER BOUND WITH KNOWN TRANSITION

Now we prove the regret lower bound (Theorem 5) for RL with sum segment feedback and known transition.

*Proof of Theorem 5.* We construct a random instance $\mathcal{I}$ as follows. As shown in Figure 5, there are $n$ bandit states $s_1, \ldots, s_n$ (i.e., there is an optimal action and multiple suboptimal actions), a good absorbing state $s_{n+1}$ and a bad absorbing state $s_{n+2}$. The agent starts from $s_1, \ldots, s_n$ with equal probability $\frac{1}{n}$. For any $i \in [n]$, in state $s_i$, one action $a_J$ is uniformly chosen from $\mathcal{A}$ as the optimal

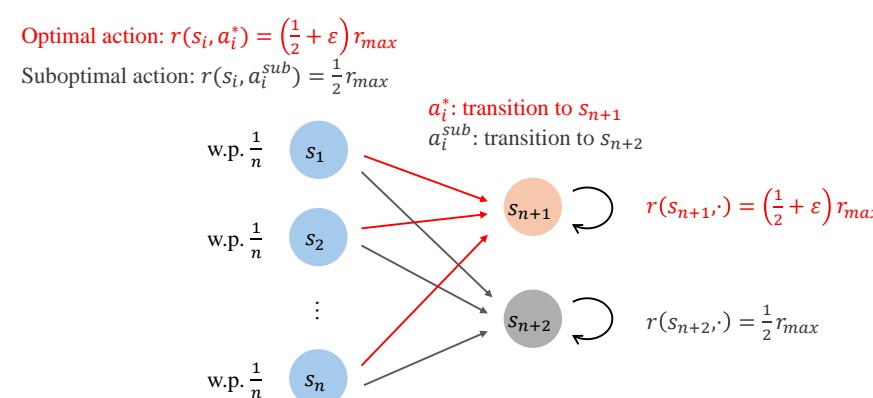

Figure 5: Instance for the lower bound under sum segment feedback and known transition.

action. In state $s_i$, under the optimal action $a_J$, the agent transitions to $s_{n+1}$ deterministically, and $r(s_i, a_J) = (\frac{1}{2} + \varepsilon)r_{\max}$, where $\varepsilon \in (0, \frac{1}{2}]$ is a parameter specified later; Under any suboptimal action $a \in \mathcal{A} \setminus \{s_J\}$, the agent transitions to $s_{n+2}$ deterministically, and $r(s_i, a) = \frac{1}{2}r_{\max}$. For all actions $a \in \mathcal{A}$, $r(s_{n+1}, a) = (\frac{1}{2} + \varepsilon)r_{\max}$ and $r(s_{n+2}, a) = \frac{1}{2}r_{\max}$. For any $(s, a) \in \mathcal{S} \times \mathcal{A}$, the reward distribution of $(s, a)$ is Gaussian distribution $\mathcal{N}(r(s, a), 1)$.

In this proof, we will also use an alternative uniform instance $\mathcal{I}_{\text{unif}}$. The only difference between $\mathcal{I}_{\text{unif}}$ and $\mathcal{I}$ is that for any $i \in [n]$, in state $s_i$, under all actions $a \in \mathcal{A}$, the agent transitions to $s_{n+2}$ deterministically, and $r(s_i, a) = \frac{1}{2}r_{\max}$.

Fix an algorithm $\mathbb{A}$. Let $\mathbb{E}_{\text{unif}}[\cdot]$ denote the expectation with respect to $\mathcal{I}_{\text{unif}}$. Let $\mathbb{E}_*[\cdot]$ denote the expectation with respect to $\mathcal{I}$. For any $i \in [n]$ and $j \in [|\mathcal{A}|]$, let $\mathbb{E}_{i,j}[\cdot]$ denote the expectation with respect to the case where $a_j$ is the optimal action in state $s_i$, and $N_{i,j}$ denote the number of episodes where algorithm $\mathbb{A}$ chooses $a_j$ in state $s_i$, i.e., $N_{i,j} = \sum_{k=1}^{K} \mathbb{1}\{\pi_1^k(s_i) = a_j\}$.

The KL divergence of the reward observations if taking $a_J$ in $s_i$ ($i \in [n]$) between $\mathcal{I}_{\text{unif}}$ and $\mathcal{I}$ is

$$\sum_{i=1}^{m} \text{KL}\left(\mathcal{N}\left(\frac{1}{2}r_{\max} \cdot \frac{H}{m}, \frac{H}{m}\right) \middle\| \mathcal{N}\left(\left(\frac{1}{2} + \varepsilon\right)r_{\max} \cdot \frac{H}{m}, \frac{H}{m}\right)\right)$$

$$= m \cdot \frac{\left(\frac{H}{m} \cdot r_{\max}\varepsilon\right)^2}{\frac{H}{m}}$$

$$= Hr_{\max}^2 \varepsilon^2.$$

In addition, the agent has probability only $\frac{1}{n}$ to arrive at (observe) state $s_i$.

Hence, using Lemma A.1 in (Auer et al., 2002), we have that for any $i \in [n]$, in state $s_i$,

$$\mathbb{E}_{i,j}[N_{i,j}] \le \mathbb{E}_{\text{unif}}[N_{i,j}] + \frac{K}{2}\sqrt{\frac{1}{n} \cdot \mathbb{E}_{\text{unif}}[N_{i,j}] \cdot Hr_{\max}^2 \varepsilon^2}.$$

Summing over $j \in [|\mathcal{A}|]$, using the Cauchy-Schwarz inequality and the fact that $\sum_{j=1}^{|\mathcal{A}|} \mathbb{E}_{\text{unif}}[N_{i,j}] = K$, we have

$$\sum_{j=1}^{|\mathcal{A}|} \mathbb{E}_{i,j}[N_{i,j}] \le K + \frac{K}{2}\sqrt{\frac{|\mathcal{A}|}{n} \cdot K \cdot Hr_{\max}^2 \varepsilon^2}$$

$$= K + \frac{Kr_{\max}\varepsilon}{2}\sqrt{\frac{|\mathcal{A}|HK}{n}}.$$

---

**Algorithm 4:** `LinUCB-Tran`

---

**Input:** $\delta, \delta' := \frac{\delta}{4}, \lambda := \frac{H}{m}, L := \log(\frac{3|\mathcal{S}||\mathcal{A}|H}{\delta'}) + S\log(8e(1+KH))$. For any $k \geq 1$,
$$\beta(k) := \sqrt{\frac{H|\mathcal{S}||\mathcal{A}|}{m}\log(1+\frac{kH^2}{\lambda|\mathcal{S}||\mathcal{A}|m}) + 2\log(\frac{1}{\delta'})} + r_{\max}\sqrt{\lambda|\mathcal{S}||\mathcal{A}|}.$$

1 **for** $k = 1, \ldots, K$ **do**

2    $\hat{\theta}_{k-1} \leftarrow (\lambda I + \sum_{k'=1}^{k-1}\sum_{i=1}^{m}\phi^{\tau_i^{k'}}(\phi^{\tau_i^{k'}})^\top)^{-1}\sum_{k'=1}^{k-1}\sum_{i=1}^{m}\phi^{\tau_i^{k'}}R_i^{k'}$;

3    $\Sigma_{k-1} \leftarrow \lambda I + \sum_{k'=1}^{k-1}\sum_{i=1}^{m}\phi^{\tau_i^{k'}}(\phi^{\tau_i^{k'}})^\top$;

4    $\pi^k \leftarrow \operatorname{argmax}_{\pi \in \Pi}((\hat{\phi}_{k-1}^\pi)^\top\hat{\theta}_{k-1} + \beta(k-1)\cdot\|\hat{\phi}_{k-1}^\pi\|_{(\Sigma_{k-1})^{-1}} + \sum_{s',a'}\mathbb{E}_{s_1\sim\rho}[B_1^{\pi;s',a';k}(s_1)])$,
     where $B_1^{\pi;s',a';k}(s_1)$ is defined in Eq. (31);

5    Play episode $k$ with policy $\pi^k$. Observe $\tau^k$ and sum segment feedback $\{R_i^k\}_{i=1}^m$;

---

Then, we have

$$\mathcal{R}(K) = \sum_{k=1}^{K}\mathbb{E}_*\left[V^* - V^{\pi^k}\right]$$

$$= \left(\frac{1}{2} + \varepsilon\right)r_{\max}HK - \frac{1}{n}\sum_{i=1}^{n}\left(\frac{1}{2}r_{\max}HK + \varepsilon r_{\max}H\cdot\frac{1}{|\mathcal{A}|}\sum_{j=1}^{|\mathcal{A}|}\mathbb{E}_{i,j}[N_{i,j}]\right)$$

$$= \varepsilon r_{\max}H\left(K - \frac{K}{|\mathcal{A}|} - \frac{Kr_{\max}\varepsilon}{2}\sqrt{\frac{HK}{n|\mathcal{A}|}}\right).$$

Recall that $n = |\mathcal{S}| - 2$. Let $|\mathcal{S}| \geq 3, |\mathcal{A}| \geq 2, K \geq \frac{n|\mathcal{A}|}{r_{\max}^2 H}$ and $\varepsilon = \frac{1}{2r_{\max}}\sqrt{\frac{n|\mathcal{A}|}{HK}}$. Then, we have

$$\mathcal{R}(K) = \Omega\left(\sqrt{|\mathcal{S}||\mathcal{A}|HK}\right).$$

$\square$

## D.3   PSEUDO-CODE AND DETAILED DESCRIPTION OF ALGORITHM `LinUCB-Tran`

Algorithm 4 presents the pseudo-code of `LinUCB-Tran`. In each episode $k$, similar to algorithm `E-LinUCB`, `LinUCB-Tran` first computes the least squares estimate of the reward parameter $\hat{\theta}_{k-1}$ and covariance matrix $\Sigma_{k-1}$ with past observations (Lines 2-3).

Then, we introduce the transition estimation in `LinUCB-Tran`. We first define some notation which also appears in algorithm `SegBiTS-Tran`. For any $k > 0$ and $(s,a) \in \mathcal{S} \times \mathcal{A}$, let $\hat{p}_k(\cdot|s,a)$ denote the empirical estimate of $p(\cdot|s,a)$, and $n_k(s,a)$ denote the number of times $(s,a)$ was visited up to the end of episode $k$. In addition, for any policy $\pi$, let $\hat{\phi}_k^\pi(s,a)$ denote the expected number of times $(s,a)$ is visited in an episode under policy $\pi$ on empirical MDP $\hat{p}_{k-1}$ (see Eq. (17) for the formal definition).

Below we establish a bound for the deviation between $\hat{\phi}_{k-1}^\pi$ and $\phi^\pi$. For ease of analysis, we first connect $\phi^\pi$ with a newly-defined visitation value function $G_h^{\pi;s',a'}(s;p)$. For any transition model $p'$, policy $\pi$ and $(s',a') \in \mathcal{S} \times \mathcal{A}$, if regarding hitting $(s',a')$ as an instantaneous reward one, then we can define a visitation value function:

$$\begin{cases} G_h^{\pi;s',a'}(s;p') = & \mathbb{1}\{s = s', \pi_h(s) = a'\} + p(\cdot|s,\pi_h(s))^\top G_{h+1}^{\pi;s',a'}(\cdot), & \forall s \in \mathcal{S}, \ \forall h \in [H], \\ G_{H+1}^{\pi;s',a'}(s;p') = & 0, & \forall s \in \mathcal{S}. \end{cases}$$
(30)

$G_h^{\pi;s',a'}(s;p')$ denotes the expected cumulative number of times $(s',a')$ was hit starting from $s$ at step $h$ under policy $\pi$ on MDP $p'$, till the end of this episode. It holds that $\phi^\pi(s',a') = \mathbb{E}_{s_1\sim\rho}[G_1^{\pi;s',a'}(s_1|p)]$ and $\hat{\phi}_{k-1}^\pi(s',a') = \mathbb{E}_{s_1\sim\rho}[G_1^{\pi;s',a'}(s_1|\hat{p}_{k-1})]$ for any $(s',a') \in \mathcal{S} \times \mathcal{A}$.

With the definition of $G_h^{\pi;s',a'}$, bounding the deviation between $\hat{\phi}_{k-1}^\pi$ and $\phi^\pi$ is similar to bounding the gap between the estimated and true value functions. Then, we can build a Bernstern-type uncertainty bound between $\hat{\phi}_{k-1}^\pi$ and $\phi^\pi$ using the variance of $G_h^{\pi;s',a'}$. For any policy $\pi$, $(s',a') \in \mathcal{S} \times \mathcal{A}$ and $k > 0$, define

$$
\begin{cases}
B_h^{\pi;s',a';k}(s) = & \min\Big\{ \Big(4\sqrt{\frac{\mathrm{Var}_{\hat{p}_{k-1}(\cdot|s,\pi_h(s))}(G_{h+1}^{\pi;s',a'}(\cdot|\hat{p}_{k-1})) \cdot L}{n_{k-1}(s,\pi_h(s))}} + \frac{13H^2 L}{n_{k-1}(s,\pi_h(s))} \\
& + \big(1 + \frac{2}{H}\big) \hat{p}_{k-1}(\cdot|s,\pi_h(s))^\top B_{h+1}^{\pi;s',a';k}(\cdot)\Big),\, H\Big\}, \quad \forall s \in \mathcal{S},\, \forall h \in [H], \\
B_{H+1}^{\pi;s',a';k}(s) = & \qquad\qquad 0, \qquad\qquad\qquad\qquad\qquad\qquad\qquad\qquad \forall s \in \mathcal{S}.
\end{cases}
\tag{31}
$$

The construction of $B_h^{\pi;s',a';k}(s)$ satisfies (see Lemma 29 for more details)

$$
|\hat{\phi}_{k-1}^\pi(s',a') - \phi^\pi(s',a')| \le \mathbb{E}_{s_1 \sim \rho}\Big[B_1^{\pi;s',a';k}(s_1)\Big], \quad \forall(s',a') \in \mathcal{S} \times \mathcal{A},
$$

$$
\|\hat{\phi}_{k-1}^\pi - \phi^\pi\|_1 \le \sum_{(s',a')} \mathbb{E}_{s_1 \sim \rho}\Big[B_1^{\pi;s',a';k}(s_1)\Big].
$$

Incorporating this transition uncertainty $\mathbb{E}_{s_1 \sim \rho}[B_1^{\pi;s',a';k}(s_1)]$ and reward uncertainty $\|\hat{\phi}_{k-1}^\pi\|_{(\Sigma_{k-1})^{-1}}$ into exploration bonuses, `LinUCB-Tran` computes the optimal policy $\pi^k$ under optimistic estimation (Line 4). After that, `LinUCB-Tran` plays episode $k$ with $\pi^k$, and collects trajectory $\tau^k$ and reward observation on each segment $\{R_i^k\}_{i=1}^m$ (Line 5).

### D.4 Proof for the Regret Upper Bound with Unknown Transition

In the following, we prove the regret upper bound (Theorem 6) of algorithm `LinUCB-Tran` for unknown transition.

Recall the definition of events $\mathcal{G}_{\mathrm{KL}}$ and $\mathcal{H}$ in Eqs. (18) and (19), respectively.

For any $k > 0$, define the set of state-action pairs

$$
D_k := \left\{ (s,a) \in \mathcal{S} \times \mathcal{A} : \frac{1}{4}\sum_{k'=1}^k w_{k'}(s,a) \ge H \log\left(\frac{|\mathcal{S}||\mathcal{A}|H}{\delta'}\right) + H \right\}.
\tag{32}
$$

$D_k$ stands for the set of state-action pairs which have sufficient visitations in expectation.

**Lemma 26.** *Assume that event $\mathcal{H}$ holds. Then, if $(s,a) \in D_k$,*

$$
n_{k-1}(s,a) \ge \frac{1}{4}\sum_{k'=1}^k w_{k'}(s,a).
$$

*Proof.* We have

$$
n_{k-1}(s,a) \ge \frac{1}{2}\sum_{k'=1}^{k-1} w_{k'}(s,a) - H \log\left(\frac{|\mathcal{S}||\mathcal{A}|H}{\delta'}\right)
$$

$$
= \frac{1}{4}\sum_{k'=1}^{k-1} w_{k'}(s,a) + \frac{1}{4}\sum_{k'=1}^{k-1} w_{k'}(s,a) - H \log\left(\frac{|\mathcal{S}||\mathcal{A}|H}{\delta'}\right)
$$

$$
= \frac{1}{4}\sum_{k'=1}^k w_{k'}(s,a) + \frac{1}{4}\sum_{k'=1}^k w_{k'}(s,a) - H \log\left(\frac{|\mathcal{S}||\mathcal{A}|H}{\delta'}\right) - \frac{1}{2}w_k(s,a)
$$

$$
\overset{(a)}{\ge} \frac{1}{4}\sum_{k'=1}^k w_{k'}(s,a) + H - \frac{1}{2}w_k(s,a)
$$

$$
\ge \frac{1}{4}\sum_{k'=1}^k w_{k'}(s,a),
$$

where (a) is due to the definition of $D_k$ (Eq. (32)). $\qquad\square$

**Lemma 27.** *It holds that*

$$\sum_{k=1}^{K}\sum_{h=1}^{H}\sum_{(s,a)\notin D_k} w_{k,h}(s,a) \le 8|\mathcal{S}||\mathcal{A}|H \log\left(\frac{|\mathcal{S}||\mathcal{A}|H}{\delta'}\right).$$

*Proof.* If $(s,a) \notin D_k$, then

$$\frac{1}{4}\sum_{k'=1}^{k} w_{k'}(s,a) < H \log\left(\frac{|\mathcal{S}||\mathcal{A}|H}{\delta'}\right) + H.$$

Thus, we have

$$\sum_{k=1}^{K}\sum_{h=1}^{H}\sum_{(s,a)\notin D_k} w_{k,h}(s,a) = \sum_{(s,a)}\sum_{k=1}^{K}\sum_{h=1}^{H}\mathbb{1}\{(s,a)\notin D_k\}\cdot w_{k,h}(s,a)$$

$$= \sum_{(s,a)}\sum_{k=1}^{K}\mathbb{1}\{(s,a)\notin D_k\}\cdot w_k(s,a)$$

$$\le 4|\mathcal{S}||\mathcal{A}|H \log\left(\frac{|\mathcal{S}||\mathcal{A}|H}{\delta'}\right) + 4|\mathcal{S}||\mathcal{A}|H$$

$$\le 8|\mathcal{S}||\mathcal{A}|H \log\left(\frac{|\mathcal{S}||\mathcal{A}|H}{\delta'}\right).$$

$\qquad\square$

**Lemma 28.** *Assume that event $\mathcal{H}$ holds. Then, we have*

$$\sum_{k=1}^{K}\sum_{h=1}^{H}\sum_{(s,a)\in D_k}\frac{w_{k,h}(s,a)}{n_{k-1}(s,a)} \le 4|\mathcal{S}||\mathcal{A}|\log(2KH).$$

*Proof.* It holds that

$$\sum_{k=1}^{K}\sum_{h=1}^{H}\sum_{(s,a)\in D_k}\frac{w_{k,h}(s,a)}{n_{k-1}(s,a)} = \sum_{k=1}^{K}\sum_{(s,a)\in D_k}\frac{w_k(s,a)}{n_{k-1}(s,a)}$$

$$= \sum_{k=1}^{K}\sum_{(s,a)}\frac{w_k(s,a)}{n_{k-1}(s,a)}\cdot\mathbb{1}\{(s,a)\in D_k\}$$

$$\overset{(a)}{\le} 4\sum_{k=1}^{K}\sum_{(s,a)}\frac{w_k(s,a)}{\sum_{k'=1}^{k} w_k(s,a)}\cdot\mathbb{1}\{(s,a)\in D_k\}$$

$$= 4\sum_{(s,a)}\sum_{k=1}^{K}\frac{w_k(s,a)}{\sum_{k'=1}^{k} w_k(s,a)}\cdot\mathbb{1}\{(s,a)\in D_k\}$$

$$\overset{(b)}{\le} 4|\mathcal{S}||\mathcal{A}|\log(2KH),$$

where (a) uses Lemma 26, and (b) follows from the analysis of Lemma 13 in (Zanette & Brunskill, 2019). $\qquad\square$

**Lemma 29** (Error in Visitation Vectors). *Assume that event $\mathcal{G}_{\mathrm{KL}}$ holds. Then, for any $k > 0$ and policy $\pi$,*

$$\|\hat{\phi}_{k-1}(\pi) - \phi(\pi)\|_1 \le \sum_{s',a'}\mathbb{E}_{s_1\sim\rho}\left[B_1^{\pi;s',a';k}(s_1)\right].$$

*Proof.* Since $\phi^\pi(s', a') = \mathbb{E}_{s_1 \sim \rho}[G_1^{\pi;s',a'}(s_1|p)]$ and $\hat{\phi}_{k-1}^\pi(s', a') = \mathbb{E}_{s_1 \sim \rho}[G_1^{\pi;s',a'}(s_1|\hat{p}_{k-1})]$, in this proof, we investigate the error in $G_h^{\pi;s',a'}$ due to the estimation of the transition model.

In the following, we prove by induction that for any $h \in [H]$ and $s \in \mathcal{S}$, $|G_h^{\pi;s',a'}(s|\hat{p}_{k-1}) - G_h^{\pi;s',a'}(s|p)| \le B_h^{\pi;s',a';k}(s)$.

When $h = H + 1$, by definition, we have $G_{H+1}^{\pi;s',a'}(s|\hat{p}_{k-1}) = G_{H+1}^{\pi;s',a'}(s|p) = B_{H+1}^{\pi;s',a';k}(s) = 0$ for any $s \in \mathcal{S}$, and then the above statement trivially holds.

When $1 \le h \le H$, if $|G_{h+1}^{\pi;s',a'}(\cdot|\hat{p}_{k-1}) - G_{h+1}^{\pi;s',a'}(\cdot|p)| \le B_{h+1}^{\pi;s',a';k}(\cdot)$ element-wise, then for any $s \in \mathcal{S}$, we have

$$
|G_h^{\pi;s',a'}(s|\hat{p}_{k-1}) - G_h^{\pi;s',a'}(s|p)|
$$

$$
= \left| \hat{p}_{k-1}(\cdot|s, \pi_h(s))^\top G_{h+1}^{\pi;s',a'}(\cdot|\hat{p}_{k-1}) - p(\cdot|s, \pi_h(s))^\top G_{h+1}^{\pi;s',a'}(\cdot|p) \right|
$$

$$
= \hat{p}_{k-1}(\cdot|s, \pi_h(s))^\top \left| G_{h+1}^{\pi;s',a'}(\cdot|\hat{p}_{k-1}) - G_{h+1}^{\pi;s',a'}(\cdot|p) \right|
$$

$$
+ \left| (\hat{p}_{k-1}(\cdot|s, \pi_h(s)) - p(\cdot|s, \pi_h(s)))^\top G_{h+1}^{\pi;s',a'}(\cdot|p) \right|
$$

$$
\overset{(a)}{\le} \hat{p}_{k-1}(\cdot|s, \pi_h(s))^\top \left| G_{h+1}^{\pi;s',a'}(\cdot|\hat{p}_{k-1}) - G_{h+1}^{\pi;s',a'}(\cdot|p) \right| + 2\sqrt{\frac{\mathrm{Var}_{p(\cdot|s, \pi_h(s))}(G_{h+1}^{\pi;s',a'}(\cdot|p)) \cdot L}{n_{k-1}(s, \pi_h(s))}}
$$

$$
+ \frac{HL}{n_{k-1}(s, \pi_h(s))}, \tag{33}
$$

where (a) is due to Lemma 37.

Here, we have

$$
\mathrm{Var}_{p(\cdot|s, \pi_h(s))}(G_{h+1}^{\pi;s',a'}(\cdot|p))
$$

$$
\overset{(a)}{\le} 2\mathrm{Var}_{\hat{p}_{k-1}(\cdot|s, \pi_h(s))}(G_{h+1}^{\pi;s',a'}(\cdot|p)) + \frac{4H^2 L}{n_{k-1}(s, \pi_h(s))}
$$

$$
\overset{(b)}{\le} 4\mathrm{Var}_{\hat{p}_{k-1}(\cdot|s, \pi_h(s))}(G_{h+1}^{\pi;s',a'}(\cdot|\hat{p}_{k-1})) + 4H\hat{p}_{k-1}(\cdot|s, \pi_h(s))^\top |G_{h+1}^{\pi;s',a'}(\cdot|\hat{p}_{k-1}) - G_{h+1}^{\pi;s',a'}(\cdot|p)|
$$

$$
+ \frac{4H^2 L}{n_{k-1}(s, \pi_h(s))},
$$

where (a) uses Lemma 38 and (b) comes from Lemma 39.

Then,

$$
\sqrt{\frac{\mathrm{Var}_{p(\cdot|s, \pi_h(s))}(G_{h+1}^{\pi;s',a'}(\cdot|p)) \cdot L}{n_{k-1}(s, \pi_h(s))}} \tag{34}
$$

$$
\le \sqrt{\frac{4\mathrm{Var}_{\hat{p}_{k-1}(\cdot|s, \pi_h(s))}(G_{h+1}^{\pi;s',a'}(\cdot|\hat{p}_{k-1})) \cdot L}{n_{k-1}(s, \pi_h(s))}}
$$

$$
+ \sqrt{\frac{1}{H}\hat{p}_{k-1}(\cdot|s, \pi_h(s))^\top |G_{h+1}^{\pi;s',a'}(\cdot|\hat{p}_{k-1}) - G_{h+1}^{\pi;s',a'}(\cdot|p)| \cdot \frac{4H^2 L}{n_{k-1}(s, \pi_h(s))}} + \frac{2HL}{n_{k-1}(s, \pi_h(s))}
$$

$$
\overset{(a)}{\le} 2\sqrt{\frac{\mathrm{Var}_{\hat{p}_{k-1}(\cdot|s, \pi_h(s))}(G_{h+1}^{\pi;s',a'}(\cdot|\hat{p}_{k-1})) \cdot L}{n_{k-1}(s, \pi_h(s))}}
$$

$$
+ \frac{1}{H}\hat{p}_{k-1}(\cdot|s, \pi_h(s))^\top |G_{h+1}^{\pi;s',a'}(\cdot|\hat{p}_{k-1}) - G_{h+1}^{\pi;s',a'}(\cdot|p)| + \frac{6H^2 L}{n_{k-1}(s, \pi_h(s))}, \tag{35}
$$

where (a) is due to the fact that $\sqrt{xy} \le x + y$.

Hence, plugging Eq. (35) into Eq. (33) and using the fact that $|G_h^{\pi;s',a'}(s)| \in [0, H]$, we have

$$|G_h^{\pi;s',a'}(s|\hat{p}_{k-1}) - G_h^{\pi;s',a'}(s|p)|$$

$$\leq \left( 4\sqrt{\frac{\text{Var}_{\hat{p}_{k-1}(\cdot|s,\pi_h(s))}(G_{h+1}^{\pi;s',a'}(\cdot|\hat{p}_{k-1})) \cdot L}{n_{k-1}(s,\pi_h(s))}} + \frac{13H^2 L}{n_{k-1}(s,\pi_h(s))} \right.$$

$$\left. + \left( 1 + \frac{2}{H} \right) \hat{p}_{k-1}(\cdot|s,\pi_h(s))^\top \left| G_{h+1}^{\pi;s',a'}(\cdot|\hat{p}_{k-1}) - G_{h+1}^{\pi;s',a'}(\cdot|p) \right| \right) \wedge H.$$

$$\leq \left( 4\sqrt{\frac{\text{Var}_{\hat{p}_{k-1}(\cdot|s,\pi_h(s))}(G_{h+1}^{\pi;s',a'}(\cdot|\hat{p}_{k-1})) \cdot L}{n_{k-1}(s,\pi_h(s))}} + \frac{13H^2 L}{n_{k-1}(s,\pi_h(s))} \right.$$

$$\left. + \left( 1 + \frac{2}{H} \right) \hat{p}_{k-1}(\cdot|s,\pi_h(s))^\top B_{h+1}^{\pi;s',a';k}(\cdot) \right) \wedge H$$

$$= B_h^{\pi;s',a';k}(s),$$

which completes the induction proof.

Therefore,

$$\left| \hat{\phi}_{k-1}^\pi(s',a') - \phi^\pi(s',a') \right| = \left| \mathbb{E}_{s_1 \sim \rho} \left[ G_1^{\pi;s',a'}(s_1|\hat{p}_{k-1}) \right] - \mathbb{E}_{s_1 \sim \rho} \left[ G_1^{\pi;s',a'}(s_1|p) \right] \right|$$

$$\leq \mathbb{E}_{s_1 \sim \rho} \left[ \left| G_1^{\pi;s',a'}(s_1|\hat{p}_{k-1}) - G_1^{\pi;s',a'}(s_1|p) \right| \right]$$

$$\leq \mathbb{E}_{s_1 \sim \rho} \left[ B_1^{\pi;s',a';k}(s_1) \right].$$

Summing over $(s',a') \in \mathcal{S} \times \mathcal{A}$, we obtain this lemma. $\qquad\square$

**Lemma 30.** *Assume that event $\mathcal{G}_{\text{KL}} \cap \mathcal{H}$ holds. Then, for any $k > 0$ and policy $\pi$,*

$$\mathbb{E}_{s_1 \sim \rho} \left[ B_1^{\pi;s',a';k}(s_1) \right]$$

$$\leq e^{12} \sum_{h=1}^H \sum_{s,a} w_h^\pi(s,a) \left( 8\sqrt{\frac{\text{Var}_{p(\cdot|s,a)}(G_{h+1}^{\pi;s',a'}(\cdot|p)) \cdot L}{n_{k-1}(s,a)}} + \frac{46H^2 L}{n_{k-1}(s,a)} \right) \wedge H,$$

*and*

$$\sum_{s',a'} \sum_{k=1}^K \mathbb{E}_{s_1 \sim \rho} \left[ B_1^{\pi^k;s',a';k}(s_1) \right]$$

$$\leq 16 e^{12} |\mathcal{S}|^{\frac{3}{2}} |\mathcal{A}|^{\frac{3}{2}} H \sqrt{KL \log(2KH)} + 192 e^{12} |\mathcal{S}|^2 |\mathcal{A}|^2 H^2 L \log(2KH).$$

*Proof.* First, we prove the first statement.

For any policy $\pi$, $k > 0$, $(s',a') \in \mathcal{S} \times \mathcal{A}$, $h \in [H]$ and $s \in \mathcal{S}$, we have

$$B_h^{\pi;s',a';k}(s) \leq 4\sqrt{\frac{\text{Var}_{\hat{p}_{k-1}(\cdot|s,\pi_h(s))}(G_{h+1}^{\pi;s',a'}(\cdot|\hat{p}_{k-1})) \cdot L}{n_{k-1}(s,\pi_h(s))}} + \frac{13H^2 L}{n_{k-1}(s,\pi_h(s))}$$

$$+ \left( 1 + \frac{2}{H} \right) \hat{p}_{k-1}(\cdot|s,\pi_h(s))^\top B_{h+1}^{\pi;s',a';k}(\cdot)$$

$$= 4\sqrt{\frac{\text{Var}_{\hat{p}_{k-1}(\cdot|s,\pi_h(s))}(G_{h+1}^{\pi;s',a'}(\cdot|\hat{p}_{k-1})) \cdot L}{n_{k-1}(s,\pi_h(s))}} + \frac{13H^2 L}{n_{k-1}(s,\pi_h(s))}$$

$$+ \left( 1 + \frac{2}{H} \right) p(\cdot|s,\pi_h(s))^\top B_{h+1}^{\pi;s',a';k}(\cdot)$$

$$+ \left( 1 + \frac{2}{H} \right) (\hat{p}_{k-1}(\cdot|s,\pi_h(s)) - p(\cdot|s,\pi_h(s)))^\top B_{h+1}^{\pi;s',a';k}(\cdot)$$

$$\overset{(a)}{\leq} 4\sqrt{\frac{\text{Var}_{\hat{p}_{k-1}(\cdot|s,\pi_h(s))}(G_{h+1}^{\pi;s',a'}(\cdot|\hat{p}_{k-1})) \cdot L}{n_{k-1}(s,\pi_h(s))}} + \frac{13H^2L}{n_{k-1}(s,\pi_h(s))}$$

$$+ \left(1 + \frac{2}{H}\right) p(\cdot|s,\pi_h(s))^\top B_{h+1}^{\pi;s',a';k}(\cdot)$$

$$+ \left(1 + \frac{2}{H}\right) \cdot \left(2\sqrt{\frac{\text{Var}_{p(\cdot|s,\pi_h(s))}(B_{h+1}^{\pi;s',a';k}(\cdot)) \cdot L}{n_{k-1}(s,\pi_h(s))}} + \frac{HL}{n_{k-1}(s,\pi_h(s))}\right)$$

$$\leq 4\sqrt{\frac{\text{Var}_{\hat{p}_{k-1}(\cdot|s,\pi_h(s))}(G_{h+1}^{\pi;s',a'}(\cdot|\hat{p}_{k-1})) \cdot L}{n_{k-1}(s,\pi_h(s))}} + \frac{13H^2L}{n_{k-1}(s,\pi_h(s))}$$

$$+ \left(1 + \frac{2}{H}\right) p(\cdot|s,\pi_h(s))^\top B_{h+1}^{\pi;s',a';k}(\cdot)$$

$$+ \left(1 + \frac{2}{H}\right) \left(2\sqrt{\frac{1}{H} p(\cdot|s,\pi_h(s))^\top B_{h+1}^{\pi;s',a';k}(\cdot) \frac{H^2L}{n_{k-1}(s,\pi_h(s))}} + \frac{HL}{n_{k-1}(s,\pi_h(s))}\right)$$

$$\overset{(b)}{\leq} 4\sqrt{\frac{\text{Var}_{\hat{p}_{k-1}(\cdot|s,\pi_h(s))}(G_{h+1}^{\pi;s',a'}(\cdot|\hat{p}_{k-1})) \cdot L}{n_{k-1}(s,\pi_h(s))}} + \frac{22H^2L}{n_{k-1}(s,\pi_h(s))}$$

$$+ \left(1 + \frac{8}{H}\right) p(\cdot|s,\pi_h(s))^\top B_{h+1}^{\pi;s',a';k}(\cdot), \tag{36}$$

where (a) uses Lemma 37, and (b) follows from the fact that $\sqrt{xy} \leq x + y$.

In addition, we have

$$\text{Var}_{\hat{p}_{k-1}(\cdot|s,\pi_h(s))}(G_{h+1}^{\pi;s',a'}(\cdot|\hat{p}_{k-1}))$$

$$\overset{(a)}{=} 2\text{Var}_{p(\cdot|s,\pi_h(s))}(G_{h+1}^{\pi;s',a'}(\cdot|\hat{p}_{k-1})) + \frac{4H^2L}{n_{k-1}(s,a)}$$

$$\overset{(b)}{\leq} 4\text{Var}_{p(\cdot|s,\pi_h(s))}(G_{h+1}^{\pi;s',a'}(\cdot|p)) + 4H p(\cdot|s,\pi_h(s))^\top \left|G_{h+1}^{\pi;s',a'}(\cdot|\hat{p}_{k-1}) - G_{h+1}^{\pi;s',a'}(\cdot|p)\right|$$

$$+ \frac{4H^2L}{n_{k-1}(s,a)}$$

$$\leq 4\text{Var}_{p(\cdot|s,\pi_h(s))}(G_{h+1}^{\pi;s',a'}(\cdot|p)) + 4H p(\cdot|s,\pi_h(s))^\top B_{h+1}^{\pi;s',a'}(\cdot|\hat{p}_{k-1}) + \frac{4H^2L}{n_{k-1}(s,a)},$$

where (a) uses Lemma 38, and (b) comes from Lemma 39.

Then,

$$\sqrt{\frac{\text{Var}_{\hat{p}_{k-1}(\cdot|s,\pi_h(s))}(G_{h+1}^{\pi;s',a'}(\cdot|\hat{p}_{k-1})) \cdot L}{n_{k-1}(s,\pi_h(s))}}$$

$$\leq \sqrt{\frac{4\text{Var}_{p(\cdot|s,\pi_h(s))}(G_{h+1}^{\pi;s',a'}(\cdot|p)) \cdot L}{n_{k-1}(s,\pi_h(s))}} + \sqrt{\frac{1}{H} p(\cdot|s,\pi_h(s))^\top B_{h+1}^{\pi;s',a'}(\cdot|\hat{p}_{k-1}) \cdot \frac{4H^2L}{n_{k-1}(s,\pi_h(s))}}$$

$$+ \frac{2HL}{n_{k-1}(s,a)}$$

$$\leq 2\sqrt{\frac{\text{Var}_{p(\cdot|s,\pi_h(s))}(G_{h+1}^{\pi;s',a'}(\cdot|p)) \cdot L}{n_{k-1}(s,\pi_h(s))}} + \frac{1}{H} p(\cdot|s,\pi_h(s))^\top B_{h+1}^{\pi;s',a'}(\cdot|\hat{p}_{k-1})$$

$$+ \frac{6H^2L}{n_{k-1}(s,\pi_h(s))} \tag{37}$$

Plugging Eq. (37) into Eq. (36) and using the clipping definition of $B_h^{\pi;s',a';k}(s)$, we have

$$B_h^{\pi;s',a';k}(s) \leq \left( 8\sqrt{\frac{\mathrm{Var}_{p(\cdot|s,\pi_h(s))}(G_{h+1}^{\pi;s',a'}(\cdot|p)) \cdot L}{n_{k-1}(s,\pi_h(s))}} + \frac{46H^2L}{n_{k-1}(s,\pi_h(s))} \right) \wedge H$$

$$+ \left( 1 + \frac{12}{H} \right) p(\cdot|s,\pi_h(s))^\top B_{h+1}^{\pi;s',a';k}(\cdot)$$

Using the above inequality, taking $s_1 \sim \rho$, and unfolding $B_1^{\pi;s',a';k}(s_1)$ over $h$, we have

$$\mathbb{E}_{s_1 \sim \rho}\left[ B_1^{\pi;s',a';k}(s_1) \right]$$

$$\leq e^{12} \sum_{h=1}^{H} \sum_{s,a} w_h^\pi(s,a) \left( 8\sqrt{\frac{\mathrm{Var}_{p(\cdot|s,a)}(G_{h+1}^{\pi;s',a'}(\cdot|p)) \cdot L}{n_{k-1}(s,a)}} + \frac{46H^2L}{n_{k-1}(s,a)} \right) \wedge H. \qquad (38)$$

Next, we prove the second statement.

It holds that

$$\sum_{s',a'} \sum_{k=1}^{K} \mathbb{E}_{s_1 \sim \rho}\left[ B_1^{\pi^k;s',a';k}(s_1) \right]$$

$$\leq e^{12} \sum_{s',a'} \sum_{k=1}^{K} \sum_{h=1}^{H} \sum_{(s,a) \in D_k} w_{k,h}(s,a) \left( 8\sqrt{\frac{\mathrm{Var}_{p(\cdot|s,a)}(G_{h+1}^{\pi^k;s',a'}(\cdot|p)) \cdot L}{n_{k-1}(s,a)}} + \frac{46H^2L}{n_{k-1}(s,a)} \right)$$

$$+ e^{12}H|\mathcal{S}||\mathcal{A}| \sum_{k=1}^{K} \sum_{h=1}^{H} \sum_{(s,a) \notin D_k} w_{k,h}(s,a)$$

$$\overset{(a)}{\leq} 8e^{12}\sqrt{L} \sum_{s',a'} \sqrt{\sum_{k=1}^{K} \sum_{h=1}^{H} \sum_{(s,a) \in D_k} w_{k,h}(s,a) \mathrm{Var}_{p(\cdot|s,a)}(G_{h+1}^{\pi^k;s',a'}(\cdot|p))} \cdot$$

$$\sqrt{\sum_{k=1}^{K} \sum_{h=1}^{H} \sum_{(s,a) \in D_k} \frac{w_{k,h}(s,a)}{n_{k-1}(s,a)}} + e^{12}|\mathcal{S}||\mathcal{A}| \cdot 46H^2L \sum_{k=1}^{K} \sum_{h=1}^{H} \sum_{(s,a) \in D_k} \frac{w_{k,h}(s,a)}{n_{k-1}(s,a)}$$

$$+ 8e^{12}|\mathcal{S}|^2|\mathcal{A}|^2H^2 \log\left( \frac{|\mathcal{S}||\mathcal{A}|H}{\delta'} \right)$$

$$\overset{(b)}{\leq} 8e^{12}|\mathcal{S}||\mathcal{A}|\sqrt{L}\sqrt{KH^2} \cdot \sqrt{4|\mathcal{S}||\mathcal{A}| \log(2KH)} + 184e^{12}|\mathcal{S}|^2|\mathcal{A}|^2H^2L \log(2KH)$$

$$+ 8e^{12}|\mathcal{S}|^2|\mathcal{A}|^2H^2 \log\left( \frac{|\mathcal{S}||\mathcal{A}|H}{\delta'} \right)$$

$$\leq 16e^{12}|\mathcal{S}|^{\frac{3}{2}}|\mathcal{A}|^{\frac{3}{2}}H\sqrt{KL \log(2KH)} + 192e^{12}|\mathcal{S}|^2|\mathcal{A}|^2H^2L \log(2KH),$$

where (a) is due to Lemma 27, and (b) follows from Lemmas 36 and 28. $\qquad \square$

**Lemma 31** (Optimism under Sum Feedback and Unknown Transition). *Assume that event $\mathcal{G}_{\mathrm{KL}}$ holds. Then, for any $k > 0$ and fixed policy $\pi$,*

$$V_1^\pi(s_1) \leq \hat{\phi}_{k-1}(\pi)^\top \hat{\theta}_{k-1} + \beta(k-1) \cdot \|\hat{\phi}_{k-1}(\pi)\|_{(\Sigma_{k-1})^{-1}} + r_{\max} \sum_{s',a'} \mathbb{E}_{s_1 \sim \rho}\left[ B_1^{\pi;s',a';k}(s_1) \right].$$

*Proof.* It holds that

$$V_1^\pi(s_1) = \phi(\pi)^\top \theta$$

$$= \hat{\phi}_{k-1}(\pi)^\top \hat{\theta}_{k-1} + \phi(\pi)^\top \theta - \hat{\phi}_{k-1}(\pi)^\top \theta + \hat{\phi}_{k-1}(\pi)^\top \theta - \hat{\phi}_{k-1}(\pi)^\top \hat{\theta}_{k-1}$$

$$\leq \hat{\phi}_{k-1}(\pi)^\top \hat{\theta}_{k-1} + \|\phi(\pi) - \hat{\phi}_{k-1}(\pi)\|_1 \cdot \|\theta\|_\infty + \beta(k-1) \cdot \|\hat{\phi}_{k-1}(\pi)\|_{(\Sigma_{k-1})^{-1}}$$

$$\overset{\text{(a)}}{\leq} \hat{\phi}_{k-1}(\pi)^\top \hat{\theta}_{k-1} + \beta(k-1) \cdot \|\hat{\phi}_{k-1}(\pi)\|_{(\Sigma_{k-1})^{-1}} + r_{\max} \sum_{s',a'} \mathbb{E}_{s_1 \sim \rho} \left[ B_1^{\pi; s', a'; k}(s_1) \right],$$

where (a) uses Lemma 29. $\qquad\square$

**Lemma 32.** *For any $K \geq 1$, we have*

$$\sum_{k=1}^K \sum_{i=1}^m \left\| \phi^{\tau_i^k} \right\|_{(\Sigma_{k-1})^{-1}} \leq H \sqrt{\frac{2K|\mathcal{S}||\mathcal{A}|}{\lambda} \log\left(1 + \frac{KH^2}{\lambda |\mathcal{S}||\mathcal{A}|m}\right)}.$$

*Proof.* We have

$$\sum_{k=1}^K \sum_{i=1}^m \left\| \phi^{\tau_i^k} \right\|_{(\Sigma_{k-1})^{-1}} \leq \sqrt{Km \sum_{k=1}^K \sum_{i=1}^m \left\| \phi^{\tau_i^k} \right\|_{(\Sigma_{k-1})^{-1}}^2}$$

$$= \sqrt{Km \sum_{k=1}^K \min\left\{ \sum_{i=1}^m \left\| \phi^{\tau_i^k} \right\|_{(\Sigma_{k-1})^{-1}}^2, \frac{H^2}{m\lambda} \right\}}$$

$$= \sqrt{\frac{H^2 K}{\lambda} \sum_{k=1}^K \min\left\{ \frac{m\lambda}{H^2} \sum_{i=1}^m \left\| \phi^{\tau_i^k} \right\|_{(\Sigma_{k-1})^{-1}}^2, 1 \right\}}$$

$$\overset{\text{(a)}}{\leq} \sqrt{\frac{2H^2 K}{\lambda} \sum_{k=1}^K \log\left(1 + \min\left\{ \frac{m\lambda}{H^2} \sum_{i=1}^m \left\| \phi^{\tau_i^k} \right\|_{(\Sigma_{k-1})^{-1}}^2, 1 \right\}\right)}$$

$$\overset{\text{(b)}}{\leq} \sqrt{\frac{2H^2 K}{\lambda} \sum_{k=1}^K \log\left(1 + \sum_{i=1}^m \left\| \phi^{\tau_i^k} \right\|_{(\Sigma_{k-1})^{-1}}^2 \right)}$$

$$\overset{\text{(c)}}{\leq} \sqrt{\frac{2KH^2|\mathcal{S}||\mathcal{A}|}{\lambda} \log\left(1 + \frac{KH^2}{\lambda |\mathcal{S}||\mathcal{A}|m}\right)},$$

where inequality (a) uses the fact that $x \leq 2\log(1+x)$ for any $0 \leq x \leq 1$, inequality (b) is due to the fact that $\lambda \leq \frac{H^2}{m}$, and inequality (c) follows from Lemma 22. $\qquad\square$

Define event

$$\mathcal{F}_{\text{reg}}^{\text{S}} := \left\{ \left| \sum_{k'=1}^k \left( \mathbb{E}_{\tau \sim \pi^{k'}} \left[ \|\phi^\tau\|_{(\Sigma_{k'-1})^{-1}} \mid F_{k'-1} \right] - \|\phi^\tau\|_{(\Sigma_{k'-1})^{-1}} \right) \right| \leq 4H \sqrt{\frac{k}{\lambda} \log\left(\frac{4k}{\delta'}\right)}, \right.$$

$$\left. \forall k > 0 \right\}. \tag{39}$$

Event $\mathcal{F}_{\text{reg}}^{\text{S}}$ is similar to $\mathcal{F}_{\text{opt}}^{\text{S}}$, except that here the universal upper bound of $\|\phi^\tau\|_{(\Sigma_{k'-1})^{-1}}$ is $\frac{H}{\sqrt{\lambda}}$ rather than 1.

**Lemma 33.** *It holds that*

$$\Pr\left[ \mathcal{F}_{\text{reg}}^{\text{S}} \right] \geq 1 - \delta'.$$

*Proof.* For any $k' \geq 1$, we have that $\|\phi^\tau\|_{(\Sigma_{k'-1})^{-1}} \leq \frac{H}{\sqrt{\lambda}}$, and then $\left| \mathbb{E}_{\tau \sim \pi^{k'}} [\|\phi^\tau\|_{(\Sigma_{k'-1})^{-1}} \mid F_{k'-1}] - \|\phi^\tau\|_{(\Sigma_{k'-1})^{-1}} \right| \leq \frac{2H}{\sqrt{\lambda}}$.

Using the Azuma-Hoeffding inequality, we have that for any fixed $k > 0$, with probability at least $1 - \frac{\delta'}{2k^2}$,

$$\left| \sum_{k'=1}^k \left( \mathbb{E}_{\tau \sim \pi^{k'}} \left[ \|\phi^\tau\|_{(\Sigma_{k'-1})^{-1}} \mid F_{k'-1} \right] - \|\phi^\tau\|_{(\Sigma_{k'-1})^{-1}} \right) \right| \leq \sqrt{2 \cdot \frac{4H^2}{\lambda} \cdot k \log\left(\frac{4k^2}{\delta'}\right)}.$$

Since $\sum_{k=1}^{\infty} \frac{\delta'}{2k^2} \le \delta'$, by a union bound over $k$, we have that with probability at least $\delta'$, for any $k \ge 1$,

$$\left| \sum_{k'=1}^{k} \left( \mathbb{E}_{\tau \sim \pi^{k'}} \left[ \|\phi^\tau\|_{(\Sigma_{k'-1})^{-1}} |F_{k'-1} \right] - \|\phi^\tau\|_{(\Sigma_{k'-1})^{-1}} \right) \right| \le \sqrt{2 \cdot \frac{4H^2}{\lambda} \cdot k \log\left( \frac{4k^2}{\delta'} \right)}$$

$$\le 4H \sqrt{\frac{k}{\lambda} \log\left( \frac{4k}{\delta'} \right)}.$$

$\square$

*Proof of Theorem 6.* Let $\delta' = \frac{\delta}{4}$. Then, we have $\Pr[\mathcal{K} \cap \mathcal{F}_{\text{reg}}^{\text{S}} \cap \mathcal{G}_{\text{KL}} \cap \mathcal{H}] \ge 1 - \delta$. Thus, it suffices to prove the regret upper bound when event $\mathcal{K} \cap \mathcal{F}_{\text{reg}}^{\text{S}} \cap \mathcal{G}_{\text{KL}} \cap \mathcal{H}$ holds.

Assume that event $\mathcal{K} \cap \mathcal{F}_{\text{reg}}^{\text{S}} \cap \mathcal{G}_{\text{KL}} \cap \mathcal{H}$ holds. For any $k > 0$, we have

$$\sum_{k=1}^{K} \left( V^*(s_1) - V^{\pi^k}(s_1) \right)$$

$$\overset{(a)}{\le} \sum_{k=1}^{K} \left( \hat{\phi}_{k-1}(\pi^*)^\top \hat{\theta}_{k-1} + \beta(k-1) \cdot \|\hat{\phi}_{k-1}(\pi^*)\|_{(\Sigma_{k-1})^{-1}} + r_{\max} \sum_{s',a'} \mathbb{E}_{s_1 \sim \rho} \left[ B_1^{\pi^*;s',a';k}(s_1) \right] \right.$$

$$\left. - V^{\pi^k} \right)$$

$$\overset{(b)}{\le} \sum_{k=1}^{K} \left( \hat{\phi}_{k-1}(\pi^k)^\top \hat{\theta}_{k-1} + \beta(k-1) \cdot \|\hat{\phi}_{k-1}(\pi^k)\|_{(\Sigma_{k-1})^{-1}} + r_{\max} \sum_{s',a'} \mathbb{E}_{s_1 \sim \rho} \left[ B_1^{\pi^k;s',a';k}(s_1) \right] \right.$$

$$\left. - V^{\pi^k} \right)$$

$$\le \sum_{k=1}^{K} \left( \hat{\phi}_{k-1}(\pi^k)^\top \hat{\theta}_{k-1} - \hat{\phi}_{k-1}(\pi^k)^\top \theta + \hat{\phi}_{k-1}(\pi^k)^\top \theta - (\phi^{\pi^k})^\top \theta \right.$$

$$\left. + \beta(k-1) \cdot \|\hat{\phi}_{k-1}(\pi^k)\|_{(\Sigma_{k-1})^{-1}} + r_{\max} \sum_{s',a'} \mathbb{E}_{s_1 \sim \rho} \left[ B_1^{\pi^k;s',a';k}(s_1) \right] \right)$$

$$\overset{(c)}{\le} \sum_{k=1}^{K} \left( 2\beta(k-1) \cdot \|\hat{\phi}_{k-1}(\pi^k)\|_{(\Sigma_{k-1})^{-1}} + 2r_{\max} \sum_{s',a'} \mathbb{E}_{s_1 \sim \rho} \left[ B_1^{\pi^k;s',a';k}(s_1) \right] \right)$$

$$\le 2\beta(K) \sum_{k=1}^{K} \|\hat{\phi}_{k-1}(\pi^k)\|_{(\Sigma_{k-1})^{-1}} + 2r_{\max} \sum_{k=1}^{K} \sum_{s',a'} \mathbb{E}_{s_1 \sim \rho} \left[ B_1^{\pi^k;s',a';k}(s_1) \right], \tag{40}$$

where (a) uses Lemma 31, (b) is due to the definition of $\pi^k$, and (c) follows from Lemma 29 and the definition of event $\mathcal{K}$.

Next, we first bound $\sum_{k=1}^{K} \|\hat{\phi}_{k-1}(\pi^k)\|_{(\Sigma_{k-1})^{-1}}$.

We have

$$\sum_{k=1}^{K} \|\hat{\phi}_{k-1}(\pi^k)\|_{(\Sigma_{k-1})^{-1}} \le \sum_{k=1}^{K} \left( \|\phi^{\pi^k}\|_{(\Sigma_{k-1})^{-1}} + \|\hat{\phi}_{k-1}(\pi^k) - \phi^{\pi^k}\|_{(\Sigma_{k-1})^{-1}} \right)$$

$$\le \sum_{k=1}^{K} \left( \|\phi^{\pi^k}\|_{(\Sigma_{k-1})^{-1}} + \frac{1}{\sqrt{\lambda}} \cdot \|\hat{\phi}_{k-1}(\pi^k) - \phi^{\pi^k}\|_2 \right)$$

$$\leq \sum_{k=1}^{K} \left( \|\phi^{\pi^k}\|_{(\Sigma_{k-1})^{-1}} + \frac{1}{\sqrt{\lambda}} \cdot \|\hat{\phi}_{k-1}(\pi^k) - \phi^{\pi^k}\|_1 \right). \quad (41)$$

Here we have

$$\sum_{k=1}^{K} \|\phi^{\pi^k}\|_{(\Sigma_{k-1})^{-1}}$$

$$= \sum_{k=1}^{K} \|\mathbb{E}_{\tau \sim \pi^k}[\phi^\tau | F_{k-1}]\|_{(\Sigma_{k-1})^{-1}}$$

$$\overset{(a)}{\leq} \sum_{k=1}^{K} \mathbb{E}_{\tau \sim \pi^k}\left[\|\phi^\tau\|_{(\Sigma_{k-1})^{-1}} | F_{k-1}\right]$$

$$= \sum_{k=1}^{K} \left( \mathbb{E}_{\tau \sim \pi^k}\left[\|\phi^\tau\|_{(\Sigma_{k-1})^{-1}} | F_{k-1}\right] - \|\phi(\tau^k)\|_{(\Sigma_{k-1})^{-1}} + \|\phi(\tau^k)\|_{(\Sigma_{k-1})^{-1}} \right)$$

$$\leq \sum_{k=1}^{K} \left( \mathbb{E}_{\tau \sim \pi^k}\left[\|\phi^\tau\|_{(\Sigma_{k-1})^{-1}} | F_{k-1}\right] - \|\phi(\tau^k)\|_{(\Sigma_{k-1})^{-1}} + \sum_{i=1}^{m} \|\phi^{\tau_i^k}\|_{(\Sigma_{k-1})^{-1}} \right)$$

$$\overset{(b)}{\leq} 4H\sqrt{\frac{K}{\lambda}\log\left(\frac{4K}{\delta'}\right)} + H\sqrt{\frac{2K|\mathcal{S}||\mathcal{A}|}{\lambda}\log\left(1 + \frac{KH^2}{\lambda|\mathcal{S}||\mathcal{A}|m}\right)}, \quad (42)$$

where (a) uses the Jensen inequality, and (b) comes from the definition of $\mathcal{F}_{\text{reg}}^{\text{S}}$ and Lemma 32.

Hence, plugging Eq. (42) into Eq. (41) and using Lemma 29, we have

$$\sum_{k=1}^{K} \|\hat{\phi}_{k-1}(\pi^k)\|_{(\Sigma_{k-1})^{-1}} \leq 4H\sqrt{\frac{K}{\lambda}\log\left(\frac{4K}{\delta'}\right)} + H\sqrt{\frac{2K|\mathcal{S}||\mathcal{A}|}{\lambda}\log\left(1 + \frac{KH^2}{\lambda|\mathcal{S}||\mathcal{A}|}\right)}$$

$$+ \frac{1}{\sqrt{\lambda}} \sum_{k=1}^{K} \sum_{s',a'} \mathbb{E}_{s_1 \sim \rho}\left[B_1^{\pi;s',a';k}(s_1)\right]. \quad (43)$$

On the other hand, according to Eq. (38), we have

$$\mathbb{E}_{s_1 \sim \rho}\left[B_1^{\pi;s',a';k}(s_1)\right] \leq e^{12} \sum_{h=1}^{H} \sum_{s,a} w_h^\pi(s,a) \left( 8\sqrt{\frac{\text{Var}_{p(\cdot|s,a)}(G_{h+1}^{\pi;s',a'}(\cdot|p)) \cdot L}{n_{k-1}(s,a)}} + \frac{46H^2 L}{n_{k-1}(s,a)} \right) \wedge H.$$

Therefore, plugging Eqs. (43) and (38) into Eq. (40), we have

$$\sum_{k=1}^{K} \left(V^* - V^{\pi^k}\right)$$

$$\leq 2\beta(K) \left( 4H\sqrt{\frac{K}{\lambda}\log\left(\frac{4K}{\delta'}\right)} + H\sqrt{\frac{2K|\mathcal{S}||\mathcal{A}|}{\lambda}\log\left(1 + \frac{KH^2}{\lambda|\mathcal{S}||\mathcal{A}|m}\right)} \right)$$

$$+ 2\left(\frac{\beta(K)}{\sqrt{\lambda}} + r_{\max}\right) \sum_{s',a'} \sum_{k=1}^{K} \mathbb{E}_{s_1 \sim \rho}\left[B_1^{\pi^k;s',a';k}(s_1)\right]$$

$$\overset{(a)}{\leq} 2\beta(K) \left( 4H\sqrt{\frac{K}{\lambda}\log\left(\frac{4K}{\delta'}\right)} + H\sqrt{\frac{2K|\mathcal{S}||\mathcal{A}|}{\lambda}\log\left(1 + \frac{KH^2}{\lambda|\mathcal{S}||\mathcal{A}|m}\right)} \right)$$

$$+ \frac{4\beta(K)}{\sqrt{\lambda}} \left( 16e^{12}|\mathcal{S}|^{\frac{3}{2}}|\mathcal{A}|^{\frac{3}{2}}H\sqrt{KL\log(2KH)} + 192e^{12}|\mathcal{S}|^2|\mathcal{A}|^2 H^2 L \log(2KH) \right)$$

$r(s_i, \cdot) = 0$

Optimal action: transition to $s_{n+1}$ w.p. $\frac{1}{2} + \varepsilon$

$s_{n+2}$ w.p. $\frac{1}{2} - \varepsilon$

w.p. $\frac{1}{n}$ $s_1$

$s_{n+1}$ $r(s_{n+1}, \cdot) = r_{max}$

w.p. $\frac{1}{n}$ $s_2$

$s_{n+2}$ $r(s_{n+2}, \cdot) = 0$

w.p. $\frac{1}{n}$ $s_n$

Suboptimal action: transition to $s_{n+1}$ w.p. $\frac{1}{2}$

$s_{n+2}$ w.p. $\frac{1}{2}$

Figure 6: Instance for the lower bound under sum segment feedback and unknown transition.

$$
= O\left(\left(\sqrt{\frac{H|\mathcal{S}||\mathcal{A}|}{m}\log\left(\left(1 + \frac{KH^2}{\lambda|\mathcal{S}||\mathcal{A}|m}\right)\frac{1}{\delta}\right)} + r_{\max}\sqrt{\lambda|\mathcal{S}||\mathcal{A}|}\right) \cdot \right.
$$

$$
\left(H\sqrt{\frac{K|\mathcal{S}||\mathcal{A}|}{\lambda}\log\left(\left(1 + \frac{KH^2}{\lambda|\mathcal{S}||\mathcal{A}|m}\right)\frac{1}{\delta}\right)} + |\mathcal{S}|^{\frac{3}{2}}|\mathcal{A}|^{\frac{3}{2}}H\sqrt{\frac{KL}{\lambda}\log(KH)}\right.
$$

$$
\left. + \frac{|\mathcal{S}|^2|\mathcal{A}|^2H^2L}{\sqrt{\lambda}}\log(KH)\right)\right)
$$

$$
\overset{\text{(b)}}{=} O\left((1 + r_{\max})|\mathcal{S}|^2|\mathcal{A}|^2H\sqrt{K}\left(\log\left(\left(1 + \frac{KH}{|\mathcal{S}||\mathcal{A}|}\right)\frac{1}{\delta}\right)\right.\right.
$$

$$
+ \sqrt{L\log(KH)}\sqrt{\log\left(\left(1 + \frac{KH}{|\mathcal{S}||\mathcal{A}|}\right)\frac{1}{\delta}\right)}\right)
$$

$$
\left.+ (1 + r_{\max})|\mathcal{S}|^{\frac{5}{2}}|\mathcal{A}|^{\frac{5}{2}}H^2L\log(KH)\sqrt{\log\left(\left(1 + \frac{KH}{|\mathcal{S}||\mathcal{A}|}\right)\frac{1}{\delta}\right)}\right)
$$

$$
= \tilde{O}\left((1 + r_{\max})|\mathcal{S}|^{\frac{5}{2}}|\mathcal{A}|^2H\sqrt{K} + (1 + r_{\max})|\mathcal{S}|^{\frac{7}{2}}|\mathcal{A}|^{\frac{5}{2}}H^2\right),
$$

where (a) comes from Lemma 30, and (b) uses the fact that $\lambda := \frac{H}{m}$. $\qquad \square$

### D.5 A LOWER BOUND FOR UNKNOWN TRANSITION AND ITS PROOF

Below we provide a lower bound for RL with sum segment feedback and unknown transition with the proof.

**Theorem 7.** *Consider the problem of RL with sum segment feedback and unknown transition. There exists a distribution of instances where the regret of any algorithm must be*

$$
\Omega\left(r_{\max}H\sqrt{|\mathcal{S}||\mathcal{A}|K}\right).
$$

*Proof of Theorem 7.* We construct a random instance $\mathcal{I}$ as follows. As shown in Figure 6, there are $n$ bandit states $s_1, \ldots, s_n$ (i.e., there are an optimal action and multiple suboptimal actions), a good absorbing state $s_{n+1}$ and a bad absorbing state $s_{n+2}$. The agent starts from $s_1, \ldots, s_n$ with equal probability $\frac{1}{n}$. For any $i \in [n]$, in state $s_i$, one action $a_J$ is uniformly chosen from $\mathcal{A}$ as the optimal action. In state $s_i$, under the optimal action $a_J$, the agent transitions to $s_{n+1}$ and $s_{n+2}$ with probabilities $\frac{1}{2} + \varepsilon$ and $\frac{1}{2} - \varepsilon$, respectively, where $\varepsilon \in (0, \frac{1}{4})$ is a parameter specified later; Under any suboptimal action $a \in \mathcal{A} \setminus \{s_J\}$, the agent transitions to $s_{n+1}$ and $s_{n+2}$ with equal probability $\frac{1}{2}$.

The rewards are deterministic for all state-action pairs. For any $a \in \mathcal{A}$, $r(s_{n+1}, a) = r_{\max}$. For any $i \in \{1, ..., n, n+2\}$ and $a \in \mathcal{A}$, $r(s_i, a) = 0$.

In this proof, we will also use an alternative uniform instance $\mathcal{I}_{\text{unif}}$. The only difference between $\mathcal{I}_{\text{unif}}$ and $\mathcal{I}$ is that for any $i \in [n]$, in state $s_i$, under all actions $a \in \mathcal{A}$, the agent transitions to $s_{n+1}$ and $s_{n+2}$ with equal probability $\frac{1}{2}$.

Fix an algorithm $\mathbb{A}$. Let $\mathbb{E}_{\text{unif}}[\cdot]$ denote the expectation with respect to $\mathcal{I}_{\text{unif}}$. Let $\mathbb{E}_*[\cdot]$ denote the expectation with respect to $\mathcal{I}$. For any $i \in [n]$ and $j \in [|\mathcal{A}|]$, let $\mathbb{E}_{i,j}[\cdot]$ denote the expectation with respect to the case where $a_j$ is the optimal action in state $s_i$, and $N_{i,j}$ denote the number of episodes where algorithm $\mathbb{A}$ chooses $a_j$ in state $s_i$, i.e., $N_{i,j} = \sum_{k=1}^{K} \mathbb{1}\{\pi_1^k(s_i) = a_j\}$.

The KL divergence of transition distribution on $(s_i, a_J)$ $(i \in [n])$ between $\mathcal{I}_{\text{unif}}$ and $\mathcal{I}$ is

$$
\text{KL}\left(\text{Ber}\left(\frac{1}{2}\right) \| \text{Ber}\left(\frac{1}{2} + \varepsilon\right)\right) = \frac{1}{2} \ln\left(\frac{\frac{1}{2}}{\frac{1}{2} - \varepsilon}\right) + \frac{1}{2} \ln\left(\frac{\frac{1}{2}}{\frac{1}{2} + \varepsilon}\right)
$$

$$
= \frac{1}{2} \ln\left(\frac{\frac{1}{4}}{\frac{1}{4} - \varepsilon^2}\right)
$$

$$
= -\frac{1}{2} \ln\left(1 - 4\varepsilon^2\right)
$$

$$
\overset{(a)}{\leq} 4\varepsilon^2,
$$

where (a) uses the fact that $-\ln(1-x) \leq 2x$ when $x \in (0, \frac{1}{4})$.

In addition, the agent has probability only $\frac{1}{n}$ to arrive at (observe) state $s_i$.

Thus, using Lemma A.1 in (Auer et al., 2002), we have that for any $i \in [n]$, in state $s_i$,

$$
\mathbb{E}_{i,j}[N_{i,j}] \leq \mathbb{E}_{\text{unif}}[N_{i,j}] + \frac{K}{2} \sqrt{\frac{1}{n} \cdot \mathbb{E}_{\text{unif}}[N_{i,j}] \cdot \text{KL}\left(\text{Ber}\left(\frac{1}{2}\right) \| \text{Ber}\left(\frac{1}{2} + \varepsilon\right)\right)}
$$

$$
\leq \mathbb{E}_{\text{unif}}[N_{i,j}] + \frac{K}{2} \sqrt{\frac{1}{n} \cdot \mathbb{E}_{\text{unif}}[N_{i,j}] \cdot 4\varepsilon^2}
$$

$$
= \mathbb{E}_{\text{unif}}[N_{i,j}] + K\varepsilon \sqrt{\frac{1}{n} \cdot \mathbb{E}_{\text{unif}}[N_{i,j}]}.
$$

Summing over $j \in [|\mathcal{A}|]$, using the Cauchy-Schwarz inequality and the fact that $\sum_{j=1}^{|\mathcal{A}|} \mathbb{E}_{\text{unif}}[N_{i,j}] = K$, we have

$$
\sum_{j=1}^{|\mathcal{A}|} \mathbb{E}_{i,j}[N_{i,j}] \leq K + K\varepsilon \sqrt{\frac{|\mathcal{A}|}{n} \cdot K}.
$$

Then, we have

$$
\mathcal{R}(K) = \sum_{k=1}^{K} \mathbb{E}_*\left[V^* - V^{\pi^k}\right]
$$

$$
= \left(\frac{1}{2} + \varepsilon\right)(H-1)r_{\max}K
$$

$$
- \frac{1}{n} \sum_{i=1}^{n} \left(\frac{1}{2}(H-1)r_{\max}K + \varepsilon(H-1)r_{\max} \cdot \frac{1}{|\mathcal{A}|} \sum_{j=1}^{|\mathcal{A}|} \mathbb{E}_{i,j}[N_{i,j}]\right)
$$

$$
\geq \varepsilon(H-1)r_{\max}\left(K - \frac{K}{|\mathcal{A}|} - K\varepsilon\sqrt{\frac{K}{|\mathcal{A}|n}}\right).
$$

Recall that $n = |\mathcal{S}| - 2$. Let $|\mathcal{S}| \geq 3$, $|\mathcal{A}| \geq 2$, $H \geq 2$, $K > |\mathcal{A}|n$ and $\varepsilon = \frac{1}{4}\sqrt{\frac{|\mathcal{A}|n}{K}}$. Then, we have

$$\mathcal{R}(K) = \Omega\left(r_{\max}H\sqrt{|\mathcal{S}||\mathcal{A}|K}\right).$$

$\square$

# E  TECHNICAL TOOLS

In this section, we introduce several technical tools.

**Lemma 34** (Self-concordance, Lemma 9 in (Faury et al., 2020)). *For any $x_1, x_2 \in \mathbb{R}$, we have*

$$\mu'(x_1)\frac{1 - \exp(-|x_1 - x_2|)}{|x_1 - x_2|} \leq \int_{z=0}^{1} \mu'((1-z)x_1 + zx_2)dz \leq \mu'(x_1)\frac{\exp(|x_1 - x_2|) - 1}{|x_1 - x_2|}.$$

*Furthermore, we have*

$$\int_{z=0}^{1} \mu'((1-z)x_1 + zx_2)dz \geq \frac{\mu'(x_1)}{1 + |x_1 - x_2|}.$$

**Lemma 35** (Value Difference Lemma, Lemma E.15 in (Dann et al., 2017)). *For any two MDPs $M'$ and $M''$ with rewards $r'$ and $r''$ and transition distributions $p'$ and $p''$, we have that for any $h \in [H]$ and $s \in \mathcal{S}$,*

$$V_h'(s) - V_h''(s) = \mathbb{E}_{p''}\left[\sum_{t=h}^{H}\left(r'(s_t, a_t) - r''(s_t, a_t) + (p'(\cdot|s_t, a_t) - p''(\cdot|s_t, a_t))^\top V_{h+1}'(\cdot)\right) | s_t = s\right].$$

**Lemma 36** (Law of Total Variance, Lemma 15 in (Zanette & Brunskill, 2019)). *For an MDP $p$ and a fixed policy $\pi$, we have*

$$\mathbb{E}_{\pi,p}\left[\left(\sum_{h=1}^{H} r(s_h, \pi_h(s)) - V_1^\pi(s_1)\right) \Big| s_1\right] = \mathbb{E}_{\pi,p}\left[\sum_{h=1}^{H} \text{Var}_{s_{h+1} \sim p(\cdot|s_h, \pi_h(s_h))}\left(V_{h+1}^\pi(s_{h+1})\right) \Big| s_1\right].$$

The idea of Lemma 36 was also used in earlier works, e.g., (Munos & Moore, 1999; Lattimore & Hutter, 2012; Gheshlaghi Azar et al., 2013).

**Lemma 37** (Lemma 10 in (Ménard et al., 2021)). *For distributions $p, q \in \triangle_\mathcal{S}$ and function $f : \mathcal{S} \to [0, b]$, if $\text{KL}(p, q) \leq \alpha$, then*

$$|(p(\cdot) - q(\cdot))^\top f(\cdot)| \leq \sqrt{2\text{Var}_q(f)\alpha} + \frac{2}{3}b\alpha.$$

**Lemma 38** (Lemma 11 in (Ménard et al., 2021)). *For distributions $p, q \in \triangle_\mathcal{S}$ and function $f : \mathcal{S} \to [0, b]$, if $\text{KL}(p, q) \leq \alpha$, then*

$$\text{Var}_q(f) \leq 2\text{Var}_p(f) + 4b^2\alpha.$$

**Lemma 39** (Lemma 12 in (Ménard et al., 2021)). *For distribution $p \in \triangle_\mathcal{S}$ and functions $f, g : \mathcal{S} \to [0, b]$, we have*

$$\text{Var}_p(f) \leq 2\text{Var}_p(g) + 2bp(\cdot)^\top|f(\cdot) - g(\cdot)|.$$

