# OpenReview forum: "Reinforcement Learning with Segment Feedback"
_ICLR.cc/2025/Conference — Submitted to ICLR 2025_

### Official Review · Reviewer_1qnB · 2024-10-27

**Soundness:** 3
**Presentation:** 3
**Contribution:** 3
**Rating:** 6
**Confidence:** 3

**Summary:**

Classic RL assumes agents receive rewards for every state-action pair, but real-world applications make this costly and difficult. Previous work using trajectory feedback raises efficiency questions, especially with long trajectories. This paper introduces *RL with segment feedback*, a balanced approach between detailed and trajectory feedback. In this model, each episode in an MDP is divided into $m$ segments, with reward feedback provided only at segment ends. We explore two feedback types: binary (binary outcomes) and sum (cumulative rewards). By examining the effect of segment count $m$ on learning, we design algorithms with regret bounds for both feedback types. Our findings show that regret decreases exponentially with more segments under binary feedback, while sum feedback is less affected by increasing $m$.

**Strengths:**

- Addressing the segment feedback problem appears realistic and valuable, especially in applications like autonomous driving or large language models (LLMs). The reviewer finds the real-world examples presented on lines 58–62 particularly convincing.

**Weaknesses:**

Thanks for the paper and your efforts, please check the questions.

**Questions:**

- On line 215, further explanation is needed on how $\nu(k)$ represents the confidence radius of the MLE estimate.
- On line 230, does $m=1$ represent the optimal case for Theorem 1 (minimizing the upper bound of Theorem 1)? If so, what is the purpose of using segments $(m \neq 1)$?
- On line 231, could you clarify why the regret depends on the inverse of the sigmoid function's derivative?
- On line 284, in Theorem 4, what is meant by "ignoring logarithmic factors"? Additionally, isn’t the upper bound a function of $\log(\frac{1}{m})$ that decreases as $m$ increases? Are you suggesting that $m$ is not a critical factor in determining the upper bound?

---

> ### Author Response · Authors · 2024-11-21
> **Response to Reviewer 1qnB**
>
> Thank you very much for your recognition of our work and insightful questions! We have revised our paper according to your comments, and highlighted the revised concent in *purple* font.
>
> **Response to Questions**
>
> **1. The Implication of $\nu(k)$**
>
> With high probability, we have
> $
> |\phi^\top \theta^* - \phi^\top \hat{\theta}_k| \leq \sqrt{\alpha} \cdot \nu(k) \cdot
> $
>
> $
> \\|\phi\\|_{\Sigma_k^{-1}} ,
> $
>
> where $\theta^*$ is the true reward parameter, $\hat{\theta}_k$ is the MLE estimate, and $\phi$ is the visitation indicator of a trajectory.
> Thus, $\nu(k)$ is part of the bound for the error between the estimated reward and the true reward for a trajectory. (There is a break in the formula due to the formula display issue of OpenReview.)
>
>
> **2. Theorem 1**
>
> In Theorem 1, the regret bound contains a factor $\exp(\frac{H r_{\max}}{2m})$, which usually dominates the bound. Thus, when the number of segments $m$ increases, the regret bound decreases at an exponential rate. Therefore, under binary feedback, increasing the number of segments helps accelerate learning.
>
> **3. The Sigmoid Function's Derivative**
>
> Intuitively, we want to distinguish between "good" actions (which give high rewards) and "bad" actions (which give low rewards) from binary feedback, which is generated by the sigmoid function $\mu(x)=\frac{1}{1+\exp(-x)}$.
> If the binary feedback is generated from the range where the sigmoid function is flat, i.e., the derivative of the sigmoid function $\mu'(x)$ is small, then the generated binary feedback is likely  always $0$ or always $1$, and it is hard to distinguish between a "good" action and a "bad" action, leading to a higher regret. On the contrary, if the binary feedback is generated from the range where the sigmoid function is steep, i.e., the derivative of the sigmoid function $\mu'(x)$ is large, then the generated binary feedback is more dispersed and will not always be $0$ or $1$, and it is easier to distinguish between a good action and a bad action, leading to a lower regret. Therefore, the regret bound depends on the inverse of the sigmoid function's derivative.
>
>
> **4. Theorem 4**
>
> We meant that in Theorem 4 (the sum feedback setting), as the number of segments $m$ increases, the regret bound does not decrease at a polynomial rate as one might expect, e.g., $\frac{1}{m}$ or $\frac{1}{\sqrt{m}}$, which is a little surprising, because larger $m$ means more observations.
>
> Yes, since $m$ influences the regret bound by a multiplicative constant of the form $\log(1+\frac{H}{m})$, it is not a critical factor affecting the regret bound in Theorem 4.

---

> ### Comment · Reviewer_1qnB · 2024-11-26
>
> Dear Author,
>
> Thank you for the comments. Answers are reasonable and all of my concerns are addressed. I will keep my score and the confidence. Thanks!
>
> Best,
> Reviewer

---

> > ### Author Response · Authors · 2024-11-26
> > **Thank you again for your recognition of our work!**
> >
> > Thank you again for your time in reviewing our paper and your recognition of our work!

---

### Official Review · Reviewer_NUmy · 2024-10-31

**Soundness:** 2
**Presentation:** 3
**Contribution:** 2
**Rating:** 6
**Confidence:** 3

**Summary:**

This paper studies the problem of RL with segment feedback in the contexts of binary feedback and sum feedback. In these settings, each episode is divided into m equal-length segments, and the agent receives feedback at the end of each segment. For the binary feedback setting, the authors propose TS-style algorithms, while for the sum feedback setting, they design UCB-style algorithms, and they provide analyses of the regret upper bounds, regret lower bounds for known transitions, and regret upper bounds for unknown transitions. Their findings show that, under binary feedback, increasing the number of segments m decreases the regret at an exponential rate, while under sum feedback, increasing m does not significantly reduce the regret.

**Strengths:**

1. The paper provides detailed theoretical analysis and proofs.
2. The E-optimal experimental design for sum feedback is very effective and interesting.
3. The conclusion that under sum feedback, increasing m does not significantly reduce the regret is intriguing, and the experimental results support this finding.

**Weaknesses:**

1. In terms of the setting, previous work has also proposed the segment feedback setting [1], although under a different name.
2. The assumption that each episode is divided into m segments of equal length is somewhat strong.
3. In the binary segment feedback setting (using a TS-based algorithm), the method for estimating the reward function is the same as in [2] (UCBVI with trajectory labels). Apart from replacing the UCB-based algorithm with a TS-based algorithm, what is the innovation in this algorithm? Additionally, the regret upper bound of the proposed algorithm SegBiTS appears worse than that of UCBVI with trajectory labels in a fair comparison when m=1. Given this, what are the specific advantages of using SegBiTS?
4. It is straightforward to extend trajectory feedback algorithms to the segment feedback setting. If the proposed algorithm has advantages, they should be clearly stated, and a experimental comparison with trajectory feedback algorithms [2,3] should also be included.

[1] Reinforcement Learning from Bagged Reward.
[2] On the Theory of Reinforcement Learning with Once-per-Episode Feedback.
[3] Reinforcement Learning with Trajectory Feedback.

**Questions:**

1. Why was the regret lower bound not analyzed in the case of unknown transitions?
2. Why was a Thompson Sampling-type algorithm chosen for binary segment feedback, while a UCB-type algorithm was used for sum segment feedback?

I will review your response and am open to further discussion.

---

> ### Author Response · Authors · 2024-11-21
> **Response to Reviewer NUmy (Part 1/3)**
>
> Thank you very much for your time and effort in reviewing our paper! We have revised our paper according to your comments, and highlighted the revised concent in *blue* font.
>
> **Response to Weaknesses**
>
> **1. Prior Work [1] on Segments**
>
> Thank you for suggesting this reference! We were not familiar with [1] before. We acknowledge that [1] proposed the setting of RL with segment feedback earlier, and have cited and discussed [1] in our revision.
>
> However, we would like to highlight the differences between our work and [1] below.
>
> - Prior work [1] is mostly an empirical work. They design an algorithm based on transformers, and do not present any theoretical guarantee except a theorem proving that the optimal policies for MDPs with segment feedback align with those for standard MDPs. Different from [1], our work is mostly a theoretical RL work. We focus more on providing theoretical analysis to show how the number of segments $m$ influences learning performance. We design provably efficient algorithms, and establish rigorous regret upper and lower bounds for our algorithms and the problem itself, respectively.
> - Prior work [1] only considers the sum reward feedback type, while we consider both the sum and binary reward feedback types. Binary reward feedback is more suitable for scenarios involving human feedback. For example, in autonomous driving, it is easier for humans to provide feedback on whether a driving segment is good than giving a score. We also design algorithms and present theoretical results to show that under binary feedback, as $m$ increases, the regret decreases at an exponential rate.
> - Our work contributes new techniques. We use E-optimal experimental design to perform an initial exploration in our algorithm, which enables us to improve the result in prior trajectory feedback work [3]. We also develop a novel lower bound analysis for binary feedback, based on the KL divergence of Bernoulli distributions with the sigmoid function as parameters. Our techniques can be of independent interests, and applied to other related RL problems.
>
>
> **2. The Setting of Equal Length**
>
> We agree with the reviewer that considering segments of unequal lengths is more general and an interesting future direction, which was also mentioned in our future work section.
> However, we would like to highlight that studying segments of equal lengths is **an important starting point and serves as a foundation** for further investigation, especially for theoretical analysis. We believe that **our result that segments help in the case of binary feedback and do not help in the case of sum feedback is quite surprising and not known previously**.
>
>
> Note that **studying segments of equal lengths is already very challenging**, due to the following reasons:
>
> - This problem cannot be solved by simply adapting prior trajectory feedback works, e.g., [Efroni et al., AAAI 2021], because they use the martingale property of subsequent trajectories, while subsequent segments are not a martingale due to the dependence among segments within a trajectory.
> - For sum feedback, there was a gap between existing upper and lower bounds for RL with trajectory feedback [Efroni et al., AAAI 2021]. So one cannot infer that segments would not help significantly in the sum feedback case (our result) from prior works.
> - For binary feedback, there is no known lower bound to indicate that the exponential factor in the regret bound is inevitable.
>
> Our paper has made a number of technical contributions to overcome these challenges.
>
> **3. Advantages and Innovation of Algorithm SegBiTS**
>
> The advantage of our SegBiTS algorithm compared to [2] is in computation. Algorithm "UCBVI with trajectory labels" [2] is computationally inefficient due to its confidence interval term. While [2] also provides another computationally-efficient algorithm called "UCBVI with trajectory labels and added exploration", this algorithm has a worse regret bound dependent on $K^{\frac{2}{3}}$, where $K$ is the number of episodes. In contrast, our SegBiTS algorithm is **computationally efficient and easy to implement**, and has a regret bound dependent on $\sqrt{K}$.
>
> SegBiTS adopts the maximum likelihood estimator for binary feedback under the Thompson sampling framework.
> Its analysis is highly non-trivial. In the analysis, we decompose the regret into  the error due to reward parameter estimation and the error due to sampled Gaussian noises, and carefully choose the conditional expectation to ensure the independence between sampled Gaussian noises and segment visitation indicators.
>
> We would like to also note that our novelty in the binary feedback setting also includes a new lower bound (Theorem 2) to demonstrate the inevitability of the exponential factor $\exp(\frac{H r_{\max}}{m})$ in the regret bound. This lower bound is novel and, to the best of our knowledge, not known previously.

---

> ### Author Response · Authors · 2024-11-21
> **Response to Reviewer NUmy (Part 2/3)**
>
> **4. Advantages of Our Algorithms and Experiments**
>
> **(1) Advantages of Our Algorithms**
>
> We would like to highlight that it is not straightforward to extend trajectory feedback algorithms to the segment feedback setting, especially for attaining theoretical results that clearly show the influence of the number of segments $m$ on learning performance. The challenges and the advantages of our algorithms include:
>
> - At a high level, trajectory feedback algorithms cannot be directly used in the case of segment feedback, because segments are strongly correlated and do not lead to a martingale as in the trajectory feedback case. Therefore, learning algorithms in the case of segment feedback have to be carefully designed to account for this correlation.
> - For sum feedback, prior trajectory feedback algorithm [3] is not optimal. To tackle this challenge, we design an algorithm E-LinUCB, which adopts E-optimal experimental design to sharpen the norm of visitation indicator $\\|\phi^{\tau}\\|$, and achieves **an optimal regret bound** with respect to $m$ and the length of each episode $H$. **Our result improves on [3]** when the problem degenerates to trajectory feedback, and reveals the optimal dependence on $m$ in the regret bound.
> - For binary feedback, prior trajectory feedback algorithms [2] are either computationally inefficient or have an $O(K^{\frac{2}{3}})$ regret bound, where $K$ is the number of episodes. To handle this challenge, we develop algorithm SegBiTS, which incorporates the maximum likelihood estimator for binary feedback into the Thompson sampling framework. SegBiTS is both **computationally efficient** and has an $O(\sqrt{K})$ regret bound. We also carefully handle the error brought by such incorporation in analysis, e.g., using a variance-adaptive concentration analysis to bound the reward estimation error and choosing the appropriate conditional expectation to bound the error due to sampled Gaussian noises.
>
>
> **(2) Experiments to Compare with [2, 3]**
>
> Thank you for your suggestion on experiments.
> We ran preliminary experiments to compare our algorithms with the adaptations of prior trajectory feedback algorithms [2, 3] to the segment setting.
>
> Since algorithm “UCBVI with trajectory labels” in [3] is computationally inefficient, in experiments, we use a small MDP with $|\mathcal{S}|=3$ and $|\mathcal{A}|=5$, and set $r_{\max}=0.5$, $\delta=0.005$ and $m \in \\{1,2,H\\}$. In the binary feedback setting, we set $H=6$ and $K=1000$, and in the sum feedback setting, we set $H=10$ and $K=5000$ (because ``UCBVI with trajectory labels" is slow for large $H$ and $K$). We perform $15$ independent runs for each algorithm, and present the average cumulative regret up to episode $K$, the standard deviation of the regret and running time across runs.
>
> | Binary Feedback ($H=6, m=1$) |  Regret (Std.)    |  Running Time (s)    |
> | -------- | -------- | -------- |
> | SegBiTS (ours)    | 363.23 (7.12)    | 35.79     |
> | "UCBVI with trajectory labels" [2]     | 285.05 (19.83)   | 145.54     |
>
> | Binary Feedback ($H=6, m=2$) |  Regret (Std.)    |  Running Time (s)    |
> | -------- | -------- | -------- |
> | SegBiTS (ours)    | 178.86 (12.62)    | 34.02     |
> | Adaptation of "UCBVI with trajectory labels" [2]     | 173.10 (13.82)    | 142.96     |
>
> | Binary Feedback ($H=6, m=6$) |  Regret (Std.)    |  Running Time (s)    |
> | -------- | -------- | -------- |
> | SegBiTS (ours)    | 80.10 (5.40)    | 34.77     |
> | Adaptation of "UCBVI with trajectory labels" [2]     | 64.31 (6.65)    | 144.43     |
>
>
>
> | Sum Feedback ($H=10, m=1$) |  Regret (Std.)    |  Running Time (s)    |
> | -------- | -------- | -------- |
> | E-LinUCB (ours)    | 4433.23 (24.09)    | 919.19     |
> | OFUL [3]      | 4577.77 (39.86)    | 904.06     |
>
> | Sum Feedback ($H=10, m=2$) |  Regret (Std.)    |  Running Time (s)    |
> | -------- | -------- | -------- |
> | E-LinUCB (ours) | 3954.66 (29.73)    | 940.15     |
> | Adaptation of OFUL [3]  | 4180.95 (49.35)    | 925.60     |
>
> | Sum Feedback ($H=10, m=10$) | Regret (Std.) | Running Time (s) |
> | -------------------------- | ------------- | ---------------- |
> | E-LinUCB (ours) | 2105.70 (14.34)         | 968.14             |
> | Adaptation of OFUL [3]  | 2272.46 (19.32)         | 946.48             |
>
> From the tables, we see that under binary feedback, while our algorithm SegBiTS has a slightly higher regret than the adaptation of algorithm “UCBVI with trajectory labels” [3], SegBiTS attains a significantly lower running time than the adaptation of ``UCBVI with trajectory labels", which demonstrates the computation efficiency of SegBiTS. Under sum feedback, our algorithm E-LinUCB achieves a lower regret than OFUL [2], which validates the effectiveness of its experimental design technique.
>
> We will include experiments with more independent runs in the revision once they finish.

---

> > ### Author Response · Authors · 2024-11-23
> > **Response to Reviewer NUmy (Part 3/3)**
> >
> > **Response to Questions**
> >
> > **1. Lower Bounds for the Unknown Transition Case**
> >
> > The regret lower bounds for the known transition case also hold for the unknown transition case, by constructing the same problem instances as those in the lower bound proofs for the known transition case, where the reward is random and the transition is deterministic.
> >
> > **2. Choice of Algorithm Type**
> >
> > One can use Thompson sampling or UCB in both cases. However, analytically to show that segments do not significantly improve performance in the case of sum feedback, we needed to get tight bounds which is easier to do with UCB. On the other hand, Thompson sampling is more computationally efficient and provided the same type of exponential regret matching the lower bound (up to constants) in the case of binary feedback. But the reviewer is correct in that we could have used UCB in the case of binary feedback as well to get a similar upper bound.
> >
> > Thank you again for agreeing to read about rebuttal. If you have any further questions, please let us know and we will be happy to address them.
> >
> >
> > ---
> >
> > References:
> > [1] Yuting Tang et al. Reinforcement learning from bagged reward. ICML Workshop, 2024.
> > [2] Niladri Chatterji et al. On the theory of reinforcement learning with once-per-episode feedback. NeurIPS, 2021.
> > [3] Yonathan Efroni et al. Reinforcement learning with trajectory feedback. AAAI, 2021.

---

> > > ### Comment · Reviewer_NUmy · 2024-11-24
> > >
> > > Thank you for your detailed response. Most of my concerns have been addressed, and I’m willing to raise my score to 6.

---

> > > > ### Author Response · Authors · 2024-11-24
> > > > **Thank you very much for increasing your score**
> > > >
> > > > Thank you very much for increasing your score and your effort in reviewing our paper!

---

### Official Review · Reviewer_R7pS · 2024-11-09

**Soundness:** 2
**Presentation:** 3
**Contribution:** 2
**Rating:** 6
**Confidence:** 4

**Summary:**

The paper studies RL with segment feedback. In this setting, the episodes $H$ are assumed to be equally divided into $m$ segments. The agent observes reward feedback only at the end of each segment, instead of observing feedback for every action. The authors design efficient algorithms that establish regret upper and lower bounds for both binary reward feedback and sum reward feedback settings. Furthermore, the authors provide empirical results to show the advantage of their algorithms.

**Strengths:**

1. The paper is well written and the theoretical results appear to be correct.
2. The paper generalizes and improves previous results in (Efroni et al., 2021).
3. The paper provides empirical simulation results to show the advantages of their algorithms.

**Weaknesses:**

1. The regret bound provided in this article is unclear. For example, in Theorem 1, the authors represent many terms that are polynomially dependent on $S, A, H$ using $v(K)$, making the regret bound appear more optimal. The authors should clarify the order of their regret in terms of $A,A,H$.
2. The assumption of equal segmentation is a bit too strong. In practice, the aggregated reward feedback is rarely uniform. In such cases, an algorithm that can adapt to the level of aggregation would make more sense.

**Questions:**

1. See weakness.
2. What happens if $ r_{\text{max}}$is greater than 1? Intuitively, $ r_{\text{max}} $ can take any value, since at most the regret is scaled by $ \exp(r_{\text{max}} \cdot H / m) $.
3. What is the main difference between the two types of reward feedback. Intuitively, logistic feedback compresses the reward information into the $[0, 1 ]$ range. As a result, the regret for logistic feedback should be $\exp(m \cdot r_{\text{max}} / H)$ times larger than that for sum regret (as stated in a similar result in Theorem 2 of the paper). I'm curious why we can't directly apply the sum reward algorithm to the logistic reward case (just repeat playing every policy $\exp(m \cdot r_{\text{max}} / H)$ times to ensure that we can get enough information).

---

> ### Author Response · Authors · 2024-11-21
> **Response to Reviewer R7pS (Part 1/2)**
>
> **Response to Weaknesses**
>
> Thank you very much for your time and effort in reviewing our paper! We have revised our paper according to your comments, and highlighted the revised concent in *red* font.
>
> **1. Unclear Regret Bound**
>
> We would like to clarify that we used $\nu(K)$ in Theorem 1 **just to keep the presentation of equations concise**. Actually, **such notation was also used in many prior logistic bandit works**, e.g.,~[Faury et al., 2020; Russac et al., 2021], and **we just followed their notation**.
>
> However, we agree with the reviewer that a complete regret bound will make the result clearer.
> We present the complete regret bound as follows, and have included it in our revision.
>
> $$
> R(K) = \tilde{O} \Bigg( \exp\bigg(\frac{H r_{\max}}{2m}\bigg) \cdot \bigg( \sqrt{ Km |S| |A| \max \Big\\{ \frac{H^2}{m \alpha \lambda}, 1 \Big\\} } + H\sqrt{ \frac{K}{\alpha \lambda} }  \bigg) \cdot \frac{m \sqrt{\lambda |S| |A|}}{H} \cdot
> $$
> $$
> \Bigg( 1 + \frac{H r_{\max} \sqrt{|S| |A|}}{m}  + \frac{H}{m\sqrt{\lambda}}  \sqrt{ 1 + \frac{H r_{\max} \sqrt{|S| |A|}}{m} } \bigg( \sqrt{\lambda} \Big( r_{\max}\sqrt{|S| |A|} + \frac{1}{2} \Big) + \frac{|S| |A|}{\sqrt{\lambda}} \bigg) + \frac{H^2}{m^2 \lambda} \cdot \bigg( \sqrt{\lambda} \Big( r_{\max}\sqrt{|S| |A|} + \frac{1}{2} \Big) + \frac{|S| |A|}{\sqrt{\lambda}} \bigg)^2  \Bigg)^{\frac{3}{2}}
> \Bigg)
> $$
>
> The dependence on $|\mathcal{S}|$, $|\mathcal{A}|$, $H$ in this regret bound are $|\mathcal{S}|^3$, $|\mathcal{A}|^3$, $\exp(\frac{H r_{\max}}{2m}) H^2$, respectively.
> Our focus was not on the polynomial factors; our main goal was to show the exponential dependence that arises due to binary segment feedback and, in this case, we note that we provide a lower bound that is also exponential (Theorem 2).
>
>
> **2. The Setting of Equal Length**
>
> We agree with the reviewer that considering segments of unequal lengths is more general and an interesting future direction, which was also mentioned in our future work section.
> However, we would like to highlight that studying segments of equal lengths is **an important starting point and serves as a foundation** for further investigation, especially for theoretical analysis. We believe that **our result that segments help in the case of binary feedback and do not help in the case of sum feedback is quite surprising and not known previously**.
>
>
> Note that **studying segments of equal lengths is already very challenging**, due to the following reasons:
>
> - This problem cannot be solved by simply adapting prior trajectory feedback works, e.g., [Efroni et al., AAAI 2021], because they use the martingale property of subsequent trajectories, while subsequent segments are not a martingale due to the dependence among segments within a trajectory.
> - For sum feedback, there was a gap between existing upper and lower bounds for RL with trajectory feedback [Efroni et al., AAAI 2021]. So one cannot infer that segments would not help significantly in the sum feedback case (our result) from prior works.
> - For binary feedback, there is no known lower bound to indicate that the exponential factor in the regret bound is inevitable.
>
> Our paper has made a number of technical contributions to overcome these challenges.
>
> **Response to Questions**
>
> **1. The Case $r_{max}>1$**
>
> Thank you for raising this question. Our algorithms and results also work for $r_{\max}>1$.  We have revised $r_{\max}\in(0,1]$ to $r_{\max}>0$ in our updated version.
>
>
> **2. The Difference between the Two Feedback Types**
>
> (i) The reward estimation methods under the sum feedback and binary feedback are different. In the sum reward feedback setting, we mainly use the least square estimator to estimate the reward parameter, and the confidence interval is a confidence ellipsoid related to the variances of noises. In contrast, in the binary reward feedback setting, we use the maximum likelihood estimator to estimate the reward parameter, and the confidence interval is a confidence ellipsoid, related to not only the variances of noises but also the derivative of the sigmoid function. (ii) The analyses  needed for these two feedback types are different. In the sum reward feedback setting, we mainly use the concentration analysis of least squares to bound the reward estimation error; In the binary reward setting, we use a more complex variance-adaptive concentration analysis and the self-concordance property of the sigmoid function's derivative to handle the reward estimation error.
>
> We do not think that the binary reward feedback setting can be trivially solved by running the algorithm for sum reward feedback for $\exp(\frac{H r_{\max}}{m})$ times. An immediate reason is that the least squares estimator and confidence interval construction in the algorithm for sum reward feedback cannot directly handle binary reward feedback, and the analysis of the algorithm for sum reward feedback also cannot give a reward estimation concentration guarantee for binary reward feedback.

---

> > ### Author Response · Authors · 2024-11-21
> > **Response to Reviewer R7pS (Part 2/2)**
> >
> > ---
> >
> > References:
> > [1] Louis Faury et al. Improved optimistic algorithms for logistic bandits. ICML, 2020.
> > [2] Yoan Russac et al. Self-concordant analysis of generalized linear bandits with forgetting. AISTATS, 2021.
> > [3] Yonathan Efroni et al. Reinforcement learning with trajectory feedback. AAAI, 2021.

---

> > > ### Author Response · Authors · 2024-11-26
> > > **We wonder if our response has addressed your concerns**
> > >
> > > Dear Reviewer R7pS,
> > >
> > > Thank you for your time in reviewing our paper again!
> > >
> > > Since the author-reviewer discussion period will end soon, we are wondering if our response has addressed your concerns? If you have any further questions, we are more than happy to answer.
> > >
> > > Best regards,
> > > Authors

---

> > > > ### Comment · Reviewer_R7pS · 2024-11-26
> > > >
> > > > Thank you for your detailed response!
> > > >
> > > > For the last question, the key idea is that given sigmoid function $y=\mu(x)$, if we have a confidence interval with width $c$ for $y$, we can immediately have a confidence interval with width $\exp(m r_{max}/H) c$ for $x$, since the gradient of $\mu(x）$ is at most $\exp(m r_{max}/H)$. In this sense, there is no main difference.
> > > >
> > > > Anyway thanks for the reply and most of my concerns have been addressed. I raise my score to 6.

---

> > > > > ### Author Response · Authors · 2024-11-26
> > > > > **Thank you very much for increasing your score**
> > > > >
> > > > > Thank you very much for increasing your score and providing this idea! We will further investigate this direction in our future work.

---

### Meta-Review · Area_Chair_7MhK · 2024-12-22

**Metareview:**

This work studies an intermediate setting of feedback in Reinforcement Learning that lies between the traditional scenario in RL with feedback only per step and once-per-episode feedback where the feedback is only given at the end of the trajectory. Since there are two models of trajectory feedback: sum of rewards and binary, the authors study a segment feedback scenario where the reward signal is revealed as an aggregate of a fixed set of segments within a trajectory. The authors propose several theoretical results, including upper and lower bounds for this setting.

The reviewers seem to have agreed this is an interesting direction. Unfortunately I have reservations about the technical contribution. In particular, it is known that in the binary feedback setting the optimal policy is non markovian. This is one of the distinguishing properties that separate sum feedback and binary feedback. There is no trace of this fact in the manuscript. Moreover, because of this, it is unclear how would one compute efficiently (as the authors claim) an optimal policy. The lack of this discussion means the results are not comparable with the previous work on binary trajectory feedback and in particular the claims of computational efficiency are put on the spot, since this is achieved only because of the reduced nature of the benchmark. I recommend the authors to revise this work to address these comments. One way to do so is to define the comparator class as the set of markovian policies (as the authors seem to be implicitly doing) but then it needs to be specified somewhere that because of this their results are not comparable with the previous literature.

**Additional Comments On Reviewer Discussion:**

As mentioned above, concerns about the lack of any discussion about the non-markovian nature of the best comparator policy in the binary feedback case is concerning. The authors discussion with reviewer R7pS about the difference between binary and sum feedback and how one can be reduced to the other, and the lack of answer regarding the non-markovian nature of the optimal binary feedback policies reveals the authors may have been unaware of this even though this is the chief difference between these two feedback modes, and not the loss used for estimating the reward parameters. This is why I am recommending rejection.

---

### Decision · Program_Chairs · 2025-01-22

Reject